# Rhizosheath inhabiting *Massilia* are linked to heterosis in roots of maize

Xiaoming He[1,2,3,4,9], Ling Gu[1,3,4,9], Danning Wang[5], Marcel Baer[3,4], Gabriel Schaaf [6], Antonios Apostolakis [7,8], Ana Meijide [8], Xinping Chen [2] ✉, Frank Hochholdinger[4] ✉ & Peng Yu [1,3] ✉

Heterosis, or hybrid vigor, describes the superior performance of $F_1$ hybrids compared to parental inbreds. While soil microbiomes are proposed to influence heterosis, it remains unclear how heterotic plants shape their microbiomes and how interactions relate to stress responses. Here, we investigate the role of rhizosheath formation—the soil tightly adhering to roots—in maize heterosis under nitrogen deprivation. Across sterilization, inoculation, and transplantation experiments, hybrids develop larger rhizosheaths than inbreds, and rhizosheath size associates with biomass heterosis. Rhizosheath-enriched genus *Massilia* correlates with lateral root density, rhizosheath size, and growth. Untargeted metabolomics and flavone-deficient mutants reveal links between *Massilia* and flavonoid pathways, while growth promotion by *Massilia* can also occur independently of host flavones. Metagenomic analysis shows that larger rhizosheaths recruit microbial functions related to nutrient cycling and stress adaptation. These findings identify rhizosheath formation as an integrative trait associated with heterosis and a promising target for breeding resilient crops.

Beneficial plant-microbiome interactions offer enormous potential for improving plant growth and abiotic stress tolerance in agriculture[1,2]. Understanding the mechanisms of plant microbiome assembly and the modification of crop microbiomes by plants has enormous potential to enhance agricultural sustainability and food security in the face of climate change[3]. The utilization of heterosis in maize breeding - the phenomenon that hybrids outperform their parental inbred lines - was one of the landmark innovations of 20th century agriculture[4,5]. Despite the enormous potential of root traits for crop improvement, the involved mechanisms driving heterosis manifestation in the root system remain poorly understood[6].

Plant root systems exhibit remarkable developmental plasticity, which contributes to their broad adaptation to diverse environmental conditions[7]. The complex interactions between crop root systems and their rhizosphere - the narrow soil volume directly influenced by root traits and their exudates - play a critical role in shaping the diversity and composition of the rhizosphere microbiome[8,9]. The rhizosheath is considered an adaptive and stress responsive root trait[10], typically defined as the mass of soil that remains tightly attached to roots upon excavation[11,12]. Importantly, the rhizosheath is distinct from the rhizosphere: while the rhizosheath refers to the firmly adhering soil layer bound by root hairs, mucilage, and microbial activity, the rhizosphere encompasses the broader volume of soil influenced by root exudates and associated microbial processes[13–15]. The heritability of the rhizosphere microbiome has been well documented[16,17], but it remains unclear whether beneficial microbiomes can be inherited from

[1]Plant Genetics, TUM School of Life Sciences, Technical University of Munich (TUM), Freising, Germany. [2]College of Resources and Environment and Academy of Agricultural Sciences, Southwest University (SWU), Chongqing, China. [3]Emmy Noether Group Root Functional Biology, Institute of Crop Science and Resource Conservation (INRES), University of Bonn, Bonn, Germany. [4]Crop Functional Genomics, Institute of Crop Science and Resource Conservation (INRES), University of Bonn, Bonn, Germany. [5]Plant Breeding, TUM School of Life Sciences, Technical University of Munich (TUM), Freising, Germany. [6]Plant Nutrition, Institute of Crop Science and Resource Conservation (INRES), University of Bonn, Bonn, Germany. [7]Department of Crop Sciences, University of Göttingen, Göttingen, Germany. [8]Environment Modeling, Institute of Crop Science and Resource Conservation (INRES), University of Bonn, Bonn, Germany. [9]These authors contributed equally: Xiaoming He, Ling Gu. ✉e-mail: chenxp2017@swu.edu.cn; hochholdinger@uni-bonn.de; pengyu.yu@tum.de

parental plants to their offspring and to what extent these specifically inherited microbial taxa contributes to the performance and resilience of $F_1$-hybrid varieties. Differences in gene expression, morphology and physiological characteristics between hybrids and their parental inbred lines are observed in cereal roots[4,5,18,19]. These differences might explain, at least in part, the observed diversity and composition of the microbiome during the manifestation of heterosis, a phenomenon previously introduced as microbiome heterosis —where microbial taxa in hybrids deviate from midparent expectations in abundance or presence. It was demonstrated that the soil microbiome can influence heterosis, particularly with respect to maize biomass. For example, a maize field experiment showed that inbred lines often perform better in sterile soil, whereas hybrids exhibit less sensitivity to microbial presence, suggesting that the microbiome might disproportionately inhibit inbred performance rather than enhance hybrid vigor[20]. However, the impact of the soil microbiome on the manifestation of heterosis, particularly for belowground root traits, remains unclear. It is also uncertain whether the interaction between complex root traits and the soil microbiome contributes to heterosis manifestation, thereby helping crops to adapt to abiotic stress.

In this study, we profiled the root and rhizosphere microbiomes of diverse maize inbred-hybrid triplets grown under different abiotic conditions using 16S rRNA gene and internal transcribed spacer 1 (ITS1) gene sequencing to explore the existence of heterosis for microbial taxa under abiotic stress. We then investigated whether the microbial taxa (*Massilia*) is associated with maize heterosis by regulating root development and metabolism in nitrogen-poor soil. Furthermore, we examined the role of the rhizosheath - a heritable root trait integrated in root metabolism and the associated rhizosphere microbiome - in maize heterosis. Understanding the association of beneficial microbiomes with heterosis manifestation through rhizosheath modulation and interactions with root traits under abiotic stress provides insights, that could contribute to develop better crops resilient to environmental challenges.

## Results

### Genotype dependent rhizosphere microbiome shifts are linked with maize heterosis

The goal of this study was to explore how maize heterosis influences root–associated microbiome interactions and their potential to enhance crop performance under abiotic stresses. We investigated microbiome assembly in a diverse panel of ten maize inbred lines that we selected from diverse heterotic groups (stiff-stalk, non-stiff-stalk and tropical varieties) and crossed them with the stiff-stalk inbred line B73 as the maternal parent (Fig. 1a and Supplementary Data 1). We assessed the response of maize growth for inbred lines and hybrids to various soil abiotic stress conditions, including drought (D), low nitrogen (Low N) and low phosphorous (Low P). To this end, we collected root and rhizosphere samples from the first-whorl of crown roots which displayed fully developed lateral roots, after a four-week growth period in soil in a phytochamber. We then profiled the root and rhizosphere microbiomes of these samples using 16S rRNA gene and ITS1 gene sequencing (Fig. 1b). The experimental soil was collected from the Dikopshof long-term field fertilization experiment (50° 48′ 21′′ N, 6° 59′ 9′′ E) of the University of Bonn, which has been subjected to a fixed fertilization regime for over 100 years, including plots that are well-fertilized with all essential nutrients, as well as plots that have received no nitrogen or no phosphorus supplementation during this period[21].

Maize hybrids consistently outperformed their parental inbred lines across all abiotic soil stress conditions tested (Fig. 1c). To quantify this effect, we calculated midparent heterosis (MPH) for shoot biomass defined as the percentage increase of a hybrid relative to the average of its two parents. MPH was evaluated under control conditions and under three stress treatments including low nitrogen, low phosphorus

and drought. Specifically, low nitrogen stress significantly increased the MPH of shoot biomass compared to the control treatment (Linear mixed model, $p = 5.84e-7$, Fig. 1d). This indicates that biomass heterosis is most pronounced under nitrogen deprivation and might be influenced by genotype–environment interactions, including root traits and microbial associations.

We next examined whether hybrid genotypes affect microbiome community composition. Although differences between inbreds and hybrids were modest, they were statistically significant (Supplementary Fig. 1). Similar findings have been reported in maize[22,23], supporting the view that hybrid breeding influences microbiome assembly in subsequent crop varieties[24,25]. To further test the genetic potential for microbiome optimization in breeding, we assessed heterosis for individual amplicon sequence variants (ASVs) using FDR-corrected $t$-tests of their variance-stabilized abundances (Supplementary Data 2). In this context, heterosis for an ASV was defined as a significant deviation in hybrid abundance relative to the midparent expectation. Across most crossing combinations, the majority of the rhizosphere bacterial ASVs exhibited some degree of midparent heterosis, although the prevalence of heterosis varied across hybrids and taxa (Fig. 1e). Among those, abundance of *Oxalobacteraceae* displayed heterosis in the rhizosphere of several hybrids (B73 × H99, B73 × H84, B73 × A554 and B73 × Mo17; Fig. 1e) in nitrogen-poor soil. Previously, we demonstrated that *Oxalobacteraceae* represent a heritable and stress-responsive component in the maize microbiome that can help to cope with low nitrogen soil stress[17]. Taken together, these results highlight that *Oxalobacteraceae* are a consistently enriched and potentially heritable microbial lineage in maize hybrids under nitrogen deprivation, making them a strong candidate for further investigation of microbiome contributions to heterosis in this stress context.

### *Massilia*-mediated lateral root development associates with maize heterosis under nitrogen stress

Based on our finding that *Oxalobacteraceae* were consistently enriched in hybrids, we tested whether they are associated with the manifestation of heterosis. We first conducted a soil sterilization experiment using three inbred lines (B73, Mo17, H84) and their $F_1$-hybrids (B73 × Mo17, B73 × H84, Mo17 × H84), which enrich *Oxalobacteraceae* under both high- and low-nitrogen conditions. To validate the reproducibility of the observed microbiome–root trait patterns independent of B73, we included the hybrid Mo17 × H84 in our analysis. In nitrogen-poor soil, the absence of the soil microbiome significantly enhanced heterosis for shoot biomass (B73 × Mo17, $p = 0.0012$; Mo17 × H84, $p = 0.024$; Fig. 2a). These results support the notion that the presence or absence of a microbiome might modify the manifestation of heterosis, particularly under nitrogen deficiency.

In a previous study, we demonstrated that a specific *Massilia* isolate, a member of the *Oxalobacteraceae* family, enhanced lateral root formation in maize mutants with defective lateral root formation, enabling them to adapt to nitrogen-poor soil[17]. Based on this, we tested whether the same *Massilia* isolate is associated with heterosis for root development. Controlled inoculation experiments were performed with the three heterotic combinations (B73 × Mo17, B73 × H84, and Mo17 × H84) in nitrogen-poor soil. Plants were grown with either (i) the *Massilia* isolate alone, (ii) a 17-member synthetic community excluding this isolate (SynCom17), or (iii) an 18-member community including both (SynCom18) (Supplementary Data 3). Root traits were quantified from the first whorl of crown roots, which were also used for single-root microbiome sequencing. Inoculation with the single *Massilia* isolate significantly promoted lateral root formation in comparison to the mock control in both inbred lines and hybrids. In contrast, SynComs with or without the isolate produced weaker or inconsistent effects (Fig. 2b). In particular, heterosis for lateral root density strongly correlated with heterosis for shoot dry biomass under nitrogen-poor conditions (Pearson's $R = 0.61$, $p = 0.0027$, Fig. 2c), suggesting that

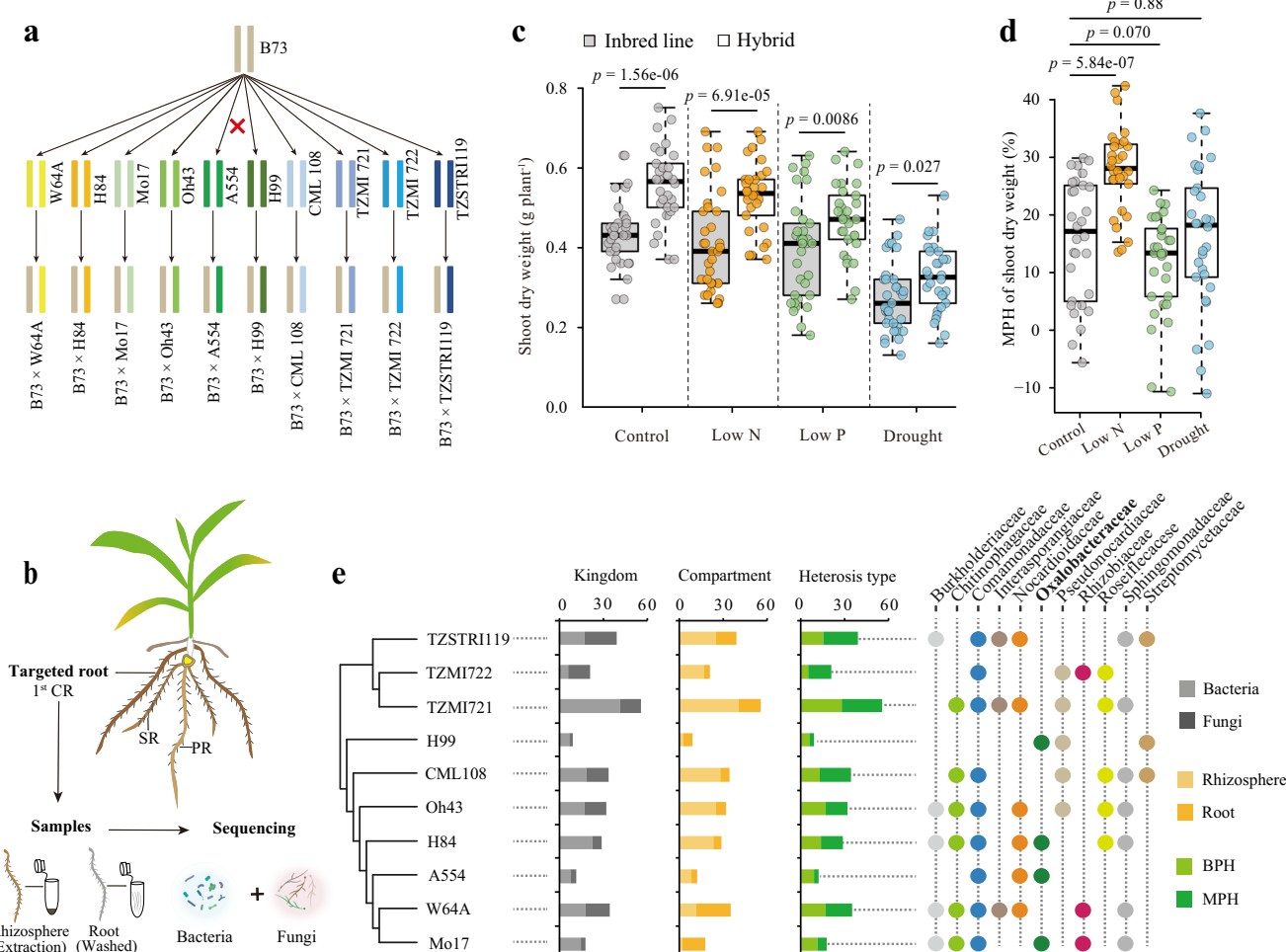

**Fig. 1 | Hybrid performance varies with soil microbiome diversity and genotype combinations under nitrogen deficiency. a** Maize inbred-hybrid triplets used in this study. The experimental design involved crosses of 10 maize inbred lines with the recurrent mother B73 to generate a diverse set of hybrids (10 hybrids). **b** Illustration of maize root types and microbiome sampling. CR crown root, PR primary root, SR seminal root. **c** The growth performance of inbred lines and hybrids under diverse abiotic stresses. The growth performance under each stress (inbred lines, $n = 33$; hybrids, $n = 30$) was measured for a variety of abiotic stress conditions, including low nitrogen. Statistically significant differences in growth performance between inbred lines and hybrids are indicated by the exact $p$ values (two-tailed Student's $t$ tests) for each stress. **d**. Effect of abiotic stress on heterosis manifestation during maize growth. A total of $n = 30$ biologically independent data points were collected (10 crossing triplets × 3 independent biological replicates). Significant differences between inbred lines and hybrids for mid-parent heterosis (MPH) are indicated by exact $p$ values (linear mixed-effects model (LMM), two-sided, with hybrid genotype included as a random effect to appropriately account

for the nested structure of the data). **c**, **d** Boxes span from the first to the third quartiles, center lines represent the median values and whiskers show data lying within 1.5× the interquartile range of the lower and upper quartiles. Data points at the ends of whiskers represent outliers. **e** Heterosis of microbial ASVs across 10 maize crossing triplets with the common maternal genotype B73 under nitrogen deficiency. The dendrogram shows phylogenetic relationships among parental maize inbred lines. Adjacent stacked bar plots summarize properties of microbial ASVs reflected by diverse heterotic patterns, where the varying bar lengths represent the number of ASVs detected in each hybrid triplet. Classified from left to right: number of bacterial (light gray) or fungal (dark gray) ASVs exhibiting heterosis, and number of ASVs showing higher relative abundance in either rhizosphere (light orange) or root compartment (dark orange), and number of ASVs significantly fitted with midparent heterosis (MPH, dark green) or better parent heterosis (BPH, light green). The rightmost panel highlights the most frequently enriched bacterial families across all hybrids that display microbiome heterosis. Each dot represents a microbial taxon showing heterosis in each crossing triplet.

---

enhanced lateral root development might be associated with biomass heterosis, although causality cannot be inferred from this observation alone.

To test the robustness of this association, we extended our experiments to a set of genetically diverse hybrids (B73 × R109B, B73 × Ky228) differing substantially in rhizosheath size (Supplementary Fig. 1a). Across these genetic backgrounds and soil types, rhizosheath heterosis varied, and soil microbiomes played a significant role in shaping this variation (Supplementary Fig. 2b, c). We then quantified whole-root system traits including lateral root density, root hair length, total root length, mean root diameter, root surface area and volume (Supplementary Data 4). Under nitrogen-poor conditions, most root traits improved in sterile soil, suggesting that the native soil

microbiome may suppress root development. Strikingly, only lateral root density was consistently promoted by *Massilia* inoculation (Fig. 2f and Supplementary Fig. 3, Supplementary Data 4), indicating a specific role of this taxon in lateral root branching. No consistent effects were observed for other root traits. These results suggest that *Massilia* might preferentially modulate lateral root development and root architecture under nitrogen-poor conditions[17,26], although its ecological role and colonization dynamics remain to be resolved.

Finally, to test whether single *Massilia* inoculation impacts heterosis in a broader set of root traits and depending on nitrogen acquisition, we quantified multiple architectural and nutritional traits across genotypes in nitrogen-poor soil (Supplementary Data 4). Although most root traits were significantly associated with plant

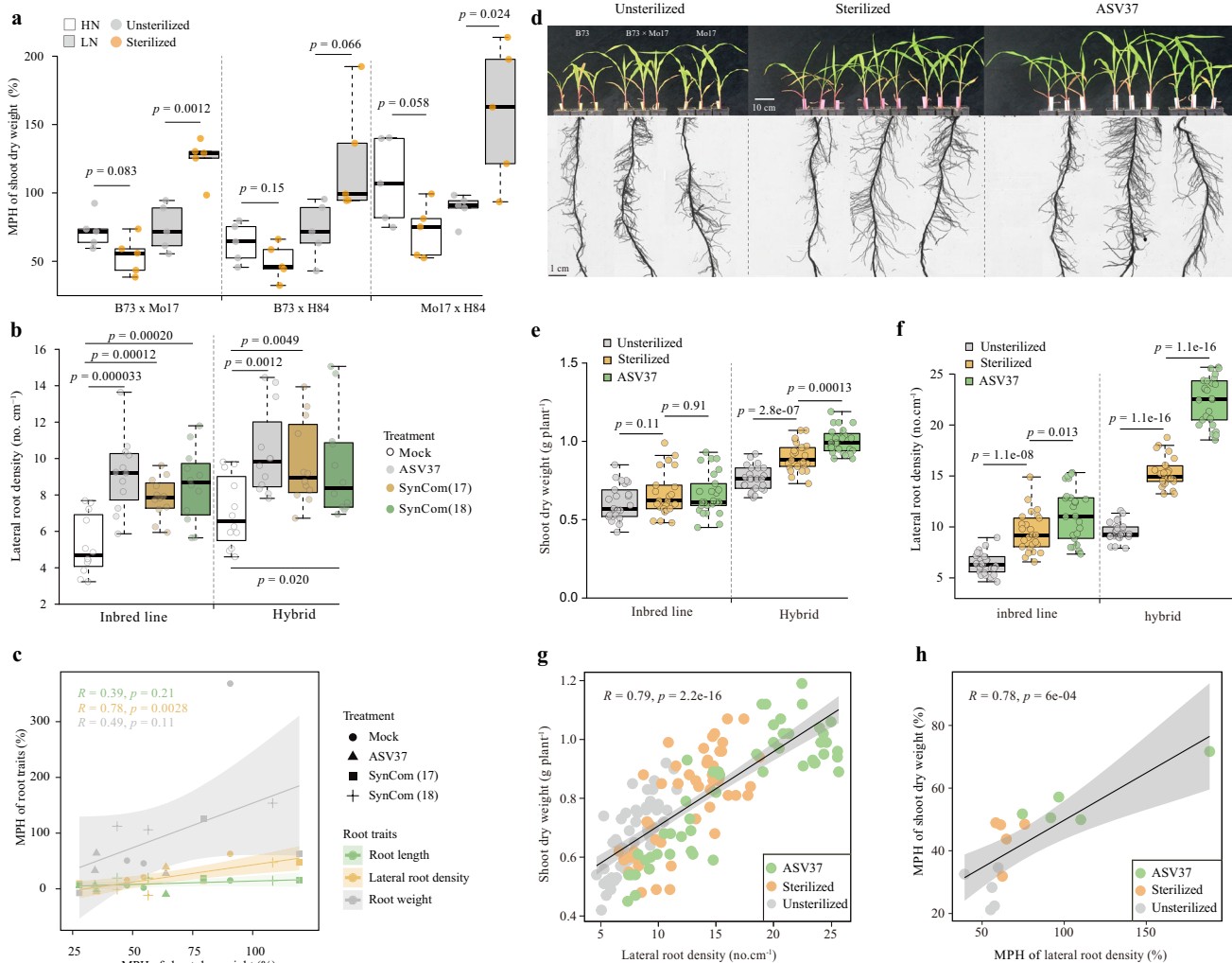

**Fig. 2 | *Massilia* is associated with enhanced lateral root formation, and its potential influence on maize heterosis in nitrogen-poor soil. a** Midparent heterosis (MPH) of shoot dry weight with and without soil microbiome in soil under two nitrogen conditions in soil. Significant differences among groups are indicated by exact *p* values (linear mixed-effects model (LMM), two-sided, with hybrid genotype included as a random effect); *n* = 5 biologically independent samples. MPH midparent heterosis, LN low nitrogen soil, HN high nitrogen soil. **b** The effect of different synthetic microbial communities (SynCom) on lateral root formation of inbred lines and hybrids. Three inbred/hybrid triplets using different SynComs (*Massilia* ASV37 alone, a 17-member synthetic bacterial community (SynCom17) of *Massilia* isolates that did not include ASV37 and an 18-member *Massilia* (Syn-Com18) including SynCom17 and ASV37). Significant differences at *p* < 0.05 among groups are indicated by the exact *p* values (two-tailed Student's *t* tests, each inoculation treatment compared to non-inoculation treatment (Mock)) within each genetic group (inbred inline and hybrid); *n* = 12 biologically independent samples. **a**, **b** Boxes span from the first to the third quartiles, center lines represent the median values and whiskers show data lying within 1.5× the interquartile range of the lower and upper quartiles. Data points at the ends of whiskers represent outliers. **c** Correlation between MPH of different root traits and MPH of shoot dry weight after inoculation with independent soil-derived *Oxalobacteraceae* isolates in nitrogen-poor soil. Scatter plot with best-fit regression line (black solid) and 95% confidence interval (shading, two-sided Pearson correlation). The regression line represents the measure of center, and the gray shading indicates the error band (95% confidence interval of the regression fit). Different symbol shapes represent different SynComs (*Massilia* ASV37 alone, SynCom17, and SynCom18). Different

colors for confidence intervals and lines represent correlation of different root traits (root weight, root length, and lateral root density). B73, Mo17, H84, B73 × Mo17, B73 × H84 and Mo17 × H84 were studied in (**a**–**c**) at single root level. **d** Growth performance of shoot and first whorl of crown roots for B73, B73 × Mo17 and Mo17 in the presence or absence of the soil microbiome and *Massilia* inoculation treatments in nitrogen-poor soil. The images show the first whorl of crown roots and overall shoot performance. B73 on the left, B73 × Mo17 in the middle, Mo17 on the right for each treatment. Shoot dry weight (**e**) and lateral root density (**f**) for maize inbred lines and hybrids in the presence or absence of the soil microbiome and *Massilia* ASV37 inoculation. **e**, **f** *n* = 25 biologically independent samples. Significant differences at *p* < 0.05 among treatments (Unsterilized vs Sterilized, Sterilized vs ASV37) are indicated by the exact *p* values (two-tailed Student's *t* tests). Boxes span from the first to the third quartile, center lines represent median values and whiskers show data within the 1.5× interquartile range of lower and upper quartiles. Data points at the ends of whiskers represent outliers. **g** Correlation between lateral root density and shoot dry weight. The regression line (black solid) and confidence intervals (gray shading) show the degree of correlation. **h** Correlation between midparent heterosis (MPH) of lateral root density and MPH of shoot dry weight. **g**, **h** Scatter plot with best-fit regression line (black solid) and 95% confidence interval (gray shading, two-sided Pearson correlation). The regression line represents the measure of center, and the gray shading indicates the error band (95% confidence interval of the regression fit). **e**–**h** B73, Mo17, H84, R109B, Ky228, B73 × Mo17, B73 × H84, B73 × R109B, B73 × Ky228 and Mo17 × H84 were selected for root trait experiments at whole root system level in nitrogen-poor soil.

growth and nitrogen uptake (Fig. 2g and Supplementary Fig. 4, Supplementary Data 4 and Data 6), only lateral root density correlated with heterosis manifestation for shoot biomass in nitrogen-poor soil (Pearson's $R = 0.78$, $p = 6e-4$, Fig. 2h and Supplementary Fig. 5, Supplementary Data 5 and Data 7). This reinforces lateral root density as a key trait linked to *Massilia* and heterotic growth under nitrogen stress. In addition, soil sterilization and inoculation experiments revealed distinct responses between hybrids and inbred lines. Specifically, *Massilia* inoculation enhanced maize growth stronger in hybrids (Fig. 2e), suggesting that inbred lines and their hybrid progeny differ in their responsiveness to beneficial microbes, potentially reflecting variation in resource acquisition or root plasticity.

### Root metabolome shifts in response to *Massilia* inoculation reveal pathways supporting maize heterosis in nitrogen-poor soil

Previous work identified a potential link between lateral root development and the keystone microbial taxon *Massilia*, mediated in part by root exudates[17,26]. To further explore this connection, we analyzed root exudates from maize inbred lines and hybrids grown under nitrogen deprivation to identify metabolites that differ between genetic backgrounds, and to test whether *Massilia* influences root metabolic profiles. We conducted untargeted metabolome analysis on root exudates collected from all heterotic combinations under three soil treatments: unsterilized soil, sterilized soil, and sterilized soil inoculated with a single *Massilia* isolate. Across samples, 11,011 metabolic features (coefficient of variation (CV) ≤ 30%) were detected, of which 3224 were annotated (Supplementary Data 8). Annotated metabolites were assigned to KEGG pathways, with the proportion of annotated metabolites falling into biosynthesis of secondary metabolites (18.5%), amino acid metabolism (16.7%), xenobiotics biodegradation and metabolism (11.4%) and carbohydrate metabolism (9.5%) (Supplementary Data 9). Partial least squares discriminant analysis (PLS-DA) revealed distinct profiles between inbred lines and hybrids regardless of soil treatment (Supplementary Fig. 6a). Removal of the native microbiome and reintroduction of the *Massilia* isolate each further altered the overall composition and relative abundance patterns of detected metabolites (Supplementary Fig. 6b). Univariate analysis of differential metabolites between inbred lines and hybrids, based on screening criteria of Variable Importance in the Projection (VIP) of the Orthogonal Partial Least Squares Discriminant Analysis (OPLS-DA) model ≥1, Fold Change ≥1.2 or ≤0.8 and *p*-value < 0.05, identified increasing numbers of significant metabolites across treatments: 599 in unsterilized soil, 729 after sterilization, and 904 after *Massilia* inoculation (Supplementary Fig. 7). These results suggest that *Massilia* alters root exudate composition under nitrogen deprivation, contributing to increased metabolic divergence between inbreds and hybrids.

Among differentially abundant metabolites, apigenin −a flavonoid known to recruit *Massilia* and promote lateral root development under nitrogen stress[17,26]− was significantly upregulated following inoculation (Fig. 3a, Supplementary Fig. 8). Pathway enrichment analysis of differential metabolites (KEGG, *p* < 0.05) further demonstrated that soil sterilization led to an increased representation of TCA cycle-related metabolites (Supplementary Fig. 9), consistent with general shifts in central metabolism under altered microbial conditions[27]. While TCA intermediates (e.g. citrate, oxaloglutarate) are typically intracellular, their presence in root exudates under nitrogen stress might reflect secretion for nutrient mobilization, passive leakage, or cell turnover. Thus, metabolomic profiles capture rhizosphere-level metabolic changes rather than direct intracellular activity. Interestingly, *Massilia* inoculation specifically led to enrichment of ABC transporter-associated and 2-Oxocarboxylic acid metabolism pathways in hybrids compared with inbred lines (Fig. 3b and Supplementary Fig. 9, Supplementary Data 8). The enrichment of ABC transporter

pathways is inferred from KEGG mapping of exuded metabolites[28] and does not reflect transporter gene expression or protein abundance. Similarly, the 2-Oxocarboxylic acid metabolism pathway links the TCA cycle to flavonoid biosynthesis[29] (Fig. 3c), suggesting a connection between carbon metabolism and secondary metabolism in the hybrid rhizosphere. Key metabolites such as oxaloglutarate (Metabolite ID: C05533) (Supplementary Data 8) were detected in exudates, where they correlated with both flavonoid biosynthesis[29] and lateral root formation in nitrogen-poor soil (Supplementary Data 4 and 6). These associations suggest that *Massilia* is linked to flavonoid-related metabolic signatures under nitrogen stress, that accompany heterotic growth under nutrient stress.

To further test the role of flavonoids, we analyzed maize mutants deficient in flavonoid biosynthesis. The *Colorless2-Inhibitor diffuse* (*C2-Idf*) mutant disrupts chalcone synthase, a key enzyme in early flavonoid biosynthesis[30], while the *flavone synthase type1* gene (*fnsi1*) disrupts flavone-specific biosynthesis. Both mutants and their respective wild-type backgrounds were grown under nitrogen-poor conditions with or without *Massilia* inoculation. Notably, *Massilia* inoculation significantly improved shoot biomass in both mutants (Two-tailed Student's *t* tests, *C2-Idf*, *p* = 0.033; *fnsi1*, *p* = 0.000058, Fig. 3d), demonstrating that growth promotion can occur even in the absence of endogenous flavone production. This finding suggests that the growth promoting effects of *Massilia* are at least partially independent of host-derived flavonoids and might involve alternative microbial–plant metabolic interactions. It also supports the idea of a reciprocal interaction between flavonoid exudation and *Massilia* colonization, although the causal direction remains unresolved.

### Rhizosheath formation as a trait of maize heterosis

We aimed to further explore the relationship between rhizosheath formation and plant heterosis. In our study, hybrids developed significantly larger rhizosheath than their parental inbred lines at both 5 days and 4 weeks after sowing (Fig. 4a, b). This consistent difference suggests that rhizosheath size might be a promising trait associated with hybrid genotypes, and a valuable phenotype for investigating belowground associations with heterosis. We therefore hypothesized that the correlative effect of *Massilia* on the manifestation of heterosis for shoot development is, at least in part, facilitated by rhizosheath formation in response to nitrogen deprivation. To examine this relationship, we measured rhizosheath size and fresh root weight in seedlings and 4-week-old plants grown in nitrogen-poor soil. Rhizosheath size correlated positively with root weight (Pearson's $R = 0.49$, $p = 1.9e−5$ for primary root weight for 5-day seedings; Pearson's $R = 0.53$, $p = 8.9e−5$ for total root weight for 4-week seedings; Supplementary Fig. 10), suggesting that rhizosheath formation reflects root growth and development to some extent, particularly under stress, where root surface traits and exudates may play a major role. Soil sterilization and *Massilia* inoculation experiments further demonstrated significant associations between rhizosheath formation, plant growth and nitrogen uptake (Fig. 4c, Supplementary Fig. 11, and Supplementary Data 4 and Data 6). Notably, rhizosheath size correlated only with heterosis for biomass (Pearson's $R = 0.61$, $p = 0.0015$), but not with heterosis of shoot nitrogen concentration or content (Fig. 4d, Supplementary Fig. 12, and Supplementary Data 5 and Data 7).

To identify root traits associated with rhizosheath formation, we analyzed phenotypic correlations in nitrogen-poor soil. Most traits, except root surface area, had a significant association with rhizosheath formation (Fig. 4c, Supplementary Fig. 13, and Supplementary Data 4 and Data 6). Among them, root hair length (Pearson's $R = 0.78$, $p = 0.00051$) and lateral root density (Pearson's $R = 0.70$, $p = 0.0035$) showed the strongest associations with heterosis for rhizosheath size. These two traits, which are critical components of root system architecture responsible for water and nutrient uptake, appear to play key roles in driving rhizosheath-associated hybrid vigor (Fig. 4d,

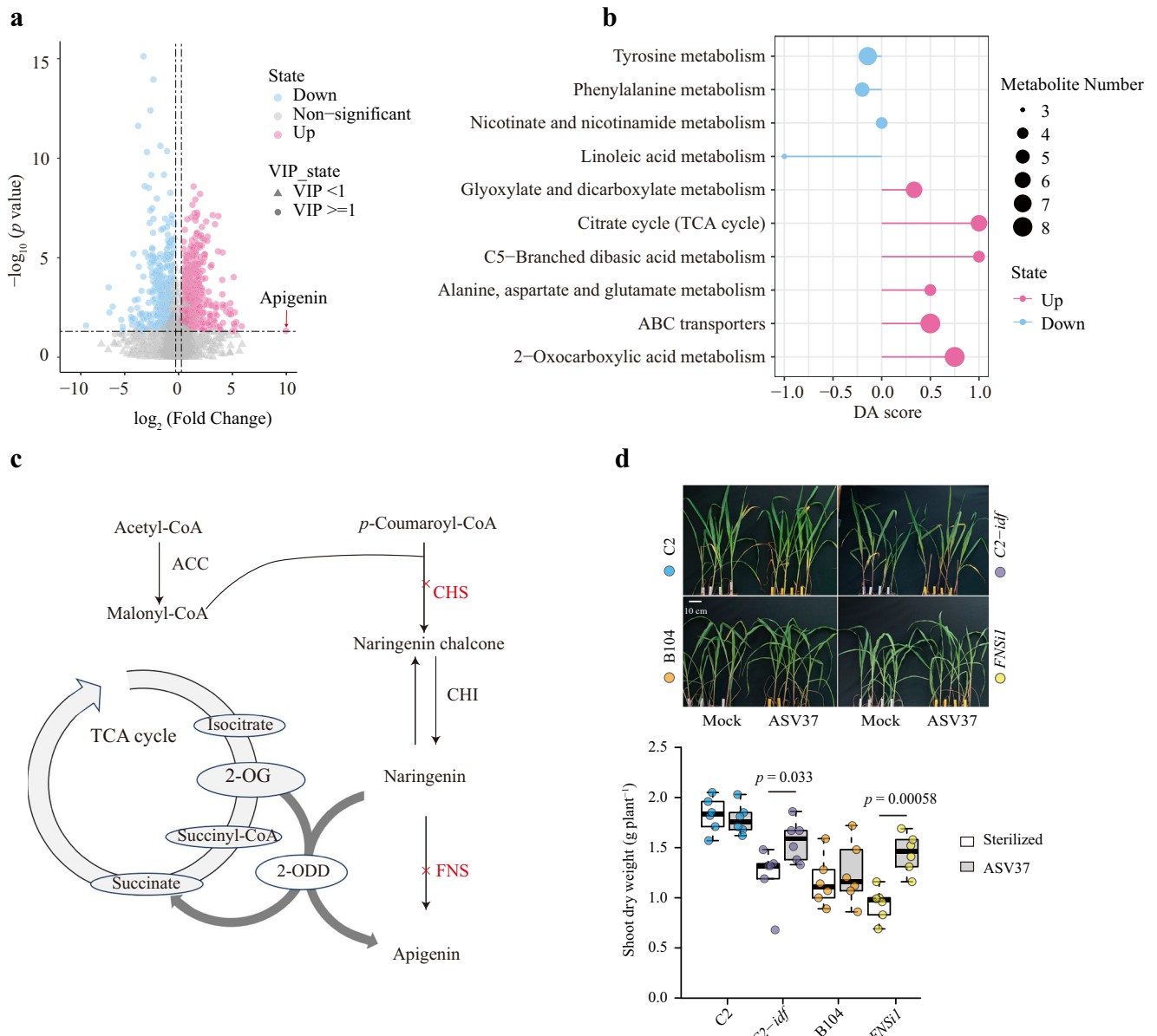

**Fig. 3 | Potential link between *Massilia*, flavonoid metabolism, and heterosis in maize under nitrogen stress. a** Volcano plot of differential metabolites after *Massilia* inoculation in nitrogen-poor soil. Significance was determined using two-sided statistical tests, with adjusted *P* values shown as $-\log_{10}(p\text{-value})$. X-axis represents the $\log_2$ fold change and the y-axis represents the $-\log_{10}$ *p*-value of each metabolite. Blue dots indicate significantly down-regulated metabolites, red dots represent significantly up-regulated metabolites and gray dots are metabolites with no significant difference. Circle-shaped markers denote metabolites with a Variable Importance in Projection (VIP) score ≥1, while triangle-shaped markers represent metabolites with a VIP score <1. **b** Metabolic pathway enrichment analysis abundance score plot after *Massilia* inoculation treatment in nitrogen-poor soil. This panel provides an overview of the metabolic pathways affected by *Massilia* inoculation, showing the differential abundance score (DA score) for each pathway. The DA score reflects the overall shift in metabolite abundance in a particular metabolic pathway. DA score of 1 indicates that all metabolites in the pathway are upregulated, while a score of −1 indicates downregulation. The size of the dots at the end of each line segment represents the number of metabolites in each pathway, with larger dots indicating more metabolites involved. **b, c** B73, Mo17, H84,

R109B, Ky228, B73 × Mo17, B73 × H84, B73 × R109B, B73 × Ky228, and Mo17 × H84 were selected for root metabolism experiment at whole root system in nitrogen-poor soil. *n* = 5 biologically replicates for each genotype. **c** A schematic overview of the flavone biosynthesis pathway. This panel presents a schematic diagram of the flavonoid biosynthesis pathway, focusing on the role of 2-oxoglutarate-dependent dioxygenase (2-ODD) in flavonoid biosynthesis. The diagram shows key enzymes involved in flavonoid synthesis, including chalcone synthase (CHS), chalcone iso-merase (CHI) and flavone synthase (FNS), with highlighted mutations in the CHS (*C2-Idf* mutant) and FNS (*FNSi1* mutant) genes. The concept of the model is adapted from refs. 26,29. **d** The effect of *Massilia* on shoot performance of flavone-deficient mutants (*C2-Idf* and *FNSi1*) and their corresponding wild types (C2 and B104) in nitrogen-poor soil. *n* = 6 biologically independent samples. Boxes span from the first to the third quartiles, center lines represent the median values and whiskers show data lying within 1.5× the interquartile range of the lower and upper quartiles. Data points at the ends of whiskers represent outliers. Statistically significant differences (*p* < 0.05) in growth performance for all wild types and mutants with and without soil microbiome indicated by the exact *p* values (two-tailed Student's *t* tests) for each stress.

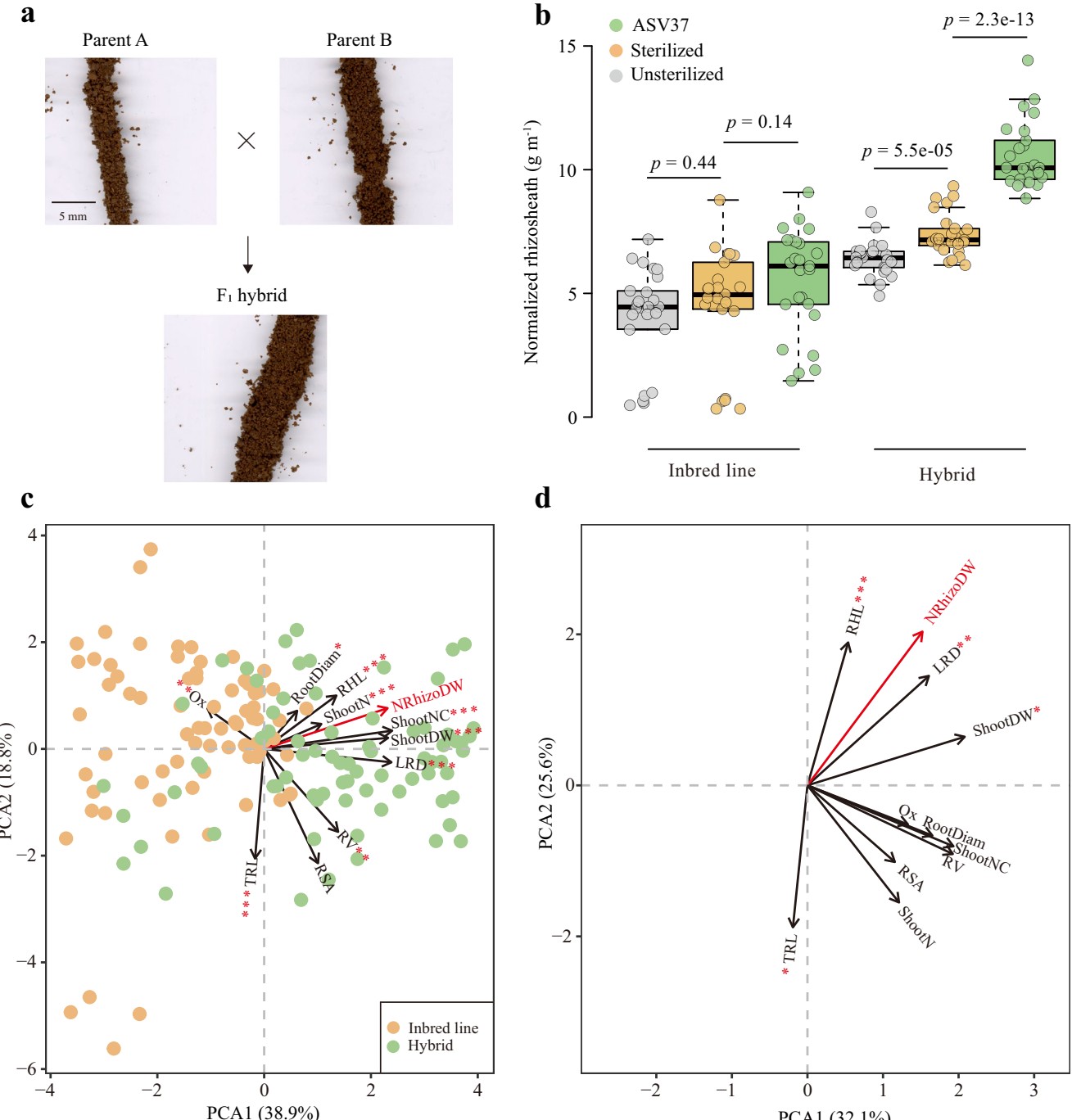

**Fig. 4 | Rhizosheath formation influenced by *Massilia* is potentially involved in maize heterosis in nitrogen-poor soil. a** Rhizosheath size of inbred lines and hybrids for 5-day-old maize seedlings in nitrogen-poor soil. **b** Rhizosheath size after different soil microbiome and *Massilia* ASV37 inoculation treatments in nitrogen-poor soil. *n* = 25 biologically independent samples. Significant differences at *p* < 0.05 among treatments (Unsterilized vs Sterilized, Sterilized vs ASV37) are indicated by the exact *p* values (two-tailed Student's *t* tests). Boxes span from the first to the third quartiles, center lines represent median values and whiskers show data lying within 1.5× interquartile range of lower and upper quartiles. Data points at the ends of whiskers represent outliers. **c** Principal component analysis (PCA) showing the correlation between rhizosheath size (red arrow) and other traits (black arrow) in nitrogen-poor soil. Significances are indicated for the correlation

between rhizosheath size and other shoot and root traits by asterisks (two-sided Pearson correlation, red asterisks). **d** PCA shows the correlation between MPH of rhizosheath size (red arrow) and MPH of other root and shoot traits (black arrow) in nitrogen-poor soil. Significances are indicated for the correlation between MPH of rhizosheath size and MPH of other shoot and root traits by asterisks (two-sided Pearson correlation, red asterisks). **c, d** \**p* ≤ 0.05; \*\**p* ≤ 0.01; \*\*\**p* ≤ 0.001. MPH mid-parent heterosis, RSA root system architecture, ShootDW shoot dry weight, ShootN shoot nitrogen concentration, ShootNC shoot nitrogen content, TRL total root length, RootDiam mean root diameter, RV root volume, RHL root hair length, LRD lateral root density, Ox oxoglutarate content, NRhizoDW normalized rhizosheath dry weight.

Supplementary Fig. 14, and Supplementary Data 5 and Data 7). In addition, the functional metabolite oxaloglutarate was tightly associated with rhizosheath formation (Fig. 4c), suggesting a potential link between flavonoid metabolism and *Massilia*-mediated promotion of rhizosheath development. Although the MPH of oxaloglutarate content did not show significant correlation with the MPH of rhizosheath size (Fig. 4d and Supplementary Data 5 and Data 7), it was significantly associated with lateral root development (Fig. 4c and Supplementary Data 4 and Data 6) − a trait known to facilitate rhizosheath formation. In summary, under nitrogen-poor conditions, rhizosheath formation is associated with maize heterosis. Rather than serving as a proven mechanism, rhizosheath size should be considered a composite indicator trait that integrates root surface features, microbial colonization (including *Massilia*), and metabolic processes. This integrative role makes rhizosheath a valuable target for future studies dissecting the genetic and microbial bases of heterosis.

### Rhizosheath linked microbiome-promoting activity influences maize growth and N₂O emissions in nitrogen-poor soil

Rhizosheath size is considered as a prominent feature helping plants to cope with abiotic stresses[13,14,31,32]. Plants with larger rhizosheaths generally exhibit stronger resilience, likely due to enhanced soil–root contacts[13]. In our study, hybrids consistently formed larger rhizosheaths than inbred lines prompting the hypothesis that this trait contributes to improved growth under nitrogen deprivation by shaping microbial recruitment. To test this idea, we performed a rhizosheath transplantation experiment using genotypes with contrasting rhizosheath sizes −Ky228 (small), B73 (medium) and R109B (large), along with their hybrids (B73 × Ky228 and B73 × R109B) (Supplementary Fig. 2). Seedlings were grown in soils previously conditioned by these donor genotypes, enabling assessment of how rhizosheath-associated microbial communities affected recipient growth in nitrogen-poor soil. Interestingly, plants grew best in soil conditioned by the large-rhizosheath genotype R109B (Fig. 5a). These findings suggest that plants with large rhizosheaths may especially foster beneficial microbiomes to improve plant growth[32].

Microbiome sequencing confirmed that genotypes significantly influenced bacterial community composition (PERMANOVA, $R^2 = 0.27$, $p = 0.0055$) (Supplementary Fig. 15b), though not overall Shannon diversity (Supplementary Fig. 15a). Interestingly, *Massilia* was preferentially enriched in genotypes with large rhizosheaths (Fig. 5c) and was positively correlated with rhizosheath size (Fig. 5e, Pearson's $R = 0.64$, $p = 0.0022$). This association suggests that rhizosheath development and *Massilia* enrichment may together support nutrient uptake under nitrogen limitation[17,26].

We next tested whether rhizosheath size and *Massilia* enrichment may influence soil nitrogen loss. Across genotypes, larger rhizosheath size was associated with reduced N₂O emissions (Supplementary Fig. 16). N₂O fluxes were negatively correlated with *Massilia* enrichment (Pearson's $R = -0.57$, $p = 0.0085$; Fig. 5f) and rhizosheath size (Pearson's $R = -0.59$, $p = 0.0065$; Fig. 5g). These findings suggest that increased rhizosheath development and *Massilia* abundance are associated with reduced soil nitrogen loss, although the causal mechanisms remain to be clarified.

To better understand the functional potential of rhizosheath-associated microbiomes, we determined the metagenomes from genotypes with contrasting rhizosheath sizes. Functional annotation using eggNOG revealed that rhizosphere conditions by small-rhizosheath genotypes were comparatively depleted in microbial pathways related to amino acid transport and metabolism, defense mechanisms, as well as energy production and conversion. In contrast, large rhizosheath genotypes enriched microbial functions linked to nutrient mobilization and resilience (Fig. 5d). Overall, these findings support the hypothesis that vigorous genotypes with large rhizosheaths are more effective in recruiting functionally beneficial microbiomes in the rhizosphere. This might in turn enhance nutrient acquisition, promote plant growth, and mitigate nitrogen losses, while smaller-rhizosheath genotypes appear less capable of establishing such favorable microbial associations.

## Discussion

Heterosis, or hybrid vigor, has been a cornerstone of modern crop breeding, improving both yield and stress tolerance[4,5]. However, the underlying mechanisms that cause heterosis, particularly under nutrient stress conditions such as nitrogen deprivation, are not yet fully understood. There is increasing evidence highlighting the importance of the rhizosphere microbiome in shaping plant growth and performance. Specific bacterial taxa play crucial roles in influencing root development[17,26,33], nutrient uptake[2], and stress tolerance[1]. The interaction between plant genetics and the microbiome is complex. Evidence suggests that hybrids host a distinct community composition than inbred lines[20,22,23] or that they might recruit a more beneficial microbiome[34–36]. This could be a direct consequence of changes in the root absorbing surface, root exudate profiles or the way in which hybrid plants interact with the soil environment[19].

### *Massilia* enrichment and lateral root development in nitrogen-poor soil

The bacterial genus *Massilia*, a keystone microbial taxon in the rhizosphere, is a promising candidate for enhancing maize growth under nitrogen-deficient conditions, particularly by influencing lateral root development[17,26]. This allows maize plants to explore a larger volume of soil for available nutrients. Interestingly, lateral root density exhibits a high degree of midparent heterosis compared to other root traits, indicating that lateral root development plays a significant role in maize growth vigor[37]. Our findings suggest that *Massilia* might be linked to maize heterosis via lateral root development under nitrogen deprivation (Fig. 2). *Massilia* promotes lateral root formation, *thus*enhancing the ability of maize plants to forage for nitrogen, increasing nitrogen uptake and biomass accumulation. This potentially reflects the heterotic performance of hybrids (Fig. 2b, f). However, the observed shifts in the rhizosphere microbiome, including the enrichment of *Massilia*, are more likely a consequence of hybrid-specific root architecture and exudation profiles, rather than direct drivers of heterosis. Importantly, our results do not imply that *Massilia* is a universal mechanism of heterosis but rather highlight a tractable model system for studying plant–microbe interactions shaped by hybrid vigor (Fig. 6). However, we recognize that functional redundancy within the rhizosphere microbiome is likely, and that *Massilia* is one of several microbial taxa that can potentially interact with hybrid roots to influence plant performance. Future studies integrating synthetic communities, targeted microbial knockouts, and host mutants will be essential to disentangle causality and functionally test whether specific microbial taxa−such as *Massilia*−can actively contribute to heterosis under defined environmental conditions.

### Rhizosheath formation as an indicator of heterosis

The rhizosheath represents a key interface between roots, soil, and microbes, and is composed of both physical soil particles and a complex microbial community[13,38]. This structure is crucial for the acquisition of nutrients and water, especially under abiotic stresses such as drought and nutrient deprivation[10,13,31,39]. Plants with larger rhizosheaths exhibit stronger growth vigor and greater resistance to environmental stresses[13,31,32,38], likely due to better soil aggregation, enhanced nutrient cycling and beneficial microbe recruitment. Interestingly, hybrid maize genotypes often exhibit a significantly larger rhizosheath than inbred lines, particularly under stress conditions (Fig. 4a and Supplementary Fig. 2). These findings suggest that rhizosheath formation might serve as a phenotypic indicator of heterosis, particularly in nutrient-poor soils where plant-microbial interactions

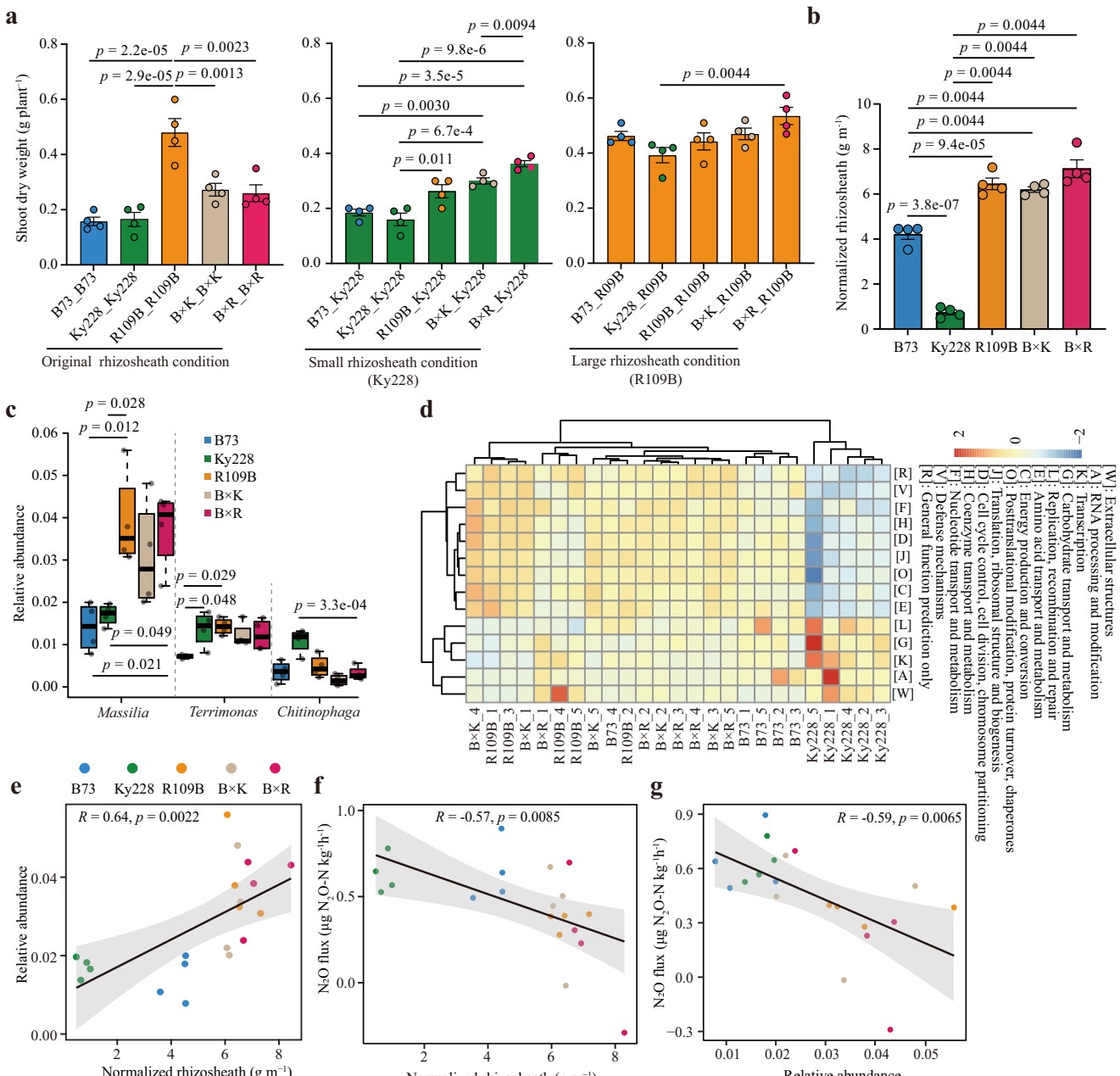

**Fig. 5 | Rhizosheath size-associated microbiome influences maize growth and N$_2$O emissions through functional feedback in nitrogen-poor soil. a** Shoot growth performance of different genotypes under different rhizosheath-size in nitrogen-poor soil. In the first panel, bars are colored differently to indicate three recipient genotypes and their F$_1$-hybrids. In the second and third panels, bar colors represent transplantation with rhizosheath from Ky228 (green) and R109B (orange), respectively, for consistent visual reference. Significant differences at $p < 0.05$ among different genotypes are indicated by the exact $p$ values (ANOVA, Tukey's HSD, two-sided). **b** Rhizosheath size of different genotypes in nitrogen-poor soil. Significant differences at $p < 0.05$ among different genotypes are indicated by the exact $p$ values (ANOVA, Bonferroni's HSD, two-sided). **a**, **b** $n = 4$ biologically independent samples. Data are mean ± SEM. **c** Effects of maize genotypes on the relative abundance of bacterial genera in the rhizosphere. $n = 4$ biologically independent samples. Boxes span from the first to the third quartiles, center lines

represent the median values and whiskers show data lying within 1.5× the interquartile range of the lower and upper quartiles. Data points at the ends of whiskers represent outliers. Significant differences at $p < 0.05$ among different genotypes are indicated by the exact $p$ values (ANOVA, Tukey's HSD, two-sided). **d** Abundance heat map of functional genes significantly differing among groups, as determined by the Kruskal−Wallis rank-sum test. Functional genes with $p$ value < 0.05 in rank sum test are shown in the heat map. Correlation between rhizosheath size and relative abundance of *Massilia* (**e**) and N$_2$O flux (**f**) in nitrogen-poor soil. **g** Correlation between relative abundance of *Massilia* and N$_2$O flux in nitrogen-poor soil. **e**–**g** Scatter plot with best-fit regression line (black solid) and 95% confidence interval (gray shading, two-sided Pearson correlation). The regression line represents the measure of center, and the gray shading indicates the error band (95% confidence interval of the regression fit). $n = 4$ biologically independent samples for each genotype.

play a critical role in influencing plant performance, rather than acting as a mechanistic driver. Among the diverse microbial communities in the rhizosheath, *Massilia* emerges as a potentially influential genus associated with maize growth and heterosis (Fig. 4b). Notably, *Massilia* was found to be more abundant in maize genotypes with larger

rhizosheaths, and might contribute to soil aggregation and microbial colonization around the root surface. The microbial association might facilitate nutrient acquisition, particularly nitrogen, possibly by altering soil pH, organic matter composition and the availability of nitrogen compounds in the rhizosphere[38].

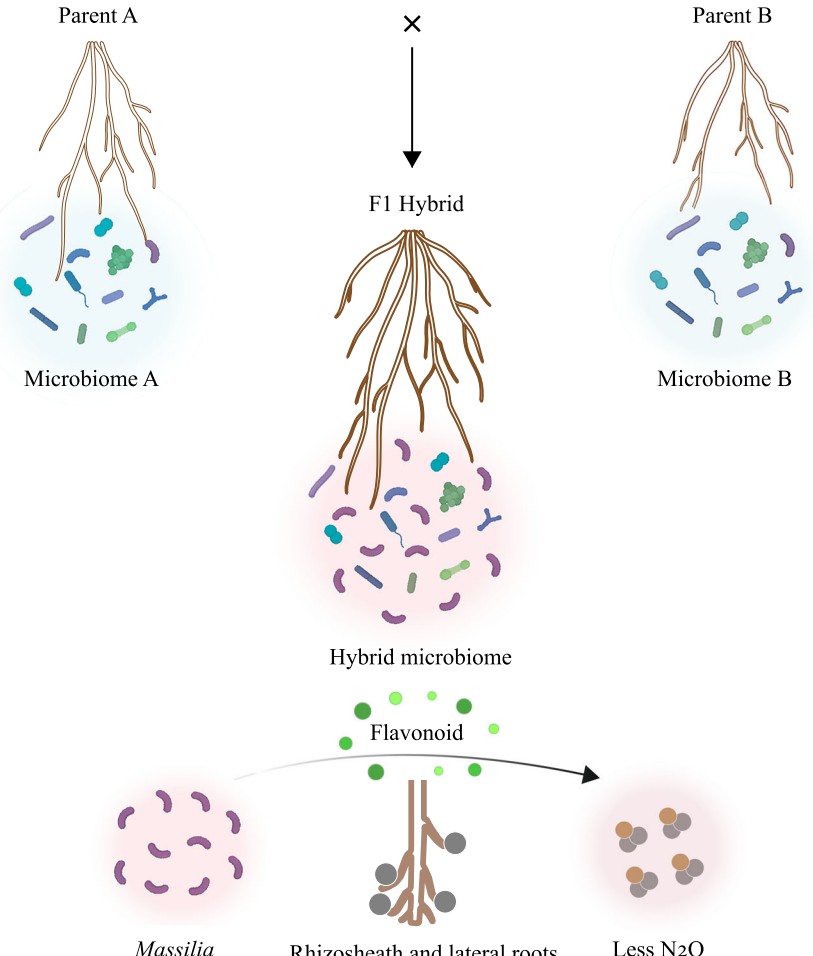

**Fig. 6 | Integrated model of root–microbiome–metabolite interactions associated with maize heterosis.** In nitrogen-poor soil, hybrids typically develop larger rhizosheaths that provide a hub for microbial colonization, including enrichment of *Massilia*. *Massilia* is associated with increased lateral root density, which in turn enhances root–soil contact and nitrogen uptake. Flavonoid secretion, such as apigenin, attract *Massilia* to the rhizosphere, while *Massilia* inoculation can in turn influence flavonoid profiles, suggesting a reciprocal feedback loop. Together, rhizosheath size, root traits, and microbial associations correlate with improved growth and reduced $N_2O$ emissions under stress. This model highlights rhizosheath formation as an integrative trait linking root development, exudation, and microbiome composition. Arrows indicate potential directions of interaction and feedback, but do not imply any established causality.

Interestingly, soil conditioned by large-rhizosheath genotypes improved the growth even of small-rhizosheath genotypes grown in this soil (Fig. 5a). This supports a model where rhizosheath-associated microbial communities create positive feedback that enhances growth of subsequent plants. Functional metagenomic analyses supported this by revealing enhanced microbial functions in terms of growth potential and resilience (Fig. 5d). Importantly, maize root traits tend to improve under nitrogen-poor conditions in the absence of native soil microbiome. Supplementary slurry experiments indicated that the growth-promoting effect of soil sterilization was more likely linked to the reduction of potentially harmful microbes than to changes in nutrient availability or soil texture (Supplementary Fig. 17). These results suggest that the bulk soil microbial community may have neutral or mildly inhibitory effects on root development. This underscores the importance of identifying specific microbial taxa, such as *Massilia*, that could mitigate broader negative interactions and promote root growth under suboptimal conditions.

### Complex role of flavonoids and reciprocal interactions with *Massilia*

The functional role of *Massilia* in rhizosheath-mediated heterosis goes beyond simply enhancing root growth and rhizosheath formation. Metagenomic and metabolomic analyses reveal that *Massilia* modulates several important pathways related to nitrogen cycling, flavonoid metabolism and stress tolerance (Figs. 3b and 5d). One such pathway involves the role of flavonoid exudation, particularly apigenin, which has been shown to regulate root development and modulate the rhizosphere microbiome[17,26]. In this study, we demonstrated that inoculation with *Massilia* significantly enhances the amount of apigenin exudated in the hybrid genotypes (Fig. 3a), which is consistent with the reported data of flavone-driven the enrichment of *Massilia* in the rhizosphere[17,26]. Interestingly, disruption of key flavonoid biosynthesis genes in maize enhanced the growth response to *Massilia* inoculation, suggesting that host flavonoids modulate plant–microbe interactions (Fig. 3d). Although it is known that flavonoids influence microbial recruitment, the specific role of flavonoid metabolism in shaping *Massilia* abundance remains unclear. It is possible that lateral root development alters the rhizosphere environment in ways that indirectly promote the enrichment of *Massilia* or potential interactions with other microbial species. Our findings suggest that *Massilia* enhances lateral root growth and supports rhizosheath-associated microbial communities. This might be achieved through interactions with flavonoid pathways in wild-type plants or by compensating for flavonoid deficiencies in mutants via alternative mechanisms, thereby improving nitrogen uptake and plant performance. Nevertheless, our current evidence does not resolve the causal

sequence or reciprocal feedback. Further time-series and mechanistic experiments are needed to determine whether flavonoids primarily drive *Massilia* enrichment, whether *Massilia* colonization induces flavonoid accumulation, or if both occurs.

### *Massilia* supports a role of mitigating $N_2O$ emissions in hybrids

Importantly, changes in microbial composition and metabolism mediated by *Massilia* are associated with reduced $N_2O$ emissions. $N_2O$ is a potent greenhouse gas, therefore these changes could potentially make nitrogen acquisition more efficient (Figs. 5 and 6). These results highlight the importance of the rhizosphere microbiome, specifically *Massilia*, in regulating maize growth and productivity in response to abiotic stress. Future research should elucidate the molecular mechanisms by which *Massilia* influences flavonoid metabolism and root development. It should also explore how *Massilia* can be integrated into crop breeding programs to improve stress resilience and sustainability. Indeed, modulating the root microbiome is a promising strategy for enhancing crop heterosis and improving nitrogen use efficiency in modern agriculture[24]. Further research should comprehensively evaluate nitrogen losses throughout the entire crop-growth period, including $N_2O$ emissions and nitrogen leaching, due to *Massilia*-mediated changes in the rhizosphere, particularly in the context of climate change where improving nutrient efficiency is a critical breeding target.

In summary, our results highlight *Massilia*, rhizosheath formation, and flavonoid metabolism as interlinked traits associated with heterosis in maize under nitrogen stress. We emphasize that these traits should be viewed as potential contributors and indicators of heterosis, rather than proven mechanisms. Our dataset provides a model system for dissecting how hybrid vigor shapes plant–microbe–soil interactions. Integrating synthetic communities, functional microbial mutants, and host trait manipulation in future studies will be crucial to resolving causality. The observed associations with $N_2O$ emissions, beyond plant growth, highlight the potential of harnessing microbiome–root trait interactions for sustainable agriculture. Modulating beneficial microbial associations could be a promising strategy to enhance crop performance and reduce nitrogen losses in the face of global climate challenges.

## Methods

### Soil collection and treatments

This study investigated two distinct types of agricultural soil to ensure robust conclusions regarding maize growth under various environmental conditions. The first soil type, referred to as Dikopshof soil, was collected from the 0–20 cm topsoil at the long-term experimental station Dikopshof, located near Cologne, Germany (50° 48′ 21′′ N, 6° 59′ 9′′ E). This station has been in operation for almost 120 years and the physicochemical properties of Dikopshof soil, including total nitrogen, organic carbon, and Olsen phosphorus, were previously characterized in detail as previously work[17]. and are not repeated here, as we used the same soil batch collected from the same long-term fertilization plots and depth (0–20 cm). Consistent collection and handling procedures ensure comparability with the published data. Three soil nutrient treatments were collected from Dikopshof station: (1) a complete fertilization treatment (control, receiving all nutrients); (2) a low nitrogen treatment (fertilized without nitrogen); and (3) a low phosphorus treatment (fertilized without phosphorus). Then we supplemented drought treatment using complete fertilization treatment.

Dikopshof soil was used to create four soil treatments:

1. Control (CK): Control soil with all nutrients supplied (33% water-holding capacity);
2. Low Nitrogen (Low N): Nutrient-deprived soil with nitrogen limitation (33% water-holding capacity);
3. Low Phosphate (Low P): Nutrient-deprived soil with phosphorus limitation (33% water-holding capacity);
4. Drought (D): Control soil with reduced moisture content (22% water-holding capacity);

All treatments maintained adequate moisture for optimal plant growth, with a water-holding capacity of 33% for nutrient treatments and 22% for drought treatment. In addition, the second type of agricultural soil, referred to as Campus soil, was collected from the 0–20 cm topsoil at Campus Klein-Altendorf, University of Bonn.

### Maize seed sterilization and germination

1. For all experimental procedures, maize seeds were surface sterilized to ensure uniformity and eliminate microbial contamination. The sterilization protocol was as follows: The seeds were washed with sterile water for ~5 min.
2. The seeds were then immersed in 70% (v/v) ethanol for 2 min to remove surface contaminants.
3. Afterward, the seeds were incubated in a bleach solution consisting of 29 ml sterile water, 15 ml NaClO (12–13% v/v stock solution) and 1 ml Tween 20 for surface sterilization.
4. Finally, the seeds were washed three times with sterile deionized water to remove any residual bleach.

After surface sterilization, the seeds were germinated for 3 days in a paper-roll system (Anchor Paper, Saint Paul, MN, USA; Hetz et al., 1996). Upon germination, the maize seedlings were transferred into small pots (7 cm × 7 cm × 20 cm) filled with a field soil mixture through a 4 mm mesh and grew in a climate-controlled phytochamber under a 16/8 h light/dark cycle at 26/18 °C (70% humidity) until harvest. Plants were watered with sterilized water daily, with the amount determined by weighing. Pots were fully randomized within phytochamber trays to minimize positional effects, and the tray position was rotated weekly.

### Sample size determination

No formal prospective statistical methods were used to predetermine sample sizes. Sample sizes were chosen based on common practice in plant biology and microbiome studies, taking into account pilot experiments conducted in our laboratory and published literature[17,26].

### Experimental design for investigating microbiome heterosis under different abiotic stresses

The objective of this study was to examine the potential hybrid vigor of maize and its effect on the microbiomes under various abiotic stresses, such as low nitrogen (Low N), low phosphorus (Low P) and drought (D). For this experiment, 10 diverse heterotic combinations of maize genotypes (Fig. 1a and Supplemental Dataset 1) were selected to investigate whether hybrid vigor is exhibited in the maize microbiome under abiotic stress conditions. The experimental design used the recurrent mother plant B73 and crossed it with ten maize inbred lines derived from different sub-populations, including stiff stalk, non-stiff stalk and tropical/sub-tropical types (Supplemental Dataset 1). These inbred lines were selected due to their large genetic variation, which ensures that the hybrids produced would represent a diverse range of genetic backgrounds. In total, ten hybrid combinations were generated, each consisting of a hybrid ($F_1$) derived from one of the B73 × inbred crosses. This provided a comprehensive array of hybrids for comparison with their parental inbred lines. The 3-day-old maize seedlings were then transplanted into small soil pots containing Dikopshof soil, which had been subjected to the four different soil treatments (CK, Low N, Low P, and D) described above. Each soil treatment was designed to mimic different environmental stressors that maize may encounter in field conditions. Root and rhizosphere samples were collected from the first whorl of the crown root of each maize plant after four weeks of growth. $n = 3$ biological replicates per genotype per treatment. Sample sizes were selected to adequately

represent the expected microbial diversity and ensure reproducibility across conditions.

## Root and rhizosphere sampling and microbiome sequencing

After four weeks of growth in the respective soil treatments, maize root and rhizosphere samples were collected from fully developed first whorl of shoot-borne crown roots of each maize plant[17] (Fig. 1b). Briefly, for rhizosphere sampling, the first whorl of crown roots was excised, gently shaken to remove large soil particles (leaving root-adhering soil defined as the rhizosphere), and placed into a 15 ml Falcon tube (Sarstedt). For root tissue sampling, the first whorl of separate crown root from the same plant was triple rinsed with top water followed by sterilized water. This root was then immediately dried, and placed into a 2 ml Falcon tube (Sarstedt). The root and rhizosphere samples were preserved in the −80 °C freezer for amplicon sequencing to profile the microbial communities in each of the treatments. The bacterial and fungal communities were analyzed by sequencing the 16S rRNA gene for bacteria and the ITS region for fungi, as previously described[17]. This allowed for a comprehensive profile of the microbial taxa present in the rhizosphere and roots of the maize plants.

## Experimental design for investigating the role of *Oxalobacteraceae* in enhancing heterosis in maize growth under low nitrogen stress

To investigate the potential role of *Oxalobacteraceae* in association with hybrid vigor in response to varying nitrogen levels, three hybrids (B73 × Mo17, B73 × H84 and Mo17 × H84) and their parental inbreds (B73, Mo17, and H84) were selected based on our initial screening, showed both strong biomass heterosis and consistent enrichment of *Oxalobacteraceae* in the rhizosphere under nitrogen stress. These hybrids were grown in soils with either high nitrogen (HN) or low nitrogen (LN) content, with and without the presence of the soil microbiome.

To prepare the low nitrogen soil (LN), natural field soil (Campus soil) was mixed with sterilized washed river sand in a 1:1 (w/w) ratio to create a nutrient-diluted base matrix. For high nitrogen soil (HN), a nitrogen dose of 50 mg kg$^{-1}$ NH$_4$NO$_3$ was added to the same nutrient-diluted soil mixture to simulate nitrogen supplementation. Additionally, such mixed soil matrix was sterilized through two cycles of autoclaving and heat incubation (121 °C for 20 min each) to eliminate microbial members, resulting in sterilized soil. Three-day-old maize seedlings were then transferred into small pots filled with high and low nitrogen soil matrix with and without soil microbiome. $n = 4$ biological replicates per genotype per treatment. This sample size was considered sufficient to detect biologically relevant differences. After six weeks, shoot dry weight was measured to assess the impact of nitrogen levels and soil microbiome on the manifestation of heterosis. Furthermore, a root inoculation experiment was conducted using three distinct synthetic microbial communities (SynComs):

1. SynCom18: a community containing all *Oxalobacteraceae* isolates (18 strains).
2. SynCom17: a community of *Oxalobacteraceae* isolates excluding *Massilia* ASV37 (17 strains).
3. *Massilia*-only SynCom: a community containing only *Massilia* ASV37.

All isolates used in this experiment were derived from a previously published study[17], with some additionally obtained from in-house maize root and rhizosphere isolations—and have provided a complete list in Supplemental Dataset 3. These inoculations were carried out on the three inbred lines and their F$_1$-hybrids (B73 × Mo17, B73 × H84 and Mo17 × H84) in low nitrogen soil to investigate whether *Oxalobacteraceae* influences root development and hybrid vigor manifestation. Detailed methods for bacterial isolation and inoculation are described in our previous work[17]. To further investigate the potential role of *Massilia* ASV37 in influencing the manifestation of heterosis under low nitrogen stress, we conducted additional experiments involving soil sterilization and root inoculation with *Massilia* ASV37 using low nitrogen Dikopshof soil. Additionally, we included two new hybrids (B73 × R019B and B73 × Ky228) which exhibit significant variation in rhizosheath formation (see Supplementary Fig. 2). In total, five hybrids (B73 × Mo17, B73 × H84, Mo17 × H84, B73 × R019B and B73 × Ky228) and their parental inbreds (B73, Mo17, H84, R019B, and Ky228) were selected to grow in low nitrogen soil for subsequent root phenotypic assessments and root metabiotic experiments.

## Microbiome sequencing for root and rhizosphere samples

These root and rhizosphere samples were collected for bacterial and fungal community analysis. DNA extraction from the root and rhizosphere samples was performed immediately after collection, following the PowerSoil DNA Isolation Kit protocol (MoBio Laboratories). For amplicon library preparation and sequencing, both bacterial 16S rRNA gene and fungal ITS gene sequencing was performed, as previously detailed[17]. Briefly, for bacterial community profiling, the V4 region of the 16S rRNA gene was amplified using the universal primers F515 (5′ GTGCCAGCMGCCGCGGTAA 3′) and R806 (5′ GGACTACHVGGGT WTCTAAT 3′). For fungal community analysis, the ITS1 region was amplified with the primers F (5′ CTTGGTCATTTAGAGGAAGTAA 3′) and R (5′ GCTGCGTTCTTCATCGATGC 3′). PCR reactions were carried out using the Phusion High-Fidelity PCR Master Mix (New England Biolabs) following the manufacturer's instructions. Only PCR products with the most prominent bands in the 400–450 bp range were selected for library preparation. Equal volumes of PCR products were mixed, and the pooled samples were purified using the Qiagen Gel Extraction Kit. Sequencing libraries were prepared using the NEBNext Ultra DNA Library Prep Kit for Illumina, incorporating sequence indices as recommended by the manufacturer. Library quality was checked using a Qubit 2.0 Fluorometer (Thermo Scientific) and an Agilent Bioanalyzer 2100 system. Finally, libraries were sequenced with 250 bp paired-end reads on an Illumina MiSeq platform.

## Data processing and microbiome analysis

All subsequent analyses were performed using R (v4.3.1), following a custom pipeline buildt upon QIIME2-processed ASV tables. Initial quality control (QC), trimming, and denoising were conducted in QIIME2 using DADA2, where forward and reverse reads were truncated at position 250 with default parameters. Bacterial taxonomy was assigned using the SILVA v138 database via the QIIME2 classify-sklearn plugin. In R, the amplicon sequence variant (ASV) tables were filtered to retain features with at least 10 reads in two or more samples, a threshold chosen to balance reproducibility with the ability to detect condition- or genotype-specific microbial taxa that may be biologically relevant under nutrient stress. To further reduce noise, ASVs with a relative abundance of ≤0.05% in ≤5% of samples were excluded. After filtering, samples with <1000 reads were also removed. This approach aligns with prior maize microbiome study conducted under controlled and abiotic stress conditions[17]. Microbial community dissimilarities were calculated using Bray-Curtis distances on variance-stabilized ASV counts using the 'varianceStabilizingTransformation' function from the DESeq2 (v1.34.0) package[40]. Constrained ordination analyses, such as principal coordinates analysis (PCoA), were performed using the 'capscale' function from the vegan package (v2.5-7) in R. Variance partitioning was carried out using the Bray-Curtis distance matrix to assess the effects of compartment, treatment and genotype on microbial community composition. This was followed by a permutation-based PERMANOVA test using the 'adonis' function in the vegan package.

## Analysis of heterosis in microbiome communities

To evaluate heterosis patterns in the microbiome, variance-stabilized counts of highly abundant (>0.05% relative abundance) and prevalent

ASVs present in at least 20% of all samples) were analyzed for 11 maize inbred lines and 10 hybrids (Supplementary Data 1), all crossed with the common mother line, B73, as previously described[22]. A linear mixed model was used to test for the effects of these ASV features using the lmer function from the lme4 (v1.1.27.1) package[41] in R. Blocks were treated as random effects, while treatments were set as fixed effects to account for any experimental noise. The residuals from this model were then used to examine heterosis patterns in the microbiome. The mean values of these residuals were calculated for the inbred lines and compared to the expected midparent heterosis (MPH) and better parent heterosis (BPH), assuming additive genetic variance. Two-sided $t$-tests were conducted to test the null hypothesis that each hybrid's microbiome trait value was equivalent to its respective MPH. BPH was tested using one-sided $t$-tests to determine whether the hybrid microbiome values fell outside the parental range. The significance of these tests was adjusted using the Benjamini–Hochberg method, with a threshold of adjusted $p$-values $< 0.05$.

### Heterosis calculation

To quantify the extent of microbiome heterosis, two different metrics were used:

1. Midparent heterosis (MPH): This metric compares the hybrid microbiome composition with the average microbiome composition of the parental inbred lines.
2. Better parent heterosis (BPH): This metric compares the hybrid microbiome composition to the more favorable parent (in terms of microbiome abundance) for each specific microbial taxon.

MPH and BPH values were calculated to describe the degree of abundance difference in the microbiome between the hybrids ($F_1$) and their parental inbred lines. This allowed us to assess whether hybrids exhibited more robust or distinct microbiome characteristics compared to their parental counterparts under abiotic stress conditions.

$$MPH = \frac{hybrid_{A \times B} - \frac{inbred_A + inbred_B}{2}}{\frac{inbred_A + inbred_B}{2}} \quad (1)$$

$$BPH = \frac{hybrid_{A \times B} - \max(inbred_A, \; inbred_B)}{\max(inbred_A, \; inbred_B)} \quad (2)$$

### Determination of rhizosheath size and shoot/root phenotypic traits

After 4-week growth in a climate-controlled phytochamber, shoot and phenotypic traits were collected from five hybrids (B73 × Mo17, B73 × H84, Mo17 × H84, B73 × R019B and B73 × Ky228) and their parental inbreds (B73, Mo17, H84, R019B, and Ky228) different soil microbiome conditions, including the presence or absence of soil microbiome and ASV37 *Massilia* inoculation treatments, in nitrogen-poor Dikopshof soil. For each genotype under each soil microbiome treatment, $n = 5$ biological replicates were used. This sample size was considered sufficient to detect biologically meaningful differences based on the observed effect sizes and variability. The method used for measuring rhizosheath size was adapted from ref. 42. Briefly, maize roots were carefully removed from the soil pot and large soil aggregates were discarded. The soil tightly adhering to the roots was defined as the rhizosheath soil. This rhizosheath soil was then washed three times with double-distilled water. Following washing, both the rhizosheath soil and the rinsing water were dried at 75 °C for 48 h to determine the dry weight of the soil. After the rhizosheath soil was washed, root traits were assessed. These traits included total root length, root hair length, lateral root density, mean root diameter, root volume and root surface area. To minimize damage to the roots, the entire root system was carefully excised and rinsed in water. Several root traits, such as total

root length, mean root diameter, root surface area and root volume, were scanned using an Epson Expression 12000XL scanner and root measurements were performed using Win-RHIZO software (Regent Instruments Inc., Quebec, Canada). Lateral root density was determined for the first whorl of crown roots by manually counting the number of emerged lateral roots per unit length (cm). Root hair length was quantified on second-order lateral roots from the first whorl of crown roots. For each plant, three lateral roots were randomly selected, and from each root, 10–15 root hairs were measured along the median portion of the root using a compact microscope (KL1500, Leica Microsystems, Germany). Root hair length was calculated as the average of all measurements per plant. The normalized rhizosheath size was calculated as the dry weight of the rhizosheath soil per unit of total root length[32]. For shoot traits, the shoot dry biomass and shoot nitrogen concentration were measured. In brief, shoot samples were harvested, dried at 70 °C for 48 h until they reached a constant weight and then weighed to obtain the dry weight of the shoot. Shoot nitrogen concentration was determined using an elemental analyzer (Euro-EA, HEKAtech).

### Root exudate collection procedure

To investigate whether *Massilia* influences root metabolism in maize seedlings, root exudates were collected from same 10 genotypes as rhizosheath size experiments (B73, Mo17, H84, R019B, Ky228, B73 × Mo17, B73 × H84, Mo17 × H84, B73 × R019B, and B73 × Ky228) from 4-week maize seedlings under different soil microbiome conditions, including the presence or absence of soil microbiome and ASV37 *Massilia* inoculation treatments, in nitrogen-poor Dikopshof soil. For each genotype under each soil microbiome treatment, $n = 5$ biological replicates were used. This sample size was considered sufficient to detect biologically meaningful differences based on the observed effect sizes and variability. The collection method was adapted from a previous publication[42]. First, the maize roots were carefully excavated from the soil, with large aggregates removed. The roots were washed under running tap water for 1 min, followed by three washes with deionized distilled water. Afterward, the whole root system was transferred to a 500 mL plastic beaker containing 200 mL of $CaSO_4$ solution (0.01 mmol L$^{-1}$) and submerged at 20 °C for 4 h. The solution was then filtered through filter paper (43–48 μm pore size; Filter-Lab®) and further filtered using a 0.22-μm membrane filter (Sartorius Stedim Biotech GmbH, Göttingen, Germany) to remove any particulate matter. The filtrate was flash-frozen using liquid nitrogen and subsequently lyophilized using a CHRIST ALPHA1-4/2-4 LSC basic freeze dryer (Germany) for metabolomic analysis.

### Mass spectrometry experiments

A total of 150 biological samples were analyzed in this study, comprising three treatment conditions: 50 samples under unsterilized condition, 50 sterilized samples under sterilized condition and 50 samples under sterilized plus *Massilia* (ASV37) inoculation condition. Each condition included $n = 50$ biological replicates. In addition, pooled quality control (QC) samples were prepared by mixing equal aliquots from all experimental samples and were injected at regular intervals throughout the analysis to monitor instrument stability.

### Metabolite extraction

Approximately 50 μg of each sample was extracted in 800 μL of precooled methanol/$H_2O$ (7:3, v/v) containing 20 μL of an internal standard mixture ($d_3$-leucine, $^{13}C_9$-phenylalanine, $d_5$-tryptophan, and $^{13}C_3$-progesterone). Samples were homogenized at 50 Hz for 10 min using a tissue grinder, followed by ultrasonication in a 4 °C water bath for 30 min. Extracts were incubated at −20 °C for 1 h and centrifuged at 14,000 rpm at 4 °C for 15 min. The resulting supernatant (600 μL) was filtered through a 0.22 μm membrane. and 20 μL of filtered solution from each sample composited the mixed QC sample to evaluate the

repeatability and stability of LC/MS analysis. Filtered samples and mixed QC samples were transferred to the 1.5 mL sample vials for instrument running.

## UPLC–MS analysis

Chromatographic separation was performed on a Waters 2777c UPLC system (Waters, USA) coupled to a Q Exactive HF high-resolution mass spectrometer (Thermo Fisher Scientific, USA). Separation was achieved on a Hypersil GOLD aQ Dim column (1.9 μm, 2.1 × 100 mm, Thermo Fisher Scientific, USA). The mobile phases consisted of 0.1% formic acid in water (A) and 0.1% formic acid in acetonitrile (B). The gradient program was: 0–2.0 min, 5% B; 2.0–22.0 min, 5–95% B; 22.0–27.0 min, 95% B; 27.1–30.0 min, 95% B (wash). Flow rate: 0.3 mL/min; injection volume: 5 μL; column temperature: 40 °C.

Mass spectrometry was performed in both positive and negative electrospray ionization modes. Full scan range was 125–1500 m/z (positive) and 100–1500 m/z (negative), resolution 120,000, automatic gain control (AGC) target 1e6, and maximum injection time 100 ms. The top three most intense ions were selected for MS/MS fragmentation at resolution 30,000, AGC target 2e5, maximum injection time 50 ms, and stepped normalized collision energies of 20, 40, and 60 eV. Source parameters were: sheath gas 40, auxiliary gas 10, spray voltage 3.80 (|KV|) (positive) or 3.20 (|KV|) (negative), capillary temperature 320 °C, auxiliary heater temperature 350 °C.

## Data processing and metabolite annotation

Raw data were processed using Compound Discoverer (v3.3, Thermo Fisher Scientific) with the following parameters: parent ion mass deviation <5 ppm, fragment ion mass deviation <10 ppm, and retention time tolerance <0.2 min. Metabolites were annotated by matching to the BGI metabolome database, mzCloud, and ChemSpider.

## Statistical analysis and differential metabolite identification

The processed peak intensity matrix was imported into metaX[43] for data preprocessing and statistical analysis. Data were normalized using probabilistic quotient normalization (PQN) and batch effects were corrected using QC-based robust LOESS signal correction (QC-RLSC). Features with a coefficient of variation >30% in QC samples were removed. Multivariate analysis was performed using partial least squares discriminant analysis (PLS-DA)[44] and orthogonal PLS-DA (OPLS-DA). Variable importance in projection (VIP) scores were used to assess the contribution of each metabolite. Differential metabolites were identified based on the combined criteria of VIP ≥ 1, fold change (FC) ≥ 1.2 or ≤ 0.83, and Student's $t$ test $p < 0.05$ (FDR-adjusted). Volcano plots were generated to visualize differential metabolites by plotting FC against statistical significance.

## Pathway analysis

Identified metabolites were mapped to the KEGG database for functional annotation and pathway enrichment analysis. Pathways with $p < 0.05$ were considered significantly enriched. The top 10 enriched pathways (ranked by $p$-value) were visualized using bubble plots to highlight differential metabolic changes among treatments.

## Functional verification of *Massilia* in association with flavone secretion

To investigate whether *Massilia* influences flavone secretion in maize, two flavone-deficient maize mutants, *C2-idf* and *FNSI1*, along with their corresponding wild-type strains, C2 and B104, were selected for this study. These mutants were chosen based on their inability to synthesize flavones, which are important secondary metabolites involved in plant-microbe interactions. To generate targeted mutations in the maize *Flavone Synthase Type I1* gene (*ZmFNSI1*, *ZmO0001d029744*), we employed CRISPR/Cas9 genome editing. A specific target site within the coding region was selected, and the following primers were used to amplify the genomic region flanking the target site: forward primer 630FW (5' CGCAGTCACCACCACCAGACATC3') and reverse primer 630RV (5' TCTTTCGACCACTACGCATTTG 3'). PCR amplification was performed using Thermo Phusion High-Fidelity DNA Polymerase with an annealing temperature of 64 °C and an elongation time of 30 s, resulting in an 832 bp product for the wild-type allele. Independent CRISPR-induced mutant alleles were identified: 193-18, Cas9 target sites as follows: CCTGTCGACGGCCGTGCACGACA; CCGCTCGCACGG CTTCTTCCAGG; TCGACGAGTTCCTGCCCGATTGG, both of which result in frameshift mutations predicted to disrupt *FNSI1*-mediated flavone biosynthesis.

In this experiment, low nitrogen soil was prepared by mixing sterilized washed river sand and natural field soil (Campus soil) in a 1:1 (w/w) ratio. Then we sterilized this soil by autoclaving and heat incubation. After seed surface sterilization, the seeds germinated in a paper-roll system for three days. We then performed root inoculation with ASV37 (*Massilia*) isolate. After 4 weeks of growth in a climate-controlled phytochamber, shoot dry weight was determined to assess the effect of *Massilia* on the growth of the two flavone-deficient maize mutants. $n = 6$ biological replicates per genotype per soil microbiome treatment. This sample size was considered sufficient to detect biologically meaningful differences based on the observed effect sizes and variability.

## Rhizosheath transplantation experiment

To investigate whether microbiota conditioned by rhizosheath size can functionally influence maize growth, we conducted rhizosheath transplantation experiments using two heterotic combinations (B73 × R109B and B73 × Ky228) and their parental inbreds (B73, R109B, and Ky228), both associated with distinct rhizosheath formations in nitrogen-poor soil. First, 3-day-old maize seedlings were transferred into small pots containing nitrogen-poor Dikopshof soil. After 4 weeks of growth in a climate-controlled phytochamber, the entire root system was carefully excavated, ensuring that most of the rhizosheath soil remained attached to the roots. The roots were then gently washed by hand to remove large aggregates. Next, the roots were placed in a water bath (TRANSSONIC T420, Elma) at room temperature (~20 °C) with high-frequency sound waves (~40 kHz) containing 25 ml of autoclaved deionized water for 5 min to remove the majority of the rhizosheath soil. The soil suspension containing the detached rhizosheath soil was returned to the original pots, preserving the microbial community associated with the roots. Finally, the newly germinated maize seedlings were transplanted into the pots, where they were either replanted in soil from the same genotype or from a different genotype with either a larger (R109B) or smaller (Ky228) rhizosheath (Supplementary Fig. 18). For each genotype, four biological replicates ($n = 4$) were measured independently. This sample size was considered sufficient to detect biologically meaningful differences based on the observed effect sizes and variability. After 4 weeks, shoot dry weight was harvested for all genotypes from different rhizosheath conditions to investigate the feedback regulation effect of plant growth in nitrogen-poor soil.

## Rhizosheath size dependent microbiome sample collection, sequencing and analysis

To investigate whether *Massilia* enrichment is associated with rhizosheath size, maize genotypes (B73, Ky228, R109B, B73 × Ky228 and B73 × R109B) exhibiting diverse rhizosheath sizes were selected. After 4 weeks of growth, rhizosphere samples from the first whorl of the crown root with developed lateral roots were harvested for 16S rRNA gene sequencing, as previously described[17]. For each genotype, four biological replicates ($n = 4$) were measured independently. Sample sizes were selected to adequately represent the expected microbial diversity and ensure reproducibility. All downstream analyses were performed using R (v4.3.1). ASV tables were filtered to include only

those with a relative abundance ≥0.05% in at least 5% of the samples. For α-diversity analysis, the Shannon index was calculated using ASV tables rarefied to 8000 reads. Any ASVs with a relative abundance ≤0.05% in ≤5% of the samples were filtered out. Bray–Curtis distances between samples were calculated based on ASV tables normalized using the variance Stabilizing Transformation function from the DESeq2 (v1.44.0) package[40]. To assess the effects of genotypes on microbial community composition, variance partitioning was performed using the Bray–Curtis distance matrix, followed by a permutation-based PERMANOVA test using the adonis function in the vegan package[45]. Principal Coordinate Analysis (PoCA) plots were generated using the ggplot2 package (v3.4.2). To identify differentially expressed genera between genotypes, ASVs were grouped by genus and unidentified genera were excluded. Only genera with >1% relative abundance in ≥2 samples were retained. Stacked charts were produced using ggplot2 to display the distribution of the top 10 genera. Genus distributions between different genotypes were compared using ANOVA with a post-hoc Tukey HSD test. Finally, a linear correlation was plotted between the midparent heterosis (MPH) of rhizosheath size and microbiome abundance using R (Pearson correlation).

## Metagenomic sample collection, sequencing and analysis

To investigate whether rhizosheath size functionally determines maize growth, we performed shotgun metagenomic sequencing to explore the functional differences associated with varying rhizosheath sizes. In this experiment, rhizosphere soil samples from the first whorl of crown roots of five genotypes (B73, R109B, Ky228, B73 × R109B and B73 × Ky228) were randomly selected from 4-week-old plants grown in a phytochamber. For each genotype, five biological replicates ($n = 4$) were measured independently. Sample sizes were selected to adequately represent the expected microbial diversity and ensure reproducibility. The integrity and purity of the isolated DNA were assessed using a NanoPhotometer spectrophotometer and 1% agarose gel electrophoresis. Subsequent steps followed the standard protocol provided by Illumina. DNA samples that passed quality assessment were fragmented. These DNA fragments underwent end repair, the addition of a 3′ A tail and adapter ligation. Properly sized products were selected, purified and then amplified to construct the sequencing library. Qualified libraries were then sequenced on an Illumina platform. Raw data quality control was performed to clean the reads for downstream bioinformatics analyses. To ensure reliable results, low-quality sequences were removed using the *fastp* software[46], generating high-quality clean tags. Host contamination was removed by aligning the clean tags to the reference genome of the host using *bowtie2*[47]. Metagenome assembly was performed using *MEGAHIT*[48], with the removal of contigs <300 bp. The assembly quality was assessed using *QUAST*[49]. For coding gene prediction, *MetaGeneMark*[50] (http://exon.gatech.edu/meta_gmhmmp.cgi, Version 3.26) was used to identify coding regions in the genome. Redundancy in the gene data was reduced using *MMseqs2*[51] (https://github.com/soedinglab/mmseqs2, Version 12-113e3), with a similarity threshold of 95% and a coverage threshold of 90%. For functional annotation, *eggNOG*[52] was used to classify genes into functional categories. Differential analysis of functional genes was performed using a Kruskal–Wallis rank-sum test (nonparametric test), with a corrected $p$-value threshold of 0.05. The results of the differential analysis were visualized using a heatmap.

## Soil slurry inoculation experiment in nitrogen-poor soil

To investigate whether soil sterilization promotes plant growth through autoclave-induced nutrient release or microbiome disruption, we conducted a slurry re-inoculation experiment using ten maize genotypes (B73, Mo17, H84, R019B, Ky228, and their hybrids). Each genotype was tested with four biological replicates ($n = 4$). This sample size was considered sufficient to detect biologically meaningful differences based on the observed effect sizes and variability. Prior to

sowing, a native microbial slurry was prepared by suspending fresh bulk soil in sterile water (1:2 w/v), shaking for 30 min at room temperature, and filtering through a 100-µm mesh to remove soil particles. The resulting suspension, containing native microbial communities with minimal nutrient content, was applied to autoclaved, nitrogen-poor soil at a rate of 5 mL per 100 g soil. To control potential nutrient effects from the slurry, a sterilized version of the same slurry (filtered through a 0.22-µm membrane) was also applied in parallel treatment. This design resulted in four microbiome soil conditions: (1) Unsterilized soil (Unsterilized), (2) Autoclaved soil without slurry (Sterilized), (3) Autoclaved soil with sterilized slurry (Sterilized_Sterilized Slurry), and (4) Autoclaved soil with unsterilized slurry (Sterilized_Unsterilized Slurry). After a 3-day germination period, seedlings were transferred to pots (7 cm × 7 cm × 20 cm) filled with a 1:1 (w/w) mixture of Campus soil and sterilized river sand. Plants were grown in a phytochamber under controlled conditions, and fresh shoot biomass was measured after 5 weeks to assess whether soil sterilization enhances plant growth via nutrient release or microbiome alteration.

## $N_2O$ measurement among genotypes with different rhizosheath size

To investigate whether rhizosheath size influences soil nitrogen loss, $N_2O$ emissions ($N_2O$ flux) were measured using a LI-7820 $N_2O$ Trace Gas Analyzer (LI-COR Environmental, Nebraska, U.S.A.). The LI7820 N2O/H2O analyzer is a high-precision, portable instrument designed for continuous and real-time measurement of nitrous oxide ($N_2O$) and water vapor ($H_2O$). To ensure the accuracy of the measurements, we conducted a two-point verification of the LI-7820 using standard gases of 0 ppb (Low) and 374 ppb (High) $N_2O$, covering the range of concentrations observed in our pot experiment (Supplementary Fig. 19). The results confirmed the instrument's accuracy. As our study focuses on relative differences in $N_2O$ fluxes among genotypes rather than absolute fluxes, potential systematic errors do not affect the comparison between genotypes. Based on this setup, five genotypes (B73, R019B, Ky228, B73 × R019B, and B73 × Ky228) with diverse rhizosheath sizes were selected to sow in nitrogen-poor soil (Campus soil mixed with sterilized washed river sand in a 1:1 (w/w) ratio). For each genotype, four biological replicates ($n = 4$) were measured independently. This sample size was considered sufficient to detect biologically meaningful differences based on the observed effect sizes and variability. In brief, at harvest (after 4 weeks of growth), a single maize plant at a time was carefully transferred into a sealed 12.5 L plastic bucket (material: PP; height: 25.5 cm; diameter: 29.3 cm) (OBI Paste Bucket 12.5 L, Germany) along with its intact root-soil system, including all adhering rhizosheath and bulk soil from the pot. This approach preserved that the natural soil structure and microbial community, minimizing disruption to gas exchange dynamics and enabling a physiologically relevant assessment of in situ $N_2O$ emissions. To improve measurement sensitivity and allow detectable changes in $N_2O$ concentration over short periods, the effective headspace volume was reduced to 10.1 L using $N_2O$-inert ballast. This reduction minimized headspace dilution and enhanced the system's responsiveness to low $N_2O$ fluxes. The bucket was sealed with a custom-fitted lid equipped with stainless steel inlet and outlet ports (Swagelok, Germany) connected to the gas analyzer via airtight tubing. Inside the chamber, a battery-powered fan (25 mm × 25 mm × 10 mm, 12 V) was placed to maintain homogenous mixing of the air during measurement, preventing stratification of $N_2O$ concentrations. During measurements, the lid was tightly secured with adhesive tape (Tesapack Ultra Strong 4124, Germany) to ensure an airtight seal and prevent leakage. $N_2O$ fluxes were measured during the same time window each day (9:00 am to 13:00 pm) to minimize circadian variation. For each sample, dry $N_2O$ concentrations were recorded over a 150-s period with a frequency of 1 Hz with an initial deadband of 30 s. Headspace temperature was monitored every 30 s using an electronic thermometer, and this value

was used to apply temperature correction in flux rate calculations. To avoid cross-contamination between measurements, the chamber was left open with the fan running for 60 s between samples. This allowed residual gases to dissipate and the chamber atmosphere to return to ambient conditions, ensuring an accurate baseline for the next sample.

The $N_2O$ flux rate (F) was calculated as following:

$$F = \frac{\Delta c}{\Delta t} \times \frac{28}{22.4} \times \frac{273.15}{273.15 + T} \times \frac{V}{M} \qquad (3)$$

where F is the $N_2O$ flux, $\frac{\Delta c}{\Delta t}$ is the change of $N_2O$ concentration with time in the chamber and calculated using a linear fit of $N_2O$ concentrations. T is the absolute temperature in the chamber during sampling, V is the measuring volume in the chamber, M is the mass of dry soil. This approach allowed us to reliably quantify $N_2O$ emissions under minimally disturbed, biologically relevant conditions across different rhizosheath phenotypes.

## Reporting summary
Further information on research design is available in the Nature Portfolio Reporting Summary linked to this article.

## Data availability
Raw bacterial 16S and fungal ITS sequencing data for root and rhizosphere samples under different stress conditions have been deposited in the Sequence Read Archive (http://www.ncbi.nlm.nih.gov/sra) under BioProject ID PRJNA889703. Raw bacterial 16S sequencing data for rhizosheath size experiment and raw soil metagenomics sequencing data have been deposited under BioProject ID PRJNA1236007. Non-targeted metabolomics data has been deposited in the MetaboLights under BioProject ID MTBLS12613. All statistical data are provided with this paper. Source data are provided with this paper.

## Code availability
The codes used in the heterosis microbiome analysis are publicly deposited at: https://doi.org/10.5281/zenodo.17192532.

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

## Acknowledgements

We thank F. Zhang and W. Zou (University of Bonn, Bonn, Germany) for helping with soil and plant sample harvest. We thank H. Hüging for providing soil from the long-term experimental station Dikopshof, and A. Glogau for technical support with the elemental analyzer. This work is supported by the Deutsche Forschungsgemeinschaft (DFG) Emmy Noether Programme 444755415 to P.Y., the DFG Priority Program (SPP2089) 'Rhizosphere Spatiotemporal Organisation - A Key to Rhizosphere Functions' grant 403671039 to F.H. and P.Y. and Germany's Excellence Strategy—EXC 2070–390732324 to A.M. and G.S. X.C.'s research is supported by The Changjiang Scholarship, Ministry of Education, China, State Cultivation Base of Eco-agriculture for Southwest Mountainous Land (Southwest University, Chongqing, China) and the National Maize Production System in China (grant no. CARS-02-15). This publication was supported by the open access publication fund of the University of Bonn.

## Author contributions

P.Y., X.C. and F.H. designed the research; P.Y supervised and coordinated the research; X.H. and L.G. performed all phytochamber experiments; X.H. and D.W. analyzed the microbiome data; X.H. and L.G. performed the shotgun metagenomic sequencing and non-targeted metabolic profiling; L.G performed soil slurry experiments; G.S. contributed through the plant nitrogen analysis; X.H. and M.B. performed the bacterial inoculation experiments. A.A. and A.M. performed the soil $N_2O$ measurement. X.H. and P.Y. wrote the manuscript. X.H., D.W., M.B., G.S., A.A., A.M., X.C., F.H. and P.Y. revised the paper. All authors reviewed the manuscript.

## Funding

## Competing interests

The authors declare no competing interests.
