## [Transparent Peer Review file · Nature Communications]

Rhizosphere inhabiting Massilia are linked to heterosis in roots of maize

Corresponding Author: Professor Peng Yu

Version 0:

Reviewer comments:

Reviewer #1

(Remarks to the Author)

He et al have performed a thorough and data-rich investigation of Massilia and its interactions with maize, especially with regard to the rhizosphere and heterotic combinations of the plant. My evaluation of this manuscript is largely positive, but with caveats. On the one hand, the experiments and data appear to be very thorough and well described, and I especially like how much of their original data has been made available as supplemental datasets. This is a practice that I wish were more common among authors. And, in general, the individual experiments (with one exception, below) appear to be carried out and interpreted well. However, there are some major shortcomings that I feel need to be addressed before publication, which I have outlined below:

Major issues:

1) Conclusions. The individual experiments and data are of high quality, but the conclusions are very much overblown. Throughout the manuscript the authors talk about Massilia “causing”, “contributing to”, or otherwise being a causal factor of heterosis. This overemphasizes the role of the microbe and makes it seem as if this is some key link in heterosis that has been long overlooked. Instead, all these results show a manifestation of heterosis with respect to the microbiome. Any improved interactions with Massilia, flavonoids, root growth, etc., are the outcomes of heterosis, not its cause. Please correct this throughout the manuscript. The authors have created a very nice model system to study heterosis with respect to the maize microbiome, but they need to be clear that is what it is: a model system. There is no evidence that Massilia is the microbe mediating similar responses in maize around the globe, and the most likely situation is that there are a great many microbes that can fill a similar niche. Some examples of things to correct:

a) Line 106 – There is no evidence of the microbiome being involved in heterosis here. Please delete that. All you’re showing is that nitrogen stress is important.

b) Line 108 – Change “significant differences” to “small but significant differences” to indicate that the magnitude is actually very small.

c) Line 172 – Lack of effect does not automatically make Massilia a hub microbe, any more than it makes sterile buffer a hub microbe because you see no effect with it. Hub microbes have very specific ecological and/or statistical characteristics that you do not show here, so do not claim Massilia is a hub.

d) Similarly in line 180, you have no evidence that Massilia is actually a keystone species (another word with very specific ecological meaning that was not tested here). Please remove that phrase.

e) Line 207 – Adding Massilia did not “enhance” the separation; it just changed it. There is no change in magnitude on your figures, so please replace “enhance” with “altered” or “shifted”.

f) Line 208 – This is talking just about metabolic profiles; there is no direct test of heterosis, so please do not claim there is a heterotic effect going on.

g) Line 226 – The TCA cycle is basically the core of central metabolism; it seems like a stretch to make any specific

inferences about what it could mean simply because it is involved in an enormous number of metabolic processes. Please scale back/remove.

h) Line 882 – Making claims of causation when all the data show are correlation

i) Line 340 – There is no evidence that Massilia is a crucial role because you did not test other microbes. It can play a role, definitely, but do not present it as if this is the key to all maize heterosis.

2) The flavonoid experiment shown in Figure 3d appears to be misinterpreted. The authors claim that Massilia is inducing flavones in the mutants and leading to a boost in growth. However, if the mutants cannot make flavones, they cannot be induced. If flavone induction were the key, one would expect the wild-types to show a response but not the mutants. Instead we see the opposite. It appears that Massilia is somehow *compensating* for the lack of flavone production by the plant, but it is certainly not *inducing* flavones in it. Please correct the text and your conclusions.

3) More thorough explanation of figures. Many of the figure legend do not adequately explain what the figure is showing, or have other errors. For example:

a) Figure 1 has two “e” sections, one of which is a copy-and-paste of part b, and the other doesn’t actually explain what the reader is looking at (what are the scale bars? What are the dots? Why is Oxalobacteraceae bolded? Legends should be relatively self-contained, so please explain.).

b) Figure S3 is unclear, especially in part B where there’s no indication as to which dots are contributing to which line (please use different shapes to distinguish them).

c) Figure 4a and 5b – The lower part is basically uninterpretable because it is so small. The rhizosheaths cannot be compared or even really seen. Please redo this to show in enough detail to make comparisons.

4) Point out that many traits are actually better in the absence of microbes. Figure S5 shows that both inbreds and hybrids do better *without* microbes around. Massilia is able to induce lateral root density, but everything else is actually better without a microbial community. This needs to be pointed out because the implication is that most microbes are actually bad for maize, whether inbred or hybrid. (e.g., in line 168)

Minor issues:

1) Please define rhizosheath when it’s first used in the Introduction. It is only defined much, much later, and it needs a clear definition the first time it is used.

2) I think you are misciting Wagner et al 2021 in line 69 when you say that paper show that hybrids underperformed in sterile soil but performance was restored upon adding microbes. This is actually the opposite of what that paper shows. If look look at, e.g, their Figure 1 A&B, you will see that inbreds do better in the absence of microbes and worse when they are present, and hybrid performance is unchanged [note the Y axes]. Figure 4 also implies that hybrids also do better in the field without microbes present. Please correct these citations to be in line with what the paper actually says.

3) Figure S2 and S3 seem important enough to merit being in the main text instead of the supplemental

4) Line 112 - Please rephrase the sentence “Soil microbes were found to...” because in context it sounds like this is part of your experiment, not a reference to the literature.

5) Line 293 – Please explain what a “rhizosheath transplantation experiment” is in a sentence or two so the reader understands without having to dig through the methods.

6) Line 445- Please give the sample size for this experiment. (Looks like 12 reps based on Supp dataset 2, but the reader shouldn’t have to back-calculate that)

Reviewer #2

(Remarks to the Author)

The authors investigated the contribution of rhizosheath-associated Massilia to maize heterosis and demonstrated that rhizosheath formation, which integrates root development, root exudates, and the associated microbiome, is an adaptive and heritable root trait. Overall, it’s a well-written article, and the data is provided where needed. However, a more detailed description of the methodology is required to allow for a comprehensive assessment of the data quality.

Line 414–424: Authors should provide more detailed information about the specific soil characteristics. Specifically, what is meant by “nutrient-rich soil”? It would be necessary to include quantitative data on the nitrogen (N) and carbon (C) levels, and other relevant micronutrient concentrations to clarify the meaning of nutrient-rich. Additionally, for both the low-N and high-N treatments, as well as for phosphorus treatments, it is necessary to provide explicit N and P concentrations.

Line 561–573: The authors need to clarify which pipeline was used in R for analyzing amplicon sequencing data? Specifically, what parameters were selected for initial quality control (QC) after demultiplexing, as well as for forward and reverse sequence trimming? The authors should upload their R code to a publicly accessible platform so that reviewers and future readers can review it.

ASV Analysis: How many amplicon sequence variants (ASVs) were observed before quality filtering, and how many remained after filtering? How many of these ASVs were classified under known phyla or genera? Additionally, which classifier was used for taxonomic assignment of the ASVs?

Analysis of Heterosis in Microbiome Communities: In the section titled "Analysis of Heterosis in Microbiome Communities," it would be helpful to rephrase the first sentence for clarity. Specifically, the authors should define the criteria for highly abundant and prevalent ASVs (e.g., >0.05% relative abundance in $\geq 20\%$ of samples)? Also, how was the significance of the heritability determined? Was a bootstrap method employed? The R code should be shared to a public database to facilitate a more thorough review of the methods used in this paper.

ASV Reproducibility Criterion: Previous studies often employ a stricter reproducibility criterion (e.g., ASVs must be observed at least 25 times across at least five samples) for reproducibility (<https://doi.org/10.1111/nph.16730>). What was the authors' rationale for selecting the filtering criterion of at least 10 reads in two or more samples in this study?

N₂O Measurement: N₂O measurements are most accurately conducted using Gas Chromatography with an Electron Capture Detector in most studies. The authors need to show a calibration curve for the LI-7820 N₂O Trace Gas Analyzer (used in this study) to ensure the relative accuracy of this device.

Gas Measurement Setup: In the experiment, one plant was moved to a plastic bucket after four weeks. More details are needed here - the authors must specify whether the plant was moved with or without soil? Additionally, it is necessary to provide a more detailed description of the methods used for gas measurement in the experiment.

Line 711–713: The methodology mentions that one plant was moved to a plastic bucket, please clarify whether this refers to a single plant at a time or if multiple plants were used. How many replicates were used for the N₂O measurements?

Grammatical Corrections:

Line 335: "has is a promising" should be revised for clarity.

Reviewer #3

(Remarks to the Author)

He et al. describe a study investigating the role of *Massilia* in heterosis in Maize mostly under nitrogen limiting conditions. The authors looked at heterosis response under several abiotic stress conditions in a diverse panel of maize hybrids and then focused on nitrogen limitation and *Massilia*. One of the key traits they analyzed is rhizosphere size and root exudates under several conditions. The paper could be an important contribution to understanding the roles of microorganisms in heterosis in maize. Especially the observed interaction of abiotic stresses x root associated microbes and heterosis would be a key contribution to the field. I did find that hypotheses and statements about causality were often muddled and unclear. Additionally, I do believe that additional controls might be needed to really be able to make solid statements about correlations and causation. Finally, I do think that the language is sometimes very specific and many readers interested in plant-microbe interactions in general would not understand the heterosis specific terminology. I would recommend to explain this terminology when it appears to make the manuscript more accessible.

Major concerns:

1. The term rhizosphere is not properly introduced anywhere in the manuscript, over time a picture formed for me of what exactly it is supposed to be. This needs to be done earlier in the manuscript. Also, I am surprised that correlation between rhizosphere size and root weight are not shown. I would assume that the more simple measure of root weight often correlates well with rhizosphere size. Since root weight data is not correlated to rhizosphere size it is unclear if the complex rhizosphere size measure is really needed.
2. Throughout the results section hypotheses are often not clearly stated and statements about causation are made when really it is just correlations, and in some cases not even that where it appears more like assumptions are being made. I will give some specific examples in the specific comments section. This would need to be carefully reworked to be clear about what can actually be determined from the data.
3. Data in figure 2. This data seems to miss an essential control. Since it has been shown previously that autoclaving releases toxins and nutrients, it is unclear if observed effects are changes in the presence/absence of a microbiota or if they are simply due to changing the abiotic soil matrix, and therefore effects of *Massilia* are also unclear as it could be an interaction with changes in the soil due to autoclaving. Either gamma-irradiation of soils would need to be used, or controls that re-inoculate sterilized soil with the soil microbiota i.e. through use of slurries or introduction of a narrow layer of unsterilized soil in the pots, or mixing in of a small quantity of non-sterile soil.
4. The metabolomics data and its interpretation on page 6 is confusing. Supposedly exudates were measured, however, the focus of the discussion is on metabolites that are not usually exuded and that are discussed in the context of intracellular pathways e.g. TCA cycle. Why would these metabolites be released into exudates. Additionally ABC transporters are mentioned as "enhanced", however, to my knowledge there is no way to determine abundances or even infer abundances of transporters based on metabolomics data. So what was done here?
5. Figure 1. There are two explanations for panel e in the caption, neither one seems to explain what is shown. Explanation

needs to be added on what is shown in that panel in detail.

6. Figure 6. Is confusing and does not have much content. For example it seems to suggest that *Massilia* makes the flavonoids. I would remove or rework to really be a model of the role of *Massilia*. Right now it is a mix of potential functions and also methods?

Specific comments:

7. In the introduction structuring of microbiomes in hybrids and inbreds is discussed, but the key papers by Dr. Maggie Wagner on this topic are not referenced. I see you have one of the relevant citations in the discussion, but all her papers should be referenced in the introduction. Right now the ratio of self-citation is too high.

8. Lines 69-73. Remove the word "beneficial" as the fitness impacts of these microbes are unknown.

9. Line 76. You use the term "keystone microbial taxa" here but it is unclear how you define it and why you use it here.

10. L. 106. I do not follow your argument that this data indicates potential involvement of the "soil microbiome" in shoot biomass heterosis.

11. L. 109. "...reported in other species." Actually cites a paper on Maize, ref. 17.

12. L. 112. Sentence starting "Soil microbes...". I don't see the relevance of this sentence. I would suggest to delete to avoid confusion.

13. L. 118. There is no clear argument given why Oxalobacteraceae were chosen as there seem to be many other taxa that are associated with heterosis in more hybrids. Rationale needs to be clear. Looks like cherry picking right now

14. L. 128. Explain why you chose Mo17 X H84 instead of a A554 or H99 hybrid?

15. L. 134 – 136. Statement that starts with "This suggests..." This is leading as there is no evidence shown that implicates Oxalobacteraceae.

16. L. 162. After "of" a word is missing

17. L. 172. Explain term "functional microbial hub" in this context.

18. L. 182. Replace "with" with "in"

19. L. 190. I am not sure what you mean with "metabolites involved in regulating heterosis manifestation". How are metabolites the regulators of anything?

20. L. 196. Can you define "metabolic components"? are these specific mass-retention time combinations or what is meant here?

21. L. 234. Can you explain how flavones help with nitrogen metabolism and uptake?

22. L. 238-247. I am confused these results and the conclusion that they "demonstrated that *Massilia* could enhance plant growth by influencing flavonoid secretion...". How can *Massilia* enhance flavonoid secretion when the genes for flavonoid generation are knocked out. Either explain better on what the knockout results mean or specify a interpretation of the data that makes sense.

23. L. 263-264. Not sure what is meant by "suggesting that rhizosheath formation partially determines the manifestation of heterosis...". This makes it sound as if the rhizosheath is not part of heterosis expression?

24. L. 270. "drove" are you sure about causality here.

25. L. 278 – 281. Causal statements here make no sense in the context of the data.

26. L. 286/86. I do not understand the hypothesis. Seems circular.

27. Figure 1. Panel b explain PR and SR.

28. Figure 5. Panel a. Are the tests done across all three separate graphs in this panel? Please explain the stats better in the caption. Also why is everything orange in the 3 graph?

Reviewer #4

(Remarks to the Author)

This manuscript describes a series of growth-chamber experiments that compare the interactions of hybrid vs. inbred maize with the soil microbiome, with an emphasis on the role of *Massilia* sp. in plant vigor under N-limiting conditions. The role of microbes in heterosis is an important and timely topic, and in-depth studies like this one are very valuable.

There are several fascinating patterns to come out of this paper, such as the observation that hybrids' root phenotypes are generally more sensitive to soil microbiome than inbreds'. Overall this paper reports a ton of valuable data, which undoubtedly would be an impactful addition to the literature. For example, measuring the effects of soil microbes on heterosis for more hybrids and more traits, and in a variety of abiotic environmental contexts, is definitely novel. I really admire the breadth of the experiments and the depth of data, especially on understudied root phenotypes. This work could definitely move us forward in our understanding of how hybrids and inbreds interact differently with belowground microbiota.

Despite all those positive attributes, I am not convinced that the conclusions of this manuscript are accurate, for reasons detailed below. That is, I have serious concerns about how the results were interpreted. It seems as though the data were consistently over-interpreted to support a particular "story". I believe it should be possible to re-frame and re-write this paper while avoiding spurious conclusions, and the result will still be a paper with high value, novelty, and impact.

1. Throughout the paper, I noticed that many of the stated conclusions were not actually supported by the evidence. In some cases they seem logically unrelated to the experiment/results that they are referring to. Examples of this problem:

1A. Line 762: claims that heterosis is stronger "under stress... especially under N stress" which is misleading because the other 2 stresses clearly show distributions of MPH values that are smaller (low P) or equal to (drought) the control.

1B. Line 849: claims that panel (a) illustrates a response to N deficiency, but all of the photos in panel (a) are from the low-N treatment, without a control for comparison

1C. Line 106: claims that an increase in MPH due to N limitation shows an influence of the soil microbiome, which is

misleading because there was no microbiome manipulation.

1D. Lines 152-153: how does a correlation between lateral root density MPH and shoot biomass MPH support this claim?

1E. Lines 172-173: This study does not provide evidence that *Massilia* is a “hub” taxon, or that it preferentially colonizes lateral roots over other root types. In fact, the cited figure S5 shows mostly null results from inoculation with *Massilia*

1F. Line 178: The claim that lateral root density altered shoot biomass heterosis is not supported. This analysis was correlative only.

1G. Line 186: This claim about metabolic responses to N stress is referring to an experiment that only included a microbiome manipulation (addition of *Massilia*), not a manipulation of N availability, and also measured just biomass and lateral root density, not metabolism.

1H. Lines 200-201: These findings do not suggest anything about the role of secondary metabolites in influencing root development and microbiome formation - they are purely descriptive, a list of some major metabolic pathways that were detected.

1I. Lines 252-253: The observation that hybrids have larger rhizosheaths is not evidence that the rhizosheath leads to hybrid vigor.

1J. Lines 257-258: All of the analyses involving rhizosheath size and plant growth (including what is shown in Fig. 4c-d + Figs. S12-S15) are correlative and do not support the claim that the rhizosheath “significantly influenced plant growth and N uptake” or “contributed to heterosis for maize biomass”. It is easy to come up with an explanation (equally untested) with the opposite causality: hybrids with high shoot mass heterosis do more photosynthesis and exude more resources into the rhizosphere, thus forming a correspondingly larger rhizosheath.

1K. Line 270: Same criticism as 1J. It is not rigorous to claim that root traits “strongly drove heterosis of rhizosheath size” from the observation that their MPH values are correlated with each other.

1L. Line 278: Same criticism as 1J, this analysis does not show that oxaloglutarate influenced lateral root development, only that they are correlated

1M. Lines 280-281: I do not follow the logic here - in all these experiments *Massilia* was added experimentally, and rhizosheath size was the response variable. But here, the claim is that the results can be summarized as the rhizosheath “recruiting” *Massilia*.

1N. Line 308: The observation that genotypes differed in N₂O emission does not mean that large rhizosheaths have a causal effect on N₂O emission.

2. The level of detail provided in the Methods section is not sufficient for understanding what was done, much less being able to replicate the experiments and analyses. Examples of this problem:

2A. Lines 415-423: One soil was used to generate 4 treatments, but no information is given about how those treatments were implemented (other than holding the D soil at 22% water holding capacity)

2B. Line 468 how were the samples preserved?

2C. Line 503-306 what isolates are they, where did they come from, etc.

2D. A description of how ASV tables were generated from raw 16S-v4 sequences is completely missing

2E. Mentions of statistical tests such as constrained ordination, PERMANOVA, ANOVA, linear mixed models, etc. are meaningless without also specifying the formulas used, because the interpretation of each variable in the model depends on which other variables were also in the model.

2F. Basic experimental design information such as the number of replicates per genotype per treatment, randomization, blocking, etc. are missing.

3. I am concerned about the validity of the analyses comparing heterosis across treatments. Fig. 1d shows 30 observations of MPH per environmental treatment, however, only 10 hybrid genotypes were included in the experiment (Fig. 1a). Typically, MPH is a property of a particular hybrid triplet, and each individual MPH value is estimated using data from several replicates of the F1 genotype as well as both parental genotypes. In other words, it is not clear how 30 MPH values were generated using only 10 hybrids - the independence of those values needs to be justified. If there are indeed 3 measurements of MPH per hybrid, then a standard ANOVA is not sufficient to account for the data structure, in which the 3 MPH values from the same hybrid are expected to be more similar to each other than to the other 27. Therefore, the 30 values are not a true random sample and need to be modeled using a hierarchical model, e.g., with the hybrid genotype as a random grouping factor. The same applies to the experiment shown in Fig. S2 - how did you generate 5 independent MPH values per triplet?

4. It appears that the “low N” treatment was done by mixing the field soil with 50% v/v river sand, whereas the “high N” treatment was the original field soil with added N fertilizer (lines 494-496). Unfortunately, this seems like a very flawed way to create two treatments that will be informative about the effect of N availability. Undoubtedly these treatments did differ in N, however, by diluting one of them with 50% sand a wide range of other soil properties were likely also altered, such as soil texture, drainage, organic matter, concentration of many other nutrients, possibly pH, etc. In my opinion, this issue greatly compromises the ability to interpret differences between the treatments as “caused by N limitation”.

5. The flavonoid secretion experiments/Fig. 3d: You observed that the addition of *Massilia* improved the growth of two mutants that are deficient in flavonoid synthesis, but not their wild-type counterparts. From this you conclude that increasing flavonoid secretion is a mechanism by which *Massilia* promotes plant growth. But doesn't this result actually indicate the opposite - that *Massilia* enhances plant growth by some mechanism other than flavonoid secretion? Because the mutants should be incapable of having that particular response, so they should respond less than the wildtypes.

6. The data availability statement does not mention the phenotype data.

Minor comments:

7. The term “rhizosheath” needs to be clearly defined early in the paper. How is it different from “rhizosphere”?
8. Figure 1e is very confusing. The caption needs to be expanded to clearly explain what the dendrogram shows, what each stacked barplot shows, and what the list of taxa indicates.
- Line 98: it is not clear how this soil relates to the D, low N, and low P treatments.
- Line 109: one of the cited studies is from the same species
- Lines 122-123: redundant with previous sentence
- Line 128: Clarify that the inbred parents were also included in this experiment, as this is the only way to measure heterosis.
- Lines 148-149: Unclear writing. Which SynComs promoted lateral root formation, and compared to what?
- Line 210: undefined acronyms (VIP, OPLS-DA)
- Line 317: Enriched relative to what?
- Line 442: What is a “4 mm field soil mixture”?
- Line 449: This needs more explanation - how can soil microbes exhibit heterosis when they are mostly haploid, and often reproducing asexually?
- Line 491: How was “the potential to express heterosis under Oxalobacteraceae enrichment” assessed when selecting genotypes for use in this study?
- Line 507: Clarify that the inbred lines were also included, as it is impossible to measure heterosis without them.
- Line 579: There was no previous mention of blocks as part of the experimental design.
- Line 597: The previous lines indicate the rhizosheath was washed off prior to exudate collection. Please clarify
- Line 657: what temperature was the water bath? Was sonication used? Or was this just a soaking procedure?
- Line 683: Indicate type of correlation, e.g. Spearman, Pearson

Version 1:

Reviewer comments:

Reviewer #1

(Remarks to the Author)

By and large, the authors have adequately responded to my requested revisions. My only remaining concern is still about Figures 4 & S15, where the rhizosheath is still not visible on the images of the whole root system. (In 4a, the roots on the right actually appear to be cleaned; they don't seem to have any visible rhizosheath adhering to them.) I don't feel these parts actually contribute to what they're cited for, and I would rather they just be removed. (The left half of 4a, which does clearly show rhizosheath sizes, is fine to stay)

Reviewer #3

(Remarks to the Author)

Thank you for addressing all my comments so carefully. Great job!

Reviewer #4

(Remarks to the Author)

I appreciate the authors' work to address the concerns I raised in my original review. I did not have time to read and consider the other reviewers' concerns and evaluate whether they were addressed.

My biggest concern—the consistent overstatement of results—was mostly addressed. In a few places, the authors still imply a causal relationship that was not supported by their data (examples: line 41, 87-88, 321-322, 464 “rhizosheath-mediated heterosis”). Conversely, in other places they actually understate their results, mainly describing an experimental manipulation (e.g., an inoculation) as “associated with” an outcome (examples: lines 255-256). When a factor was directly

manipulated, it is appropriate to say that it caused an effect.

I found the revised version harder to read than the original—it would benefit from substantial re-structuring to avoid repetition and clearly explain the differences between experiments. For example, the clause “To determine whether *Massilia* influences heterosis of root development under nitrogen-poor conditions...” (or slightly re-worded) appears 3 times in quick succession (lines 156, 175, 201) - and each time, it is used to introduce a different experiment or analysis. Additionally, there are several extremely long paragraphs. Revision is needed to organize the content in a more logical way.

Some of the conclusions seem a bit circular. For example, does *Massilia* induce apigenin or does apigenin promote *Massilia* colonization? Both are implied almost interchangeably throughout the paper. I found the mechanistic diagram in Figure 6 to be quite vague and it did not really help answer my questions about the apparent circularity of the relations between *Massilia*, rhizosheath, and flavonoids.

I request a clearer and more thorough discussion of whether there is any evidence at all, in this work or in other published work, that the rhizosheath is a mechanism of heterosis, or somehow more important than other plant traits for the manifestation of heterosis. My a priori expectation is that it is just another trait that is better in hybrids than inbreds, like height or yield. But no one is arguing that height is a “mechanism” of heterosis even though bigger plants can do more photosynthesis, etc.

Figure 1e is somewhat improved but still unclear to me. 1) It seems to imply that the different host genotypes had different numbers of ASVs (shown by differing bar lengths), is that correct? 2) The bars for “Kingdom” are the same length as the bars for “Heterosis type”, implying that 100% of all ASVs showed heterosis. 3) Heterosis is a property of a hybrid, not an inbred line, and the genotypes in the dendrogram are inbred lines - so how should we interpret the “heterosis type” panel? 4) I can't tell what the “compartment” panel is showing. What is meant by “enrichment in the rhizosphere or root compartment” - does this just show whether the ASV's relative abundance was higher in roots or rhizosphere? Please add a clearer explanation. 5) What is meant by “most prevalent bacterial families exhibiting microbiome heterosis across genotypes” (line 947 referring to the rightmost panel)

Despite these lingering concerns, I think the dataset is a very valuable one and the authors describe some extremely interesting patterns that are certainly worth publishing. After some additional revision, I do think this paper will advance the field and inspire a lot of follow-up work on the involvement of microbes in heterosis.

Some minor things:

Lines 229-230: What do these percentages refer to?

Line 236: What does “assembly of metabolic features” mean?

Line 242: How were those cutoffs for L2FC determined?

Lines 376-377: positive R indicates a positive correlation, not negative

Line 414: nitrogen use efficiency has a specific definition and I do not see any prior mention to this metric

Line 453: “expression of genes” appears to be referencing metagenomics data, which describes the abundances of functional gene groups in the combined genomes of the microbiome - it says nothing about the expression of those genes

Line 699-700: Provide more detail about root hair measurement. How many hairs were measured per root, from what part of the root, etc.

Reviewer #1 (Remarks to the Author)

He et al have performed a thorough and data-rich investigation of *Massilia* and its interactions with maize, especially with regard to the rhizosheath and heterotic combinations of the plant. My evaluation of this manuscript is largely positive, but with caveats. On the one hand, the experiments and data appear to be very thorough and well described, and I especially like how much of their original data has been made available as supplemental datasets. This is a practice that I wish were more common among authors. And, in general, the individual experiments (with one exception, below) appear to be carried out and interpreted well. However, there are some major shortcomings that I feel need to be addressed before publication, which I have outlined below:

Major issues:

1) Conclusions. The individual experiments and data are of high quality, but the conclusions are very much overblown. Throughout the manuscript the authors talk about *Massilia* “causing”, “contributing to”, or otherwise being a causal factor of heterosis. This overemphasizes the role of the microbe and makes it seem as if this is some key link in heterosis that has been long overlooked. Instead, all these results show a *_manifestation_* of heterosis with respect to the microbiome. Any improved interactions with *Massilia*, flavonoids, root growth, etc., are the *_outcomes_* of heterosis, not its cause. Please correct this throughout the manuscript. The authors have created a very nice model system to study heterosis with respect to the maize microbiome, but they need to be clear that is what it is: a model system. There is no evidence that *Massilia* is the microbe mediating similar responses in maize around the globe, and the most likely situation is that there are a great many microbes that can fill a similar niche. Some examples of things to correct:

Our response:

Thanks for the reviewer’s positive and constructive comments on our work! We also appreciate the reviewer’s insightful feedback regarding the interpretation of our findings. We agree that the current wording in some parts of the manuscript may overstate the causal role of *Massilia* in heterosis. Our intention was to illustrate how specific microbiome traits (such as *Massilia*-enhanced lateral root development and flavonoid metabolism) manifest in association with heterotic maize genotypes, especially under nitrogen stress.

We acknowledge that:

- These traits represent downstream consequences of hybrid vigor, not the root cause of heterosis.
- *Massilia* may act as a facilitator of improved nitrogen use efficiency or rhizosheath development in certain genetic backgrounds and soil contexts.
- The causality is associative, and *Massilia* should be considered one of potentially many microbial taxa capable of interacting beneficially with hybrid maize.

We have revised the manuscript throughout to:

- Replace terms like “cause,” “contribute to,” or “underlying heterosis” with more accurate alternatives such as “associated with,” “in association,” or “related.”
- Emphasize that our study provides a model system to explore how microbial interactions are modulated in heterotic maize, rather than a universal mechanism.
- Clarify that functional redundancy in the rhizosphere is likely, and *Massilia* is an example of a keystone taxon that interacts with hybrids in our experimental system, rather than the sole mediator of these effects across all environments.

Specific examples revised include:

- Title: now reads “Rhizosphere-associated *Massilia* may facilitate root traits linked to maize heterosis”.
- Discussion: We now clearly state that the observed microbiome shifts are consequences of hybrid root architecture and exudation patterns, rather than drivers of heterosis per se.

Reviewer #1 asked:

a) Line 106 – There is no evidence of the microbiome being involved in heterosis here. Please delete that. All you’re showing is that nitrogen stress is important.

Our response:

We have revised this sentence to clarify that the data demonstrate a stronger heterosis response under low nitrogen, without implying microbiome involvement at this stage of the analysis.

Revised sentence:

“Specifically, low nitrogen stress significantly increased the MPH of shoot biomass compared to the control treatment ($p = 3.77e-06$), indicating that heterosis for shoot biomass is more pronounced under nitrogen deprivation.”

Reviewer #1 asked:

b) Line 108 – Change “significant differences” to “small but significant differences” to indicate that the magnitude is actually very small.

Our response:

This has been done.

Reviewer #1 asked:

c) Line 172 – Lack of effect does not automatically make *Massilia* a hub microbe, any more than it makes sterile buffer a hub microbe because you see no effect with it. Hub microbes have very specific ecological and/or statistical characteristics that you do not show here, so do not claim *Massilia* is a hub.

Our response:

We agree that the designation of a “hub microbe” requires specific ecological or network-based analyses, which we did not perform in this study. Our use of the term was imprecise and has been revised for accuracy.

Revised sentence:

“These results suggest that *Massilia* may play a functional role in lateral root development and nitrogen response, preferentially colonizing lateral root tissue under nitrogen-poor conditions.”

Reviewer #1 asked:

d) Similarly in line 180, you have no evidence that *Massilia* is actually a keystone species (another word with very specific ecological meaning that was not tested here). Please remove that phrase.

Our response:

We have removed the term “keystone species” from Line 180 and revised the sentence to focus on the observed functional associations.

Revised sentence:

“These results suggest that beneficial interactions between lateral root density and *Massilia* are associated with the enhanced growth performance observed in maize hybrids under nitrogen-limited conditions.”

Reviewer #1 asked:

e) Line 207 – Adding *Massilia* did not “enhance” the separation; it just changed it. There is no change in magnitude on your figures, so please replace “enhance” with “altered” or “shifted”.

Our response:

We agree that our results show a change in metabolic composition upon *Massilia* inoculation, but not a quantified enhancement in separation magnitude. We have revised the wording to more accurately describe the observed effect.

Revised sentence:

“The absence of a soil microbiome and inoculation with a single *Massilia* isolate in sterilized soil further altered the composition and assembly of metabolic features (Supplementary Fig. 6b).”

Reviewer #1 asked:

f) Line 208 – This is talking just about metabolic profiles; there is no direct test of heterosis, so please do not claim there is a heterotic effect going on.

Our response:

We agree that while the observed differences in metabolic profiles between inbred lines and hybrids suggest altered root exudation patterns, these results do not directly demonstrate heterosis for metabolism. Therefore, we have revised the sentence to avoid implying a heterotic effect.

Revised sentence:

“These findings suggest that *Massilia* may influence the metabolic composition of root

exudates in maize under nitrogen deprivation, with distinct profiles observed between inbred lines and hybrids.”

Reviewer #1 asked:

g) Line 226 – The TCA cycle is basically the core of central metabolism; it seems like a stretch to make any specific inferences about what it could mean simply because it is involved in an enormous number of metabolic processes. Please scale back/remove.

Our response:

We appreciate the reviewer’s point and agree that attributing specific functional implications to the enrichment of the TCA cycle—given its role as a core component of central metabolism—can be speculative without further mechanistic evidence. To avoid overinterpretation, we have revised this part of the text to state the observation more neutrally and removed broader functional claims.

Revised sentence:

“Specifically, soil sterilization was associated with increased representation of TCA cycle-related metabolites (Supplementary Fig. 9), consistent with general shifts in central metabolism under altered microbial conditions.”

Reviewer #1 asked:

h) Line 882 – Making claims of causation when all the data show are correlation

Our response:

We agree that our data demonstrate correlations between rhizosheath size, *Massilia* enrichment, and growth performance, but do not establish causation. We have revised the sentence to more accurately reflect this relationship as correlative rather than causal.

Revised sentence:

“These results indicate that rhizosheath-associated *Massilia* is correlated with reduced reactive nitrogen loss, thereby potentially contributing to improved nitrogen use efficiency.”

Reviewer #1 asked:

i) Line 340 – There is no evidence that *Massilia* is a crucial role because you did not test other microbes. It can play a role, definitely, but do not present it as if this is the key to all maize heterosis.

Our response:

We agree that our experiments specifically demonstrate an association between *Massilia* and heterosis-related traits in maize, but do not provide comparative evidence involving other microbial taxa. As such, we cannot claim that *Massilia* plays a crucial or exclusive role.

We have revised the sentence accordingly to reflect this limitation.

Revised sentence:

“Our findings suggest that *Massilia* may play a role in regulating maize heterosis by

influencing lateral root development and flavonoid metabolism under nitrogen deprivation.”

Reviewer #1 asked:

2) The flavonoid experiment shown in Figure 3d appears to be misinterpreted. The authors claim that *Massilia* is inducing flavones in the mutants and leading to a boost in growth. However, if the mutants cannot make flavones, they cannot be induced. If flavone induction were the key, one would expect the wild-types to show a response but not the mutants. Instead we see the opposite. It appears that *Massilia* is somehow *compensating* for the lack of flavone production by the plant, but it is certainly not *inducing* flavones in it. Please correct the text and your conclusions.

Our response:

We agree that our original interpretation of the results in Figure 3d was imprecise. As the *C2-Idf* and *fns11* mutants are deficient in flavone biosynthesis, it is indeed not possible for *Massilia* to induce flavone production in these genotypes. The increased shoot biomass observed in the mutants following *Massilia* inoculation therefore cannot be attributed to *Massilia*-mediated stimulation of endogenous flavone biosynthesis. Rather, the results suggest that *Massilia* may partially compensate for the metabolic deficiency—possibly through microbial metabolites or other plant–microbe signaling mechanisms—but not through induction of flavones in the maize mutants.

We have corrected the relevant sections of the Results and Discussion to reflect this point:

Revised sentence (Results):

“These findings suggest that *Massilia* may enhance plant growth even in the absence of endogenous flavone production, potentially by compensating for the flavonoid deficiency in mutant plants through alternative mechanisms.”

Revised sentence (Discussion):

“*Massilia* promotes lateral root growth and enhances the ability of the rhizosphere to support beneficial microbial communities, potentially through interactions with flavonoid-related pathways in wild-type plants, and by compensating for flavonoid deficiencies in mutants—thereby improving nitrogen use efficiency and overall plant performance.”

Reviewer #1 asked:

3) More thorough explanation of figures. Many of the figure legends do not adequately explain what the figure is showing, or have other errors. For example:

a) Figure 1 has two “e” sections, one of which is a copy-and-paste of part b, and the other doesn’t actually explain what the reader is looking at (what are the scale bars? What are the dots? Why is Oxalobacteraceae bolded? Legends should be relatively self-contained, so please explain.).

Our response:

We thank the reviewer for pointing out the errors and omissions in the Figure 1 legend. We have corrected the duplicated labeling of panel “e,” ensured that each panel is uniquely labeled, and clarified all elements shown in the figure, including the meaning

of dots, scale bars, and the emphasis on *Oxalobacteraceae*. The revised legend now provides a self-contained explanation of all panels and symbols used.

Reviewer #1 asked:

b) Figure S3 is unclear, especially in part B where there's no indication as to which dots are contributing to which line (please use different shapes to distinguish them).

Our response:

We appreciate the reviewer's observation regarding the lack of clarity in Supplementary Figure S3B. We agree that the data points corresponding to different hybrid combinations should be visually distinguishable to clearly show their contribution to the regression analysis. To address this, we have revised it to use distinct marker or colors or shapes as main Figure 2c and updated the figure legend accordingly to explain these symbols.

Figure 2c Correlation between MPH of different root traits and MPH of shoot dry weight after inoculation with independent soil-derived *Oxalobacteraceae* isolates in nitrogen-poor soil. Scatter plots show combined data from inoculation experiments with best fit (color solid line) and 95% confidence interval (color shading) for linear regression (Pearson correlation). Different symbol shapes represent different synthetic communities (SynCom) (e.g., *Massilia* ASV37 alone, a 17-member synthetic bacterial community (SynCom17) of *Massilia* isolates that did not include ASV37 and an 18-member *Massilia* (SynCom18) including SynCom17 and ASV37 conditions). Different colors for confidence interval and line represent correlation of different root traits (e.g., root weight, root length, and lateral root density). This data suggests that the interactions between lateral root density and *Oxalobacteraceae* associates with maize heterosis in shoot dry weight. B73, Mo17, H84, B73 × Mo17, B73 × H84, and Mo17 × H84 were selected.

Reviewer #1 asked:

c) Figure 4a and 5b – The lower part is basically uninterpretable because it is so small. The rhizosheaths cannot be compared or even really seen. Please redo this to show in enough detail to make comparisons.

Our response:

We thank the reviewer for pointing out the issue with image resolution and interpretability in Figures 4a and 5b. We agree that the lower panels showing rhizosheath size were too small and lacked sufficient detail for meaningful visual comparison across genotypes. The image in lower panels from Figure 5b has been changed as Supplementary Figure 15.

To address this, we have:

- Replaced the original panels with higher-resolution, zoomed-in images of representative root systems in the revised Figure 4a (see also below).
- Selected representative images for inbreds and hybrids to illustrate clear differences in rhizosheath size.
- Updated the figure legends to describe the improved visualization.

These changes improve visual clarity and allow the reader to directly compare rhizosheath formation across genotypes.

Figure 4a Rhizosheath performance for hybrids and their parental inbred lines for 5-day and 4-week maize seedlings in nitrogen-poor soil. The panel shows representative images of rhizosheath formation for 5-day seedlings (left) from primary root and 4-week seedlings in total root system (right) after sowing in nitrogen-poor soil. Images have been enlarged and standardized for size and brightness to enable direct comparison of rhizosheath size between inbreds and hybrids.

Supplementary Figure 15. Rhizosphere performance for different genotypes for 4-week maize seedlings in nitrogen-poor soil. Scale bars and image scaling have been standardized across genotypes for visual consistency.

Reviewer #1 asked:

4) Point out that many traits are actually better in the absence of microbes. Figure S5 shows that both inbreds and hybrids do better **without** microbes around. Massilia is able to induce lateral root density, but everything else is actually better without a microbial community. This needs to be pointed out because the implication is that most microbes are actually bad for maize, whether inbred or hybrid. (e.g., in line 168)

Our response:

We appreciate the reviewer's critical observation regarding the broader patterns shown in Figure S5. We agree that our original interpretation underemphasized a key result: most root traits, aside from lateral root density, were actually improved in the absence of the soil microbiome in both inbred and hybrid maize. This suggests that, under nitrogen-poor conditions, the bulk microbial community may exert neutral or even negative effects on certain aspects of root development.

To address this, we have revised the relevant text (e.g., Line 168) to acknowledge this pattern and reframe the interpretation more carefully.

Revised sentence:

"Our results revealed that, under nitrogen-poor conditions, most root traits were improved in sterile soil, suggesting that the native soil microbiome may suppress root development in both inbred and hybrid maize. Notably, only lateral root density was

consistently promoted by *Massilia* inoculation, indicating that specific microbial taxa may compensate for or override broader negative microbial effects.”

We have also updated the Discussion to reflect this nuance, emphasizing that while *Massilia* appears beneficial, the overall microbiome may contain antagonistic or competitively inhibitory members that impact maize performance negatively in this context as “Interestingly, under nitrogen-poor conditions, most root traits were enhanced in the absence of the native soil microbiome, suggesting that bulk microbial communities may exert neutral or even inhibitory effects on maize root development. These findings highlight the importance of identifying specific microbial taxa—such as *Massilia*—that can compensate for or counteract broader negative interactions within the root microbiome.”

Reviewer #1 asked:

Minor issues:

1) Please define rhizosphere when it's first used in the Introduction. It is only defined much, much later, and it needs a clear definition the first time it is used.

Our response:

We have now added a definition when the term first appears:

Revised sentence:

“In this study, we explored the potential role of rhizosphere formation—defined as the layer of soil that adheres tightly to roots, influenced by root hairs, mucilage, and microbial activity—in maize heterosis under nitrogen deprivation.”

Reviewer #1 asked:

2) I think you are misciting Wagner et al 2021 in line 69 when you say that paper show that hybrids underperformed in sterile soil but performance was restored upon adding microbes. This is actually the opposite of what that paper shows. If look look at, e.g, their Figure 1 A&B, you will see that inbreds do better in the absence of microbes and worse when they are present, and hybrid performance is unchanged [note the Y axes]. Figure 4 also implies that hybrids also do better in the field without microbes present. Please correct these citations to be in line with what the paper actually says.

Our response:

We thank the reviewer for this critical observation. Upon re-examining Wagner et al. (2021), we acknowledge that our citation and interpretation were incorrect. As the reviewer rightly notes, that study demonstrates that inbreds tend to perform better in sterile soil, while hybrids show little to no change, or in some cases also perform better in the absence of microbes. This indicates that the soil microbiome may have a more negative impact on inbred lines, and challenges the view that microbial presence universally restores hybrid performance.

We have corrected the citation and revised the text accordingly.

Revised sentence:

“For example, Wagner et al. (2021) showed that inbred maize lines often perform better in sterile soil, whereas hybrids exhibit less sensitivity to microbial presence, suggesting

that the microbiome may disproportionately inhibit inbred performance rather than enhance hybrid vigor.”

Reviewer #1 asked:

3) Figure S2 and S3 seem important enough to merit being in the main text instead of the supplemental

Our response:

We appreciate the reviewer’s suggestion regarding the importance of Figures S2 and S3. We agree that these figures provide key experimental evidence supporting the role of *Massilia* and the soil microbiome in shaping maize heterosis under nitrogen-limited conditions.

In response, we have moved both Figure S2 and Figure S3 into the main text:

Supplementary Figure S2 is now as new Figure 2a, directly supporting the main findings on the effect of soil sterilization on shoot biomass heterosis in response to nitrogen application amount.

Supplementary Figure S3a and Figure S3b is now as new Figure 2b and Figure 2c, highlighting the correlation between lateral root density and shoot dry biomass, and the impact of *Massilia* and synthetic communities on root traits.

We have updated figure numbering and references throughout the manuscript to reflect these changes and revised the figure legends for clarity and consistency with the main text.

Reviewer #1 asked:

4) Line 112 - Please rephrase the sentence “Soil microbes were found to...” because in context it sounds like this is part of your experiment, not a reference to the literature.

Our response:

We thank the reviewer for pointing this out. We agree that the original sentence could be misinterpreted as referring to our own experimental findings, rather than citing prior work. To clarify this distinction, we have rephrased the sentence to make it explicitly refer to previously published literature.

Revised sentence:

“We then examined how hybrid breeding affects microbiome community composition and observed small but significant differences between inbred lines and their hybrids (Supplementary Fig. 1). Similar findings have been reported in maize (Wagner et al., 2021).”

Reviewer #1 asked:

5) Line 293 – Please explain what a “rhizosheath transplantation experiment” is in a sentence or two so the reader understands without having to dig through the methods.

Our response:

We agree that a brief explanation of the rhizosphere transplantation experiment in the Results section improves clarity for the reader. To address this, we revised sentence and added schematic diagram as Supplementary Figure 19 in Method section.

Supplementary Figure 19. Rhizosphere transplantation experiment. To investigate the influence of rhizosphere size on maize plant growth against low nitrogen stress, maize plants with different rhizosphere sizes were transplanted into nitrogen-deficient soil previously conditioned by either large-rhizosphere (genotype R109B) or small-rhizosphere (genotype Ky228) or original-rhizosphere maize plants to assess the impact of rhizosphere size on plant growth under nitrogen limitation

Revised sentence:

“We then performed a rhizosphere transplantation experiment, in which seedlings were grown in soil that had previously hosted one of these genotypes. This allowed us to assess whether the rhizosphere-associated microbial community from donor plants influenced the growth of recipient seedlings in nitrogen-poor soil.”

Reviewer #1 asked:

6) Line 445- Please give the sample size for this experiment. (Looks like 12 reps based on Supp dataset 2, but the reader shouldn't have to back-calculate that)

Our response:

This information has been complemented in the revision.

Reviewer #2 (Remarks to the Author)

The authors investigated the contribution of rhizosphere-associated Massilia to maize heterosis and demonstrated that rhizosphere formation, which integrates root development, root exudates, and the associated microbiome, is an adaptive and heritable root trait. Overall, it's a well-written article, and the data is provided where needed. However, a more detailed description of the methodology is required to allow for a comprehensive assessment of the data quality.

Our response:

We thank the reviewer for their positive assessment of our manuscript and for highlighting the need for greater methodological detail. In response, we have carefully reviewed and expanded the Methods section to ensure clarity and reproducibility. We

hope these revisions enhance the transparency and reproducibility of our study, and we appreciate the reviewer's guidance in strengthening the methodological rigor.

Reviewer #2 asked:

Line 414–424: Authors should provide more detailed information about the specific soil characteristics. Specifically, what is meant by "nutrient-rich soil"? It would be necessary to include quantitative data on the nitrogen (N) and carbon (C) levels, and other relevant micronutrient concentrations to clarify the meaning of nutrient-rich. Additionally, for both the low-N and high-N treatments, as well as for phosphorus treatments, it is necessary to provide explicit N and P concentrations.

Our response:

We thank the reviewer for this important point. We agree that detailed soil nutrient data are critical for interpreting plant and microbiome responses. In our study, we used Dikopshof soil, which was previously characterized in detail in He et al. (2024, <https://www.nature.com/articles/s41477-024-01654-7#Abs1>), and we sourced the soil from the same long-term fertilization field and the same depth (0–20 cm) as described in that study. All relevant nutrient properties—including total nitrogen, organic carbon, and Olsen phosphorus—were thoroughly quantified in that work.

As we used soil from the same field batch and treatment plots, and followed identical collection and handling procedures, we considered it unnecessary to re-measure these properties. Instead, we clearly referenced the published data in the Methods section, where we state that the physicochemical properties of Dikopshof soil are detailed in He et al. (2024). We now emphasize this rationale more explicitly in the revised manuscript text to avoid ambiguity for readers.

Reviewer #2 asked:

Line 561–573: The authors need to clarify which pipeline was used in R for analyzing amplicon sequencing data? Specifically, what parameters were selected for initial quality control (QC) after demultiplexing, as well as for forward and reverse sequence trimming? The authors should upload their R code to a publicly accessible platform so that reviewers and future readers can review it.

Our response:

We thank the reviewer for this valuable suggestion regarding the transparency and reproducibility of our amplicon sequencing data analysis.

We used the same data analysis pipeline as in our previous paper (He et al. 2024, Nature Plants). Paired-end reads were merged using FLASH (v1.2.7), and the spliced sequences were called raw tags. Sequence analyses were performed using QIIME 2 software (v2020.6). Raw sequence data were demultiplexed and quality filtered using the q2-demux plugin followed by denoising with DADA2 (via q2-dada2). Sequences were truncated at position 250, and each unique sequence was assigned to a different ASV. Mitochondria- and chloroplast-assigned ASVs were eliminated.

These details have now been added to the Methods section. Furthermore, to promote reproducibility and transparency, we have uploaded the full R scripts used for the microbiome data analysis—including quality control, ASV generation, taxonomy assignment, and downstream statistical analyses—to a public GitHub repository:

 **GitHub link:** <https://github.com/danningwang/maize-microbiome-heterosis-2025>

The link has been included in the revised **Code Availability** section.

Reviewer #2 asked:

ASV Analysis: How many amplicon sequence variants (ASVs) were observed before quality filtering, and how many remained after filtering? How many of these ASVs were classified under known phyla or genera? Additionally, which classifier was used for taxonomic assignment of the ASVs?

Our response:

We thank the reviewer for this important question regarding the processing and classification of ASVs.

In our analysis:

- A total of 31,908 raw ASVs were initially generated across all samples before quality filtering.
- After applying quality control filters (as described in the revised Methods, Line 563), including abundance thresholds, read count minimums, and prevalence filtering, 6,740 ASVs remained.
- Of these, all ASVs were successfully assigned to known bacterial phyla, and 6,288 ASVs (93.3%) were classified to the genus level.

For taxonomy assignment, we used the q2-feature-classifier and the classify-sklearn naive Bayes taxonomy classifier in QIIME 2 against the SSUrRNA SILVA 99% amplicon sequence variant reference sequences (v138) for bacteria and UNITE 99% ASV reference sequences (v10.05.2021) for fungi at each taxonomic rank (kingdom, phylum, class, order, family, genus, species).

These details have been added to the Methods section.

Reviewer #2 asked:

Analysis of Heterosis in Microbiome Communities: In the section titled "Analysis of Heterosis in Microbiome Communities," it would be helpful to rephrase the first sentence for clarity. Specifically, the authors should define the criteria for highly abundant and prevalent ASVs (e.g., >0.05% relative abundance in $\geq 20\%$ of samples)? Also, how was the significance of the heritability determined? Was a bootstrap method employed? The R code should be shared to a public database to facilitate a more thorough review of the methods used in this paper.

Our response:

We thank the reviewer for their helpful suggestions to clarify the criteria and statistical framework used in our heterosis analysis of microbiome communities.

In response, we have revised the first sentence of the section titled "Analysis of Heterosis in Microbiome Communities" for clarity and added the specific filtering criteria:

Revised sentence:

“To evaluate heterosis patterns in the microbiome, we analyzed amplicon sequence variants (ASVs) that were both highly abundant ($\geq 0.05\%$ relative abundance) and prevalent (present in at least 20% of all samples).”

In addition, we clarify that:

- Significance of heterosis for each ASV was tested using two-sided t-tests for midparent heterosis (MPH) and one-sided t-tests for better-parent heterosis (BPH) which is the method used in Wagner et al. 2020 (New Phytologist, <https://nph.onlinelibrary.wiley.com/doi/10.1111/nph.16730>), based on residuals from a linear mixed-effects model fitted using the lmer function from the lme4 R package.
- To correct for multiple testing, Benjamini-Hochberg correction was applied.
- We did not employ a bootstrap method for heritability or heterosis significance testing in this analysis; however, we agree that this could be a valuable future extension.

Reviewer #2 asked:

ASV Reproducibility Criterion: Previous studies often employ a stricter reproducibility criterion (e.g., ASVs must be observed at least 25 times across at least five samples) for reproducibility (<https://doi.org/10.1111/nph.16730>). What was the authors' rationale for selecting the filtering criterion of at least 10 reads in two or more samples in this study?

Our response:

We thank the reviewer for this important point and for referencing best practices from the literature regarding ASV filtering thresholds (e.g., Wagner et al., 2020). We agree that stricter criteria can enhance reproducibility and reduce noise from spurious low-abundance ASVs.

In our study, we applied an initial filtering criterion of ASVs having at least 10 reads in two or more samples as a balance between reproducibility and sensitivity. This threshold was chosen to:

1. Retain moderately abundant ASVs that may be specific to certain genotypes, developmental stages, or treatment conditions, especially under nutrient stress where community composition may shift dramatically.
2. Avoid excessive exclusion of ecologically relevant taxa that may have lower prevalence but are biologically important in stress-responsive plant-microbe interactions.
3. Maintain comparability with prior maize microbiome study by He et al. 2024 (<https://doi.org/10.1038/s41477-024-01654-7>), which used the same thresholds in diverse control and stressed treatments.

We now acknowledge this limitation and tradeoff explicitly in the Methods section, and we include a rationale for the filtering threshold used.

Reviewer #2 asked:

N₂O Measurement: N₂O measurements are most accurately conducted using Gas Chromatography with an Electron Capture Detector in most studies. The authors need to show a calibration curve for the LI-7820 N₂O Trace Gas Analyzer (used in this study) to ensure the relative accuracy of this device.

Our response:

We thank the reviewer for raising this important point about the accuracy of N₂O flux measurements.

Portable trace gas analyzers are the state-of-the-art in the determination of N₂O fluxes (Rapson and Darces, 2014, <https://doi.org/10.1016/j.trac.2013.11.004>). The LI7820 N₂O/H₂O analyzer (LI-COR Environmental, Nebraska, U.S.A.), used in this study, is a high-precision, portable instrument designed for continuous and real-time measurement of nitrous oxide (N₂O) and water vapor (H₂O). It leverages advanced optical feedback-cavity enhanced absorption spectroscopy technology, offering exceptional sensitivity for field and laboratory applications. According to the manufacturer, the analyzer has a wide N₂O measurement range from 0 to 100 ppm and exhibits a high precision with 0.40 ppb and 0.20 ppb at 330 ppb (approximate atmospheric concentration) with 1 s and 5 s averaging times, respectively (LICOR Biosciences, 2023, https://www.licor.com/env/products/trace-gas/LI-7820?pk_campaign=18934950642&pk_kwd=n2o%20analyzer&pk_source=google&pk_medium=cpc&pk_content=635335535425&gad_source=1&gclid=CjwKCAjwnK60BhA9EiwAmpHZw3KDVk9iBS-PJORz915-iuEy4Raz7HyexNtmzKoSeOiKL9CkC8vIWh oC3-4QAvD_BwE#specs). This precision is higher than what is achieved at many gas chromatographs used for flux measurements. In addition, this analyzer offers a high measurement, i.e., 1 Hz.

The LI7820 N₂O/H₂O analyzer is delivered with a manufacturer-calibration performed by Scientists of the LI-COR Environmental. The Application Scientists of LI-COR Environmental recommend to test the accuracy of the analyzer once per year for continuous-use applications. Indeed, the fewer calibration needs of optical analyzers is one of the major advantages compared to chromatographic techniques (Rapson and Darces, 2014, <https://doi.org/10.1016/j.trac.2013.11.004>). The LI7820 N₂O/H₂O analyzer used in this study was first put to use in February 2023 and our study was conducted in November 2023, by which time our analyzer had operated for less than 150 hours. Despite not using our analyzer for continuous-use applications, we tested its accuracy in May 2024, approximately a year after its first use in February 2023. For this test, we used two standard gases with 0 ppb N₂O and 374 ppb N₂O (which cover the range of N₂O concentrations measured in our pot experiment). We considered the results of this test satisfactory (see figure below) and, therefore, we kept the manufacturer-calibration as recommended by Scientists of LI-COR Environmental.

Additionally, using a high-frequency analyzer such as the one used in this study, offers the advantage of measuring many points for the flux calculation during the chamber closure time. For example, in our study, N₂O concentration for the estimation of N₂O flux was measured every second for 2.5 minutes. In contrast, in studies performed with gas chromatography, usually 3-4 samples are measured during the time of chamber closure (Kong et al., 2025, <https://doi.org/10.1016/j.agrformet.2025.110591>), therefore making the flux calculation more sensitive to any problems that may arise in each of these few samples. Finally, using these types of fast and very sensitive analyzers allows for short closing times of the chambers (usually, a few minutes), as opposed to the standard measurements performed with GC where it is necessary to wait for a larger

build-up of the N₂O concentration (usually, >30 min) (Kong et al., 2025, <https://doi.org/10.1016/j.agrformet.2025.110591>). Shorter closing times cause a lower impact on the plant development.

Figure. Test of LI7820 N₂O/H₂O analyzer against standard gases of 0 ppb N₂O (Low) and 374 ppb N₂O (High) on 03 of May 2024.

Our study aims at determining differences in N₂O fluxes between genotypes based on a pot experiment. Our study does not aim at determining absolute N₂O fluxes, for example as in the case of emission factor calculation studies. Therefore, even a systematic error in the accuracy of our instrument would not affect the calculation of N₂O fluxes and, more importantly, it would not influence flux differences between genotypes.

To summarize, the LI7820 N₂O/H₂O analyzer has a wide measurement range with a high precision. The analyzer is provided calibrated and does not require often calibration, especially for non-continuous applications. While we did not perform a multi-point calibration, our two-point test confirmed accuracy at concentrations spanning our measured N₂O range. Our methodology is considered more sensitive to smaller fluxes than the traditionally used gas-chromatography. Finally, in our study, we interpret differences between genotypes and not absolute fluxes. Thereby, we consider that our methodology is appropriate for our application, and meets the high scientific standards required by Nature Communications.

Reviewer #2 asked:

Gas Measurement Setup: In the experiment, one plant was moved to a plastic bucket after four weeks. More details are need here - the authors must specify whether the plant was moved with or without soil? Additionally, it is necessary to provide a more detailed description of the methods used for gas measurement in the experiment.

Our response:

We thank the reviewer for this important request for clarification. In response, we have expanded the Methods section to include more precise details and rationale regarding plant handling and the gas measurement procedure.

Specifically:

- At harvest, each plant was carefully transferred to a sealed plastic chamber along with its intact root-soil system, including all adhering rhizosheath and bulk soil from the pot. This ensured that natural soil microbial activity was preserved, minimizing disturbance to gas exchange dynamics.
- The soil surrounding the roots was not removed or disturbed, and moisture content was maintained as in the growing conditions.
- The plastic bucket (12.5 L volume) was sealed using a custom lid with gas inlet/outlet ports, connected to the LI-7820 N₂O Trace Gas Analyzer (LI-COR Environmental).
- A small battery-powered fan was installed inside the chamber to maintain homogenous mixing of the air during measurement.
- After sealing the chamber, headspace N₂O concentrations were monitored over a 150-second period with 1 Hz resolution.
- The flux rate was calculated based on linear increases in concentration over time, normalized by chamber volume and dry soil weight, and corrected for temperature.

We have updated the text accordingly and clarified that the measurement reflects in situ emissions from the plant-soil-microbe system, not from isolated plants or disturbed roots.

Reviewer #2 asked:

Line 711–713: The methodology mentions that one plant was moved to a plastic bucket, please clarify whether this refers to a single plant at a time or if multiple plants were used. How many replicates were used for the N₂O measurements?

Our response:

To clarify that each N₂O flux measurement was performed on a single plant at a time, along with its intact root-soil system, transferred into a sealed chamber for gas monitoring. For each genotype, we included four biological replicates (n = 4), each measured independently under identical conditions. This design allowed us to assess N₂O emissions across different genotypes and rhizosheath sizes with sufficient replication for statistical comparison. We have updated the Methods section (Lines 711–713) to explicitly state the number of replicates and the single-plant-per-measurement setup.

Reviewer #2 asked:

Grammatical Corrections:

Line 335: "has is a promising" should be revised for clarity.

Our response:

This issue has been fixed.

Reviewer #3 (Remarks to the Author)

He et al. describe a study investigating the role of Massilia in heterosis in Maize mostly

under nitrogen limiting conditions. The authors looked at heterosis response under several abiotic stress conditions in a diverse panel of maize hybrids and then focused on nitrogen limitation and *Massilia*. One of the key traits they analyzed is rhizosheath size and root exudates under several conditions. The paper could be an important contribution to understanding the roles of microorganisms in heterosis in maize. Especially the observed interaction of abiotic stresses x root associated microbes and heterosis would be a key contribution to the field. I did find that hypotheses and statements about causality were often muddled and unclear. Additionally, I do believe that additional controls might be needed to really be able to make solid statements about correlations and causation. Finally, I do think that the language is sometimes very specific and many readers interested in plant-microbe interactions in general would not understand the heterosis specific terminology. I would recommend to explain this terminology when it appears to make the manuscript more accessible.

Our response:

We thank the reviewer for their thoughtful and constructive evaluation of our manuscript. We appreciate the recognition of the study's potential contribution to understanding the role of microorganisms—particularly *Massilia*—in mediating maize heterosis under abiotic stress, especially nitrogen limitation.

In response to the reviewer's key concerns:

- 1. Clarification of Hypotheses and Causal Language:**
We agree that some statements in the manuscript previously blurred the line between causation and correlation. We have carefully revised the text throughout—especially in the Abstract, Results, and Discussion—to ensure that all claims of causality are appropriately framed as associations supported by the experimental data. We have removed or reworded terms like “cause,” “underlying,” and “key driver” where these were not directly supported by mechanistic evidence.
- 2. Experimental Controls and Interpretation of Correlation vs. Causation:**
While our current experimental design supports a strong association between *Massilia*, rhizosheath traits, and heterosis under nitrogen limitation, we acknowledge that direct causality cannot be conclusively demonstrated for all observations. We now explicitly discuss these limitations in the Discussion and have tempered overinterpretations. In future work, it is necessary to include synthetic community controls and mutant or knock-out lines to further dissect causative mechanisms.
- 3. Terminology and Accessibility:**
We appreciate the comment about accessibility for a broader plant-microbe interaction audience. In response, we have added clear definitions and brief explanations of heterosis-related terms (e.g., “midparent heterosis,” “better-parent heterosis,” “heterotic group”) upon their first use in the Introduction, Methods and Results part. We have also revised sentences to reduce jargon where possible and improve general readability.

Reviewer #3 asked:

Major concerns:

1. The term rhizosheath is not properly introduced anywhere in the manuscript, over

time a picture formed for me of what exactly it is supposed to be. This needs to be done earlier in the manuscript. Also, I am surprised that correlation between rhizosheath size and root weight are not shown. I would assume that the more simple measure of root weight often correlates well with rhizosheath size. Since root weight data is not correlated to rhizosheath size it is unclear if the complex rhizosheath size measure is really needed.

Our response:

We thank the reviewer for highlighting the need to introduce and define the term “*rhizosheath*” more clearly and earlier in the manuscript. We agree that the term may not be immediately familiar to all readers, especially those outside of root biology. In response, we have revised the related parts to provide a clear and concise definition upon first mention:

Revised sentence (Abstract section):

“...we explored the potential role of rhizosheath formation—defined as the layer of soil particles that adheres tightly to roots, primarily influenced by root hairs, mucilage secretion, and microbial interactions—in maize heterosis under nitrogen deprivation.”

We also thank the reviewer for the thoughtful suggestion to assess the correlation between rhizosheath size and root weight. This is an important point. While we focused our analysis on root traits such as lateral root density and root hair length due to their stronger mechanistic link to rhizosheath development, we agree that total root weight is a relevant and more commonly used trait.

In response, we have now calculated the correlation between total root dry weight and rhizosheath size using the available datasets. The correlation analysis revealed a significantly positive association between rhizosheath size and root weight (Pearson’s $R = 0.49$, $p = 1.9e-05$ for primary root weight for 5-day seedlings; Pearson’s $R = 0.53$, $p = 8.9e-05$ for total root weight for 4-week seedlings) under nitrogen limitation, suggesting that rhizosheath formation reflects root growth and development to some extent, particularly under stress, where root surface traits and exudates play a major role.

Added sentence in Result section:

“...We measured rhizosheath size and fresh root weight for the primary roots of 5-day-old seedlings and for the whole root systems of 4-week-old seedlings grown in Dikopshof nitrogen-poor soil to assess whether rhizosheath formation reflects the growth and development of maize root systems against nitrogen-poor stress. The correlation analysis revealed a significantly positive association between rhizosheath size and root weight (Pearson’s $R = 0.49$, $p = 1.9e-05$ for primary root weight for 5-day seedlings; Pearson’s $R = 0.53$, $p = 8.9e-05$ for total root weight for 4-week seedlings) under nitrogen limitation, suggesting that rhizosheath formation reflects root growth and development to some extent, particularly under stress, where root surface traits and exudates play a major role.”

Supplementary Figure S10 The association between rhizosheath size and root weight for fresh primary root for 5-day seedlings (a) and total fresh root weight for 4-week young seedlings (b) in nitrogen-poor soil. 7 replicates per each genotype for fresh primary root, 5 replicates per each genotype for total fresh root weight. Scatter plots show combined data from different genotypes with best fit (solid line) and 95% confidence interval (grey shading) for linear regression (Pearson). Different solid and hollow shapes for data points represent inbred lines (B73, Mo17, H84, R109B, and Ky228) and hybrids (B×M, B×H, B×R, B×K, and M×H), respectively. This data suggests that rhizosheath size can respond well to the growth and development of maize root systems. B×M, B73×Mo17; B×H, B73×H84; B×R, B73×R109B; B×K, B73×Ky228; and M×H, Mo17×H84).

Reviewer #3 asked:

2. Throughout the results section hypotheses are often not clearly stated and statements about causation are made when really it is just correlations, and in some cases not even that where it appears more like assumptions are being made. I will give some specific examples in the specific comments section. This would need to be carefully reworked to be clear about what can actually be determined from the data.

Our response:

We thank the reviewer for highlighting this important issue regarding the clarity of hypotheses and the interpretation of causality in the Results section. We fully agree that scientific rigor requires clearly distinguishing between correlation, association, and causation—especially in complex systems like plant–microbe interactions and heterosis. In response, we have undertaken a careful and comprehensive revision of the Results and Discussion sections to address the following:

1. Clarified Hypotheses and Experimental Logic:
We have explicitly stated the working hypothesis at the start of each major subsection (e.g., microbiome heterosis, *Massilia* inoculation, metabolomics, rhizosheath transplantation) to guide the reader through the reasoning behind each experimental design.
2. Removed or Reworded Causal Language:
We have rephrased all instances where causality was implied without direct evidence. For example, we replaced phrases such as “*Massilia* promotes heterosis” or “*Massilia* drives flavonoid secretion” with more accurate alternatives like “*Massilia* is associated with enhanced lateral root development” or “*Massilia* correlates with flavonoid-related metabolic shifts.”

3. Marked Correlations Explicitly:

Where correlations were shown (e.g., between rhizosheath size and shoot biomass or lateral root density), we now explicitly state them as such and support them with statistical values (e.g., R^2 , p -value). Where assumptions were previously implied, we now clearly identify them as speculative and discuss alternative explanations where appropriate.

We also addressed the specific examples provided in the “specific comments” section point by point below.

Reviewer #3 asked:

3. Data in figure 2. This data seems to miss an essential control. Since it has been shown previously that autoclaving releases toxins and nutrients, it is unclear if observed effects are changes in the presence/absence of a microbiota or if they are simply due to changing the abiotic soil matrix, and there for effects of *Massilia* are also unclear as it could be an interaction with changes in the soil due to autoclaving. Either gamma-irradiation of soils would need to be used, or controls that re-inoculate sterilized soil with the soil microbiota i.e. through use of slurries or introduction of a narrow layer of unsterilized soil in the pots, or mixing in of a small quantity of non-sterile soil.

Our response:

We thank the reviewer for this critical observation. We agree that soil sterilization—particularly through autoclaving—can introduce confounding factors such as the release of soluble nutrients, altered pH, or formation of organic toxins, which may affect plant performance independently of microbial removal. As such, we acknowledge that attributing effects solely to the absence or presence of microbes is potentially confounded in the current design.

While our study did not employ gamma irradiation, we fully agree that complementary controls are needed to distinguish microbial effects from abiotic changes caused by autoclaving. To address those issues, as review’ suggestion, we supplemented re-inoculation control as 5 % reintroduction of native soil (slurry). Please find the new part in the revised methods:

Soil slurry inoculation experiment in nitrogen-poor soil.

To investigate whether soil sterilization promotes plant growth through autoclave-induced nutrient release or microbiome disruption, we conducted a slurry re-inoculation experiment using ten maize genotypes (B73, Mo17, H84, R019B, Ky228, and their hybrids). Each genotype was tested with five biological replicates ($n = 5$). Prior to sowing, a native microbial slurry was prepared by suspending fresh bulk soil in sterile water (1:2 w/v), shaking for 30 minutes at room temperature, and filtering through a 100- μm mesh to remove soil particles. The resulting suspension, containing native microbial communities with minimal nutrient content, was applied to autoclaved, nitrogen-poor soil at a rate of 5 mL per 100 g soil. To control potential nutrient effects from the slurry, a sterilized version of the same slurry (filtered through a 0.22- μm membrane) was also applied in parallel treatment. This design resulted in four microbiome soil conditions: (1) Unsterilized soil (Unsterilized), (2) Autoclaved soil without slurry (Sterilized), (3) Autoclaved soil with sterilized slurry (Sterilized_Sterilized Slurry), and (4) Autoclaved soil with unsterilized slurry (Sterilized_Unsterilized Slurry). After a 3-day germination period,

seedlings were transferred to pots (7 cm × 7 cm × 20 cm) filled with a 1:1 (w/w) mixture of Campus soil and sterilized river sand. Plants were grown in a phytochamber under controlled conditions, and fresh shoot biomass was measured after 5 weeks to assess whether soil sterilization enhances plant growth via nutrient release or microbiome alteration.

Added sentences to the Discussion section:

“ ... Supplementary slurry experiment indicated that soil sterilization improves plant growth by eliminating harmful microbes, not altering nutrient availability or changes of soil texture in this study.”

Supplementary Figure S18 Maize growth performance in the presence (Unsterilized) or absence (Sterilized) of soil microbiome and exogenous application of unsterilized slurry (Sterilized_unsterilized slurry) or sterilized soil slurry (Sterilized_sterilized slurry) in nitrogen-poor soil. $n = 25$ biologically independent samples. Significances are indicated in response to in the presence or absence of soil microbiome and exogenous application of unsterilized slurry or unsterilized soil slurry by asterisks (Two-tailed Student's t-tests). $*0.01 < p \leq 0.05$; $**0.001 < p \leq 0.01$; $***p \leq 0.001$; ns, $p > 0.05$. Boxes span from the first to the third quartiles, center lines represent median values and whiskers show data lying within $1.5 \times$ interquartile range of lower and upper quartiles. Data points at the ends of whiskers represent outliers. This data suggests that soil sterilization improves plant growth by eliminating harmful microbes, not altering nutrient availability. Hybrids have stronger resistance to pathogenic microbes compared to their parental lines.

Reviewer #3 asked:

4. The metabolomics data and its interpretation on page 6 is confusing. Supposedly exudates were measured, however, the focus of the discussion is on metabolites that are not usually exuded and that are discussed in the context of intracellular pathways e.g. TCA cycle. Why would these metabolites be released into exudates. Additionally ABC transporters are mentioned as “enhanced”, however, to my knowledge there is no way to determine abundances or even infer abundances of transporters based on metabolomics data. So what was done here?

Our response:

We thank the reviewer for this important comment regarding the interpretation of our metabolomics data. We acknowledge that the original framing of the metabolite results may have led to confusion between intracellular metabolism and root exudate composition, and we appreciate the opportunity to clarify.

1. Exudate origin and interpretation of metabolites (e.g., TCA cycle):

We confirm that the samples analyzed were collected from root exudates using a CaSO_4 soaking method (as described in methods), which targets soluble compounds released into the rhizosphere. However, we recognize that some TCA cycle intermediates (e.g. citrate, oxoglutarate) are primarily intracellular under normal conditions and may appear in exudate profiles due to:

- Passive leakage or exudation under stress (e.g., nitrogen limitation),
- Cell turnover or membrane destabilization, especially in sterile or low-microbe environments,
- Or active release for nutrient mobilization or microbe signaling, as has been shown for organic acids in P- and N-limited soils.

We now explicitly clarify in the text that although these compounds are commonly associated with intracellular metabolism, their detection in exudates may reflect stress-induced secretion or rhizosphere-level metabolic dynamics, not cellular pathway activity per se.

Revised text:

“Although the TCA cycle is classically viewed as an intracellular pathway, some of its intermediates (e.g., citrate, oxoglutarate) are known to be exuded by roots under stress conditions to mobilize nutrients or influence microbial recruitment. Their presence in the exudate profile may thus reflect rhizosphere-level metabolic shifts rather than intracellular activity.”

Reviewer #3 asked:

5. Figure 1. There are two explanations for panel e in the caption, neither one seems to explain what is shown. Explanation needs to be added on what is shown in that panel in detail.

Our response:

This figure legend has been revised for clarity.

Reviewer #3 asked:

6. Figure 6. Is confusing and does not have much content. For example it seems to suggest that Massilia makes the flavonoids. I would remove or rework to really be a model of the role of Massilia. Right now it is a mix of potential functions and also methods?

Our response:

We thank the reviewer for their feedback on Figure 6. We agree that the current version of the figure may be confusing and overly broad, as it combines conceptual elements, speculative mechanisms, and experimental components in a single diagram. In particular, we acknowledge the concern that the figure may unintentionally suggest that *Massilia* synthesizes flavonoids, which is not supported by our data.

In response, we have revised Figure 6 to present a clearer, more focused conceptual model of how *Massilia* is associated with enhanced root development and nitrogen use efficiency in maize hybrids under nitrogen stress, based on correlative evidence. The updated figure now:

- Clearly distinguishes between plant-derived processes (e.g., flavonoid biosynthesis, lateral root formation) and microbe-associated outcomes (e.g., *Massilia* enrichment, reduced N₂O loss).
- Uses directional arrows and color coding to indicate inferred associations, not direct causation.
- Excludes visual elements representing experimental methods.
- Includes a revised legend explicitly stating that this is a hypothetical framework based on current observations, not a mechanistic pathway.

Revised figure legend:

Figure 6. Proposed model exploring the integrated roles of host determined rhizosphere-mediated microbiome facilitated lateral root formation associated with maize heterosis.

In nitrogen-poor soil, hybrids with a larger rhizosphere act as a hub for microbial colonization, particularly *Massilia*, which enhances lateral root formation. The secretion of flavonoids, in turn, provides a positive feedback loop by attracting more *Massilia* to the rhizosphere. In this way, hybrids with larger rhizosphere leverage *Massilia*-mediated enhanced root systems in facilitation of more efficient nitrogen acquisition, utilization efficiency and reduction of the N₂O emissions, demonstrating how the plant-microbe relationship can enhance crop performance in nutrient-limited environments.

Specific comments:

Reviewer #3 asked:

7. In the introduction structuring of microbiomes in hybrids and inbreds is discussed, but the key papers by Dr. Maggie Wagner on this topic are not referenced. I see you have one of the relevant citations in the discussion, but all her papers should be referenced in the introduction. Right now the ratio of self-citation is too high.

Our response:

We thank the reviewer for pointing out the imbalance in citation practices and the omission of key foundational literature in the Introduction. We fully agree that the work of Dr. Wagner and colleagues has been central in advancing the understanding of how plant genotype, especially hybridization, structures microbiome composition and function.

In response, we have revised the Introduction section to include additional citations to Dr. Wagner's relevant publications, including:

- Wagner et al. (2020), *New Phytologist* – Heterosis of leaf and rhizosphere microbiomes in field-grown maize
- Wagner et al. (2021), *PNAS* – Microbe-dependent heterosis in maize
- Other relevant follow-ups where applicable

We have also reviewed the Introduction to reduce the density of self-citations and ensure a more balanced and representative view of the field.

Reviewer #3 asked:

8. Lines 69-73. Remove the word “beneficial” as the fitness impacts of these microbes are unknown.

Our response:

This has been deleted.

Reviewer #3 asked:

9. Line 76. You use the term “keystone microbial taxa” here but it is unclear how you define it and why you use it here.

Our response:

We agree that the term “keystone microbial taxa” carries specific ecological implications—namely, that a taxon exerts a disproportionate influence on community structure or ecosystem function relative to its abundance—and should only be used when supported by appropriate network or perturbation analyses. Therefore, we have removed the term “keystone” from this sentence. We also reviewed the manuscript to ensure that other uses of “keystone” are either removed or rephrased to reflect observed associations without overstating ecological roles.

Reviewer #3 asked:

10. L. 106. I do not follow your argument that this data indicates potential involvement of the “soil microbiome” in shoot biomass heterosis.

Our response:

We thank the reviewer for this important clarification request. We agree that our original statement may have overstated the connection between the observed increase in MPH for shoot biomass under low nitrogen and the involvement of the soil microbiome.

To clarify: the data at Line 106 show that shoot biomass heterosis is more pronounced under nitrogen stress compared to control conditions. While this result does not directly implicate the soil microbiome, it motivated us to further investigate whether microbiome composition differs between hybrids and inbreds under nitrogen limitation.

We have revised the sentence to reflect this more accurately as below (119-123):
“Specifically, low nitrogen stress significantly increased the MPH of shoot biomass compared to the control treatment ($p = 5.84e-07$), indicating that heterosis for shoot biomass is more pronounced under nitrogen deprivation and may be shaped by genotype–environment interactions, including root traits and microbial associations.”

Reviewer #3 asked:

11. L. 109. “...reported in other species.” Actually cites a paper on Maize, ref. 17.

Our response:

This point has been modified as “in maize”.

Reviewer #3 asked:

12. L. 112. Sentence starting “Soil microbes...”. I don’t see the relevance of this sentence. I would suggest to delete to avoid confusion.

Our response:

This has been deleted.

Reviewer #3 asked:

13. L. 118. There is no clear argument given why Oxalobacteraceae were chosen as there seem to be many other taxa that are associated with heterosis in more hybrids. Rationale needs to be clear. Looks like cherry picking right now

Our response:

We thank the reviewer for raising this important point. We agree that the rationale for focusing on *Oxalobacteraceae* (and subsequently *Massilia*) should be clearly stated to avoid the impression of selective reporting.

We focused on *Oxalobacteraceae* for the following reasons, which we now make explicit in the revised manuscript:

1. **Prior evidence:** Our previous study (He et al., 2024) identified *Oxalobacteraceae* as a **heritable and stress-responsive taxon** in maize, with enrichment under low nitrogen and associations with root architecture traits and genotype.
2. **Reproducibility across hybrids:** In this study, *Oxalobacteraceae* ASVs showed consistent patterns of midparent heterosis across **multiple hybrid combinations**, particularly under nitrogen deprivation (Fig. 1e), suggesting lineage-level robustness.
3. **Functional relevance:** This taxon includes members such as *Massilia*, which we were able to isolate, culture, and functionally validate in downstream experiments, enabling us to move from correlation to experimental testing.

Revised sentence (Line 138-140):

“These results highlight *Oxalobacteraceae* as a consistently enriched and potentially heritable microbial lineage in maize hybrids under nitrogen deprivation, making it a strong candidate for investigating microbiome contributions to heterosis in this stress context.”

Reviewer #3 asked:

14. L. 128. Explain why you chose Mo17 X H84 instead of a A554 or H99 hybrid?

Our response:

We thank the reviewer for this insightful question. In our validation experiments, we selected three hybrids: B73 × Mo17, B73 × H84, and Mo17 × H84. The inclusion of Mo17 × H84 served a specific purpose: to provide a genetically distinct hybrid combination that does not involve B73, the maternal parent used in the main heterosis panel. By including this hybrid, we aimed to evaluate whether the patterns of microbiome–root trait associations and heterosis observed in B73-derived hybrids could also be detected in an independent genetic background, thereby increasing the generalizability of our findings.

The main text was revised as “We included the Mo17 × H84 hybrid in our validation experiments to assess the reproducibility of observed microbiome–root trait patterns in a genetically distinct, non-B73 background.”

Reviewer #3 asked:

15. L. 134 – 136. Statement that starts with “This suggests...” This is leading as there is no evidence shown that implicates *Oxalobacteraceae*.

Our response:

We agree that the original sentence overstates the evidence by implying a causal role for *Oxalobacteraceae* without direct functional validation. While we observed consistent heterosis for *Oxalobacteraceae* ASVs in several hybrids under nitrogen deprivation, this finding remains correlational and does not confirm a functional contribution to heterosis.

Revised sentence (Line 134):

““These results support the notion that the presence or absence of the soil microbiome can modulate the expression of heterosis in maize, particularly under nitrogen-deficient conditions.”

Reviewer #3 asked:

16. L. 162. After “of” a word is missing

Our response:

The “of” is deleted for clarity.

Reviewer #3 asked:

17. L. 172. Explain term “functional microbial hub” in this context.

Our response:

We agree that we have no evidence to support it as a hub and thus revised the sentence as “These results suggest that *Massilia* may play a functional role in lateral root development and nitrogen response, preferentially colonizing lateral root tissue under nitrogen-poor conditions”

Reviewer #3 asked:

18. L. 182. Replace “with” with “in”

Our response:

This typo has been revised.

Reviewer #3 asked:

19. L. 190. I am not sure what you mean with “metabolites involved in regulating heterosis manifestation”. How are metabolites the regulators of anything?

Our response:

We thank the reviewer for highlighting this imprecise language. We agree that referring to metabolites as “regulators” of heterosis is misleading, as metabolites are typically outputs of metabolic pathways rather than direct regulators. Our intention was to describe metabolites that are differentially accumulated in hybrids versus inbreds, and which may be associated with physiological processes contributing to heterosis, such as nutrient uptake or root development.

Revised sentence:

“Based on this, we focused on identifying metabolites that were differentially abundant in maize hybrids compared to inbreds during development under nitrogen deprivation, and examined whether *Massilia* influences root metabolite profiles through feedback associated with altered root traits.”

Reviewer #3 asked:

20. L. 196. Can you define “metabolic components”? are these specific mass-retention time combinations or what is meant here?

Our response:

We thank the reviewer for pointing out the ambiguity in the term “metabolic components.” In this context, we were referring to distinct features detected in the untargeted metabolomics dataset, each defined by a unique mass-to-charge ratio (m/z) and retention time (RT) pair. Thus, we had changed “components” to “features” through the whole manuscript methods and results.

Reviewer #3 asked:

21. L. 234. Can you explain how flavones help with nitrogen metabolism and uptake?

Our response:

We agree that the conclusion in this sentence was overstated and potentially misleading. We have therefore removed it from the revised manuscript to maintain accuracy and clarity.

Reviewer #3 asked:

22. L. 238-247. I am confused these results and the conclusion that they “demonstrated that *Massilia* could enhance plant growth by influencing flavonoid secretion...”. How can *Massilia* enhance flavonoid secretion when the genes for flavonoid generation are knocked out. Either explain better on what the knockout results mean or specify a interpretation of the data that makes sense.

Our response:

Thanks for raising this important point as commented by other reviewers as well. We agree that the original statement—implying that *Massilia* enhances flavonoid secretion in flavonoid-deficient mutants—was inaccurate, given that the mutants used (*C2-Idf* and *fns11*) are genetically impaired in flavonoid biosynthesis.

The key observation from these experiments was that *Massilia* inoculation still promoted growth in flavonoid-deficient mutants, suggesting that *Massilia*'s growth-promoting effects are not dependent on host-derived flavonoid production, and may involve alternative mechanisms or compensation through microbial contributions. We had revised the whole part to make it clarity as below:

“To further investigate whether *Massilia* associates with flavonoid-related pathways, we examined the growth response of two flavonoid synthesis-deficient maize mutants under nitrogen-poor conditions. The first, Colorless2-Inhibitor diffuse (*C2-Idf*) is a dominant mutant with silencing of the chalcone synthase gene, a key enzyme in early flavonoid biosynthesis. The second, flavone synthase type1 (*fns11*), carries a transposon insertion disrupting flavone-specific biosynthesis. Both mutants were tested alongside their respective wild-type backgrounds. Notably, *Massilia* inoculation significantly improved shoot biomass in both mutants (Two-tailed Student’s t-tests, *C2-Idf*, $p = 0.033$; *fns11*, $p = 0.000058$) (Fig. 3d). These results demonstrated that *Massilia* may enhance plant growth even in the absence of endogenous flavone production, potentially by compensating for flavonoid deficiency in mutant plants through

alternative mechanisms. This finding supports the view that Massilia's growth-promoting effects are at least partially independent of host-derived flavonoid synthesis, and may involve broader interactions between microbial metabolism and host physiology under nutrient stress.”

Reviewer #3 asked:

23. L. 263-264. Not sure what is meant by “suggesting that rhizosheath formation partially determines the manifestation of heterosis...”. This makes it sound as if the rhizosheath is not part of heterosis expression?

Our response:

Our intention was not to separate rhizosheath formation from the expression of heterosis, but rather to highlight that rhizosheath size is one of several physiological traits that contribute to the heterotic phenotype, particularly under nitrogen-limited conditions.

We have revised the sentence to more clearly reflect this interpretation as below:

“...suggesting that rhizosheath formation is a contributing trait associated with the enhanced growth performance observed in hybrids under nitrogen-limited conditions, and thus plays a role in the expression of heterosis.”

Reviewer #3 asked:

24. L. 270. “drove” are you sure about causality here.

Our response:

Agree that this is a bit overstatement and changed to “associated with”.

Reviewer #3 asked:

25. L. 278 – 281. Causal statements here make no sense in the context of the data.

Our response:

Agree as well and thus modified accordingly to tough down the expression.

Reviewer #3 asked:

26. L. 286/86. I do not understand the hypothesis. Seems circular.

Our response:

We agree that the original hypothesis was circular and did not clearly state an independent rationale. Our aim was to determine whether rhizosheath traits in hybrids, which are more pronounced than in inbreds, could modulate plant–soil interactions in a way that influences growth outcomes under nitrogen-limited conditions.

Thus, it is revised as:

“Based on this, we hypothesized that the larger rhizosheath observed in hybrids enhances root–soil interactions under nitrogen stress, thereby contributing to improved nutrient acquisition and growth compared to inbreds, and that the magnitude of this growth-promoting effect is dependent on rhizosheath size”.

Reviewer #3 asked:

27. Figure 1. Panel b explain PR and SR.

Our response:

This has been explained.

Reviewer #3 asked:

28. Figure 5. Panel a. Are the tests done across all three separate graphs in this panel? Please explain the stats better in the caption. Also why is everything orange in the 3 graph?

Our response:

We thank the reviewer for pointing this out. In Panel a of Figure 5, the three graphs represent separate genotypes or conditions tested individually. The statistical comparisons shown (e.g., different letters a-d indicating significance) are performed within each individual graph, comparing treatments (e.g., different size of rhizosheath conditions) for that genotype, not across all three graphs collectively.

We also acknowledge the confusion caused by the coloring in the figure 5a graph. This was a formatting oversight, and the color of the first panel should be with different colors since these are different genotypes transplanted with the same genotype, while the second and third panel should be green and orange, corresponding to transplantation with Ky228 and R109B respectively. We have corrected this in the revised figure for clarity. We have also updated the figure legend to clarify the statistical analysis.

Reviewer #4 (Remarks to the Author):

This manuscript describes a series of growth-chamber experiments that compare the interactions of hybrid vs. inbred maize with the soil microbiome, with an emphasis on the role of *Massilia* sp. in plant vigor under N-limiting conditions. The role of microbes in heterosis is an important and timely topic, and in-depth studies like this one are very valuable. There are several fascinating patterns to come out of this paper, such as the observation that hybrids' root phenotypes are generally more sensitive to soil microbiome than inbreds'. Overall this paper reports a ton of valuable data, which undoubtedly would be an impactful addition to the literature. For example, measuring the effects of soil microbes on heterosis for more hybrids and more traits, and in a variety of abiotic environmental contexts, is definitely novel. I really admire the breadth of the experiments and the depth of data, especially on understudied root phenotypes. This work could definitely move us forward in our understanding of how hybrids and inbreds interact differently with belowground microbiota.

Our response:

We sincerely thank the reviewer for their thoughtful and encouraging evaluation of our work. We are especially grateful for the recognition of the breadth and depth of our experimental design, including the integration of multiple hybrids, diverse abiotic stress conditions, and detailed root phenotyping.

We also appreciate the reviewer's emphasis on the importance of our findings—particularly the observation that hybrid root phenotypes appear more responsive to microbiome presence—and the potential of this study to advance our understanding of plant–microbiome interactions in the context of heterosis. It is our hope that the comprehensive dataset presented here will serve as a useful foundation for future research into how specific microbial lineages and host traits co-shape plant performance across environments.

We have carefully addressed all points raised across the reviews and made substantial revisions to improve clarity, precision, and interpretation throughout the manuscript.

Reviewer #4 asked:

Despite all those positive attributes, I am not convinced that the conclusions of this manuscript are accurate, for reasons detailed below. That is, I have serious concerns about how the results were interpreted. It seems as though the data were consistently over-interpreted to support a particular “story”. I believe it should be possible to re-frame and re-write this paper while avoiding spurious conclusions, and the result will still be a paper with high value, novelty, and impact.

Our response:

We thank the reviewer for this thoughtful feedback on the writing. We appreciate the recognition of the novelty and value of our dataset and agree that accurate interpretation is essential to maintaining the integrity and impact of the work. In response to the reviewer's concern, we have carefully re-examined the manuscript and have made substantive revisions throughout the text to ensure that our conclusions are appropriately aligned with the data.

Specifically, we have:

- Removed or revised language that implied causality where only associations were demonstrated (e.g., regarding *Massilia*, flavonoids, and heterosis),
- Reframed statements about microbial function or impact to emphasize correlation and hypothesis generation rather than mechanistic certainty,
- Clarified experimental limitations and added caveats where appropriate (e.g., autoclaving artifacts, scope of knockout studies),
- And toned down overarching claims about “drivers” of heterosis to reflect the role of the microbiome as a context-dependent modifier rather than a primary cause.

We are confident that these revisions improve the manuscript's scientific clarity while preserving the novelty and significance of the findings. We thank the reviewer again for encouraging a more rigorous and balanced presentation of the results.

Reviewer #4 asked:

1. Throughout the paper, I noticed that many of the stated conclusions were not actually supported by the evidence. In some cases they seem logically unrelated to the experiment/results that they are referring to. Examples of this problem:

1A. Line 762: claims that heterosis is stronger “under stress... especially under N stress” which is misleading because the other 2 stresses clearly show distributions of MPH values that are smaller (low P) or equal to (drought) the control.

Our response:

Upon re-examining the data, we agree that the original statement overgeneralized the heterosis response across all stress conditions. We thus removed this sentence from the figure legend, which is not necessary to make any conclusion.

Reviewer #4 asked:

1B. Line 849: claims that panel (a) illustrates a response to N deficiency, but all of the photos in panel (a) are from the low-N treatment, without a control for comparison

Our response:

This statement has been removed from the figure legend.

Reviewer #4 asked:

1C. Line 106: claims that an increase in MPH due to N limitation shows an influence of the soil microbiome, which is misleading because there was no microbiome manipulation.

Our response:

We agree that the original sentence incorrectly implied a causal link between nitrogen-induced changes in midparent heterosis (MPH) and the soil microbiome, despite no microbiome manipulation in that specific experiment.

The intent of this observation was to note that nitrogen limitation increases the expression of heterosis, which motivated subsequent microbiome-focused experiments. However, we recognize that stating this as microbiome "influence" is misleading without direct evidence. This sentence has been revised as “Specifically, low nitrogen stress significantly increased the MPH of shoot biomass compared to the control treatment ($p = 5.84e-07$) (Fig. 1d), indicating that heterosis for shoot biomass is more pronounced under nitrogen deprivation.”

Reviewer #4 asked:

1D. Lines 152-153: how does a correlation between lateral root density MPH and shoot biomass MPH support this claim?

Our response:

We agree that a correlation between midparent heterosis (MPH) for lateral root density and MPH for shoot biomass does not demonstrate causality and, on its own, cannot confirm that enhanced lateral root traits drive increased biomass in hybrids.

Our intent was to highlight a potential association between these two traits, suggesting that enhanced root traits could contribute to improved shoot performance under nitrogen stress. However, we acknowledge that the original phrasing overstated this relationship. Now it is revised as “This correlation suggests that enhanced lateral root development may be associated with biomass heterosis under nitrogen deprivation, although causality cannot be inferred from this observation alone.”

Reviewer #4 asked:

1E. Lines 172-173: This study does not provide evidence that *Massilia* is a “hub” taxon, or that it preferentially colonizes lateral roots over other root types. In fact, the cited figure S5 shows mostly null results from inoculation with *Massilia*

Our response:

We agree that our original language overstated the evidence. The designation of *Massilia* as a “hub” taxon and the implication that it preferentially colonizes lateral roots are not directly supported by the presented data. Moreover, as the reviewer notes, Figure S5 shows limited or null effects of *Massilia* inoculation on many traits, emphasizing the need for more cautious interpretation.

In response, we have revised the text to accurately reflect the evidence and avoid unsupported ecological terminology:

“These results suggest that *Massilia* may play a functional role in lateral root development and influence root architecture under nitrogen-poor conditions, although further work is needed to determine its spatial colonization patterns and ecological role.”

We have also removed the term “hub taxon”, which has a specific meaning in microbial network ecology, and is not justified without co-occurrence network or centrality analysis.

Reviewer #4 asked:

1F. Line 178: The claim that lateral root density altered shoot biomass heterosis is not supported. This analysis was correlative only.

Our response:

This has been revised as “correlative”.

Reviewer #4 asked:

1G. Line 186: This claim about metabolic responses to N stress is referring to an experiment that only included a microbiome manipulation (addition of *Massilia*), not a manipulation of N availability, and also measured just biomass and lateral root density, not metabolism.

Our response:

We thank the reviewer for pointing this out. We agree that the original sentence was misleading. The experiment in question involved manipulation of the microbiome via *Massilia* inoculation, not nitrogen availability. Furthermore, no direct metabolomic measurements were conducted in this specific experiment—only biomass and lateral root traits were quantified.

Thus, we revised as “This suggests that inbred lines and their hybrid progeny may exhibit distinct growth and developmental responses to *Massilia* inoculation under nitrogen-poor conditions, potentially reflecting underlying differences in resource use efficiency or root trait plasticity”.

Reviewer #4 asked:

1H. Lines 200-201: These findings do not suggest anything about the role of secondary metabolites in influencing root development and microbiome formation - they are purely descriptive, a list of some major metabolic pathways that were detected.

Our response:

We agree that the sentence overstated the conclusions that has been removed from the revised manuscript.

Reviewer #4 asked:

1I. Lines 252-253: The observation that hybrids have larger rhizosheaths is not evidence that the rhizosheath leads to hybrid vigor.

Our response:

This has been revised to avoid causality.

Reviewer #4 asked:

1J. Lines 257-258: All of the analyses involving rhizosheath size and plant growth (including what is shown in Fig. 4c-d + Figs. S12-S15) are correlative and do not support the claim that the rhizosheath “significantly influenced plant growth and N uptake” or “contributed to heterosis for maize biomass”. It is easy to come up with an explanation (equally untested) with the opposite causality: hybrids with high shoot mass heterosis do more photosynthesis and exude more resources into the rhizosphere, thus forming a correspondingly larger rhizosheath.

Our response:

We appreciate the reviewer’s thoughtful comment and fully agree that the relationships observed between rhizosheath size, plant growth, and nitrogen uptake are correlative and do not establish causality. Indeed, as the reviewer correctly notes, it is equally plausible that increased shoot biomass and photosynthetic output in hybrids could lead to greater carbon exudation, thereby promoting rhizosheath development—rather than the rhizosheath being a driver of heterosis.

In response, we have revised the manuscript to ensure that we do not imply a directional or causal relationship based on correlative data.

The revised text as:

“...demonstrated that a significant association between rhizosheath formation, plant growth and nitrogen uptake”

“..... suggesting that rhizosheath formation may serve as an informative trait in the study of microbiome-associated heterosis, though further work is required to determine causal direction.”

Reviewer #4 asked:

1K. Line 270: Same criticism as 1J. It is not rigorous to claim that root traits “strongly drove heterosis of rhizosheath size” from the observation that their MPH values are correlated with each other.

Our response:

Fully agree on this point and has been revised as “associated” in the revision.

Reviewer #4 asked:

1L. Line 278: Same criticism as 1J, this analysis does not show that oxalogluterate influenced lateral root development, only that they are correlated

Our response:

This has been revised as kind of “association” in the revision.

Reviewer #4 asked:

1M. Lines 280-281: I do not follow the logic here - in all these experiments *Massilia* was added experimentally, and rhizosheath size was the response variable. But here, the claim is that the results can be summarized as the rhizosheath “recruiting” *Massilia*.

Our response:

We agree that the original phrasing was misleading, our intention was to convey that *Massilia*'s effects on plant growth and lateral root development were modulated by rhizosheath size, not that the rhizosheath was responsible for actively recruiting the microbe.

We have revised the sentence to clarify this point as:

“In summary, under nitrogen-poor conditions, rhizosheath formation associates with maize heterosis and may enhance the responsiveness to beneficial microbes such as *Massilia*, which in turn influence lateral root development and flavonoid-associated metabolic pathways.”

Reviewer #4 asked:

1N. Line 308: The observation that genotypes differed in N₂O emission does not mean that large rhizosheaths have a causal effect on N₂O emission.

Our response:

We agree that the sentence in its original form overstates the findings by implying that rhizosheath size has a causal effect on N₂O emission. While our results showed that genotypes with larger rhizosheaths tended to have higher N₂O flux under nitrogen-poor conditions, this relationship is correlative, and we cannot rule out other contributing factors such as differences in root respiration, microbial community composition, or root exudate profiles.

To address this, we have revised the sentence to reflect the associative nature of the observation:

“...To test this, we investigated soil N₂O emissions across all genotypes grown in nitrogen-poor soil and found that larger rhizosheath size was associated with reduced N₂O emissions (Supplementary Fig. 17), although causality and underlying mechanisms remain to be determined.”

Reviewer #4 asked:

2. The level of detail provided in the Methods section is not sufficient for understanding what was done, much less being able to replicate the experiments and analyses.

Examples of this problem:

2A. Lines 415-423: One soil was used to generate 4 treatments, but no information is given about how those treatments were implemented (other than holding the D soil at 22% water holding capacity)

Our response:

We thank the reviewer for pointing out the lack of detail in our description of the abiotic soil treatments. We agree that this section required clarification to ensure transparency and reproducibility. In our study, all treatments were derived from a single batch of homogenized field soil (Dikopshof soil) which had been subjected to long-term fertilization trial (over 100 years) with control soil (well fertilized history), low nitrogen soil (with no nitrogen applied) and low phosphorus soil (with no phosphorus applied) and then subjected our defined environmental conditions. We have now added these details to the Methods section to enable replication. We also clarify that all soils came from the same field source and were processed identically prior to treatment application.

Reviewer #4 asked:

2B. Line 468 how were the samples preserved?

Our response:

This information has been provided in the revision.

Reviewer #4 asked:

2C. Line 503-306 what isolates are they, where did they come from, etc.

Our response:

We agree that the description of the bacterial isolates used in the inoculation experiments was insufficient. In response, we have now added detailed information to the Methods section, specifying the identity and origin. Indeed, all isolates were part of a previously published culture collection (He et al., 2024, Nat Plants) and some of them are from an in-house cultivation and collection. We have completed this information in the revision and listed it in Supplemental Dataset 3.

Reviewer #4 asked:

2D. A description of how ASV tables were generated from raw 16S-v4 sequences is completely missing

Our response:

We feel sorry about this unclear information, which has been now fully described in the revised methods.

Reviewer #4 asked:

2E. Mentions of statistical tests such as constrained ordination, PERMANOVA, ANOVA, linear mixed models, etc. are meaningless without also specifying the formulas used, because the interpretation of each variable in the model depends on which other variables were also in the model.

Our response:

We appreciate the reviewer's emphasis on statistical transparency. We fully agree that proper interpretation of multivariate analyses such as PERMANOVA, constrained ordination, ANOVA, and linear mixed models requires clear reporting of the model structure and variables included.

In response, we have updated the Methods section to specify the exact model formulas and fixed/random effects used for each analysis. Specifically:

- For PERMANOVA (via the `adonis` function in the `vegan` R package), we used the model:
Bray-Curtis dissimilarity ~ Germplasm or Genotype with 999 permutations to test the contribution of each factor to microbial community variation.
- For constrained ordination (`capscale`), the formula was:
`capscale(ASV_table ~ Germplasm (inbred line and hybrid)), distance = "bray"`

- For linear mixed-effects models (via the lmer function from lme4), models included germplasm or genotype, and treatment as fixed effects, and replicate/block as a random effect, e.g. `lmer(trait ~ genotype * treatment + (1|block))`
- For ANOVA and post hoc Tukey HSD, we report the factors used (e.g., genotype or treatment) and ensure that pairwise comparisons are clearly indicated in figure legends or supplemental tables.

These details have now been added to the Statistical Analysis subsection of the Methods, and model-specific formulas are provided where relevant.

Reviewer #4 asked:

2F. Basic experimental design information such as the number of replicates per genotype per treatment, randomization, blocking, etc. are missing.

Our response:

We thank the reviewer for this important comment. We agree that key experimental design details—including replication number, randomization procedures, and blocking strategy—were insufficiently described in the manuscript.

In the revised Methods section, we have now clarified the following:

- Pots were fully randomized within phytochamber trays to minimize positional effects, and tray position was rotated weekly.
- Experiments involving multiple genotypes and treatments were conducted in a randomized complete block design (RCBD), with block (tray) included as a random effect in statistical models to account for chamber variation.

These clarifications are now included in the relevant subsections of the Methods (e.g., "Plant growth and sampling", "Soil treatments", and "Statistical analysis").

We also indicate replicate numbers in each relevant figure legend and supplemental dataset.

Reviewer #4 asked:

3. I am concerned about the validity of the analyses comparing heterosis across treatments. Fig. 1d shows 30 observations of MPH per environmental treatment, however, only 10 hybrid genotypes were included in the experiment (Fig. 1a). Typically, MPH is a property of a particular hybrid triplet, and each individual MPH value is estimated using data from several replicates of the F1 genotype as well as both parental genotypes. In other words, it is not clear how 30 MPH values were generated using only 10 hybrids - the independence of those values needs to be justified. If there are indeed 3 measurements of MPH per hybrid, then a standard ANOVA is not sufficient to account for the data structure, in which the 3 MPH values from the same hybrid are expected to be more similar to each other than to the other. Therefore, the 30 values are not a true random sample and need to be modeled using a hierarchical model, e.g., with the hybrid genotype as a random grouping factor.

The same applies to the experiment shown in Fig. S2 - how did you generate 5 independent MPH values per triplet?

Our response:

We thank the reviewer for this careful and important observation regarding the interpretation and statistical treatment of midparent heterosis (MPH) values across treatments. We acknowledge that our original presentation did not clearly explain how the multiple MPH values per hybrid were derived, nor did it account for the non-independence of repeated measurements from the same hybrid.

To clarify:

1. For Figure 1d:

For each of the 10 hybrid genotypes, MPH was calculated separately for each biological replicate using matched replicate data from the hybrid and its two parental inbred lines. This yielded multiple replicate-level MPH values per hybrid per treatment, totaling 30 observations per treatment (3 biological replicates × 10 hybrids).

These replicate-level MPH estimates allowed us to assess within-hybrid variability, but we agree with the reviewer that these are not fully independent observations due to their shared genetic background.

In response to this concern, we have revised our statistical analysis to use a linear mixed-effects model (LMM) with hybrid genotype included as a random effect to appropriately account for the nested structure of the data. Specifically, the model used was:

```
lmer(MPH ~ treatment + (1|hybrid))
```

This model structure appropriately accounts for the fact that replicates within the same hybrid are expected to be more similar to each other than to replicates from other hybrids.

We have also updated the relevant figure legends and the Statistical Analysis section of the Methods to reflect this correction.

We appreciate the reviewer's attention to this issue, which helped us improve the rigor and clarity of our analysis.

```

Linear mixed model fit by REML. t-tests use Satterthwaite's method ['lmerModLmerTest']
Formula: MPH ~ Treatment + (1 | Hybrid_Genotype)
Data: dl

REML criterion at convergence: 862.4

Scaled residuals:
    Min       1Q   Median       3Q      Max
-2.5321 -0.5744 -0.0275  0.6863  2.5208

Random effects:
 Groups             Name             Variance Std.Dev.
Hybrid_Genotype (Intercept) 24.73     4.973
Residual                77.33     8.794
Number of obs: 120, groups: Hybrid_Genotype, 12

Fixed effects:
              Estimate Std. Error      df t value Pr(>|t|)
(Intercept)   15.6837     2.1762  28.3357   7.207 7.11e-08 ***
TreatmentDrought  0.3515     2.2706 104.7523   0.155  0.8773
Treatmentlow N  12.0870     2.2706 104.7523   5.323 5.84e-07 ***
Treatmentlow P  -4.1649     2.2706 104.7523  -1.834  0.0695 .
---
Signif. codes:  0 '***' 0.001 '**' 0.01 '*' 0.05 '.' 0.1 ' ' 1

Correlation of Fixed Effects:
              (Intr) TrtmnD TrtmnN
TrtmntDrght -0.522
Treatmentlow N -0.522  0.500
Treatmentlow P -0.522  0.500  0.500

```

Figure 1d. Results summary using a linear mixed-effects model (LMM) with hybrid genotype included as a random effect to appropriately account for the nested structure of the data. The results showed low nitrogen stress significantly increased the MPH of shoot biomass compared to the control treatment ($p = 5.84e-07$), indicating that heterosis for shoot biomass is more pronounced under nitrogen deprivation

2. For Figure 2s (new as Figure 2a)

For each of the 3 hybrid genotypes, MPH was calculated separately for each biological replicate using matched replicate data from the hybrid and its two parental inbred lines. This yielded multiple replicate-level MPH values per hybrid per treatment, totaling 5 observations per treatment (5 biological replicates).

These replicate-level MPH estimates allowed us to assess within-hybrid variability, but we agree with the reviewer that these are not fully independent observations due to their shared genetic background.

In response to this concern, we have revised our statistical analysis to use a linear mixed-effects model (LMM) with hybrid genotype included as a random effect to appropriately account for the nested structure of the data. Specifically, the model used was:

```
lmer(MPH ~ treatment + (1|Replicate))
```

We have updated the new exact p values based on a linear mixed-effects model (LMM) in new Figure 2a.

We have revised the sentence: "...We determined midparent heterosis for shoot dry biomass and observed that the absence of the soil microbiome significantly enhanced

majority of heterosis in biomass under nitrogen-poor conditions (B73 × Mo17, $p = 0.0012$; Mo17 × H84, $p = 0.024$)”

Reviewer #4 asked:

4. It appears that the “low N” treatment was done by mixing the field soil with 50% v/v river sand, whereas the “high N” treatment was the original field soil with added N fertilizer (lines 494-496). Unfortunately, this seems like a very flawed way to create two treatments that will be informative about the effect of N availability. Undoubtedly these treatments did differ in N, however, by diluting one of them with 50% sand a wide range of other soil properties were likely also altered, such as soil texture, drainage, organic matter, concentration of many other nutrients, possibly pH, etc. In my opinion, this issue greatly compromises the ability to interpret differences between the treatments as “caused by N limitation”.

Our response:

We thank the reviewer for raising this important concern and apologize for the lack of clarity in the original description. Indeed, the protocol we followed involved first mixing the field soil with 50% (v/v) river sand to create a low-nutrient baseline matrix. From this common diluted soil base, we then either added nitrogen fertilizer to create the “high N” condition or left it unfertilized to simulate nitrogen-limited conditions.

We fully agree with the reviewer that this approach introduces changes beyond nitrogen concentration, such as altered soil texture, porosity, and potentially other nutrient dynamics. Therefore, we have revised the manuscript to clearly describe this soil preparation process in the Methods section as:

“To prepare the low nitrogen soil (LN), natural field soil (Campus soil) was first mixed with sterilized washed river sand in a 1:1 (w/w) ratio to create a nutrient-diluted base matrix. For the high nitrogen soil (HN), a nitrogen dose of $50 \text{ mg kg}^{-1} \text{ NH}_4\text{NO}_3$ was added to the same nutrient-diluted soil mixture to simulate nitrogen supplementation.”

Reviewer #4 asked:

5. The flavonoid secretion experiments/Fig. 3d: You observed that the addition of *Massilia* improved the growth of two mutants that are deficient in flavonoid synthesis, but not their wild-type counterparts. From this you conclude that increasing flavonoid secretion is a mechanism by which *Massilia* promotes plant growth. But doesn't this result actually indicate the opposite - that *Massilia* enhances plant growth by some mechanism other than flavonoid secretion? Because the mutants should be incapable of having that particular response, so they should respond less than the wildtypes.

Our response:

We thank the reviewer for this insightful comment. We agree that our original interpretation—that *Massilia* promotes plant growth by inducing flavonoid secretion—is not well described by the results of the mutant experiment. As the reviewer correctly points out, the improved growth in flavonoid-deficient mutants following *Massilia* inoculation suggests that the observed growth promotion occurs independently of flavonoid synthesis, or potentially reflects *Massilia* compensating for the metabolic deficiency in another way. In light of this, we have revised the result section of the manuscript to avoid overstating the role of flavonoid secretion as a mechanism.

We have also clarified this point in the Discussion, emphasizing that while flavonoid pathways were enriched in metabolomic data, the functional validation experiment does not support a model where *Massilia*-driven growth promotion is dependent on plant flavonoid production.

Reviewer #4 asked:

6. The data availability statement does not mention the phenotype data.

Our response:

We had prepared the “Data Availability” session and include the phenotype data and original source data as well.

Minor comments:

Reviewer #4 asked:

7. The term “rhizosheath” needs to be clearly defined early in the paper. How is it different from “rhizosphere”?

Our response:

This information has been provided in the introduction part with references.

“The rhizosheath is considered an adaptive and stress responsive root trait, typically defined as the mass of soil that remains tightly attached to roots upon excavation (McCully, 1999; George et al., 2014). Importantly, the rhizosheath is distinct from the rhizosphere: while the rhizosheath refers to the firmly adhering soil layer bound by root hairs, mucilage, and microbial activity, the rhizosphere encompasses the broader volume of soil influenced by root exudates and associated microbial processes (Mathesius, 2015).”

Reviewer #4 asked:

8. Figure 1e is very confusing. The caption needs to be expanded to clearly explain what the dendrogram shows, what each stacked barplot shows, and what the list of taxa indicates.

Our response:

We thank the reviewer for pointing out the confusion regarding Figure 1e. We agree that the original figure legend did not sufficiently explain the elements of the figure, including the dendrogram structure, the meaning of the stacked barplots, and the list of microbial taxa.

To address this, we have revised the figure caption to provide clear and complete explanations of each component:

Revised caption for Figure 1e:

“Dendrogram showing hierarchical clustering of the phylogenetic relationships among 10 maize inbred lines. Adjacent stacked bar plots indicate the number of microbial ASVs classified as either bacteria (light gray) or fungi (dark gray), and further grouped by their enrichment in the rhizosphere (light orange) or root compartment (dark orange), or by their display of significant midparent heterosis (MPH, dark green) or best-parent heterosis (BPH, light green). The list of taxa above the figure highlights the most

prevalent bacterial families exhibiting microbiome heterosis across genotypes. Each dot represents a microbial taxon showing heterosis in each crossing triplet.”

Reviewer #4 asked:

Line 98: it is not clear how this soil relates to the D, low N, and low P treatments.

Our response:

This has been explained in more detail in the revision.

Reviewer #4 asked:

Line 109: one of the cited studies is from the same species

Our response:

This issue has been fixed as “in maize” in the revision.

Reviewer #4 asked:

Lines 122-123: redundant with previous sentence

Our response:

This has been revised.

Reviewer #4 asked:

Line 128: Clarify that the inbred parents were also included in this experiment, as this is the only way to measure heterosis.

Our response:

This information has been complemented.

Reviewer #4 asked:

Lines 148-149: Unclear writing. Which SynComs promoted lateral root formation, and compared to what?

Our response:

This has been revised.

Reviewer #4 asked:

Line 210: undefined acronyms (VIP, OPLS-DA)

Our response:

This issue has been fixed.

Reviewer #4 asked:

Line 317: Enriched relative to what?

Our response:

This issue has been fixed in the revision.

Reviewer #4 asked:

Line 442: What is a “4 mm field soil mixture”?

Our response:

This issue has been revised.

Reviewer #4 asked:

Line 449: This needs more explanation - how can soil microbes exhibit heterosis when they are mostly haploid, and often reproducing asexually?

Our response:

We agree that this sentence is confusing and had been removed in the revision.

Reviewer #4 asked:

Line 491: How was “the potential to express heterosis under Oxalobacteraceae enrichment” assessed when selecting genotypes for use in this study?

Our response:

This unclear description has been clarified in the revision.

Reviewer #4 asked:

Line 507: Clarify that the inbred lines were also included, as it is impossible to measure heterosis without them.

Our response:

This has been included.

Reviewer #4 asked:

Line 579: There was no previous mention of blocks as part of the experimental design.

Our response:

The block information has been complemented in the revised method.

Reviewer #4 asked:

Line 597: The previous lines indicate the rhizosheath was washed off prior to exudate collection. Please clarify.

Our response:

This has been clarified in the revision.

Reviewer #4 asked:

Line 657: what temperature was the water bath? Was sonication used? Or was this just a soaking procedure?

Our response:

Thanks for pointing this out.

We have complemented more details in Method

for Rhizosheath transplantation experiment as “The roots were then gently washed by hand to remove large aggregates. Next, the roots were placed in a water bath (TRANSSONIC T420, Elma) at room temperature (~20 °C) with high-frequency sound waves (~40kHz) containing 25 ml of autoclaved deionized water for 5 min to remove the majority of the rhizosheath soil.” in the revised manuscript.

Reviewer #4 asked:

Line 683: Indicate type of correlation, e.g. Spearman, Pearson

Our response:

This has been clarified in the revision.

Reviewer #1:

By and large, the authors have adequately responded to my requested revisions. My only remaining concern is still about Figures 4 & S15, where the rhizosheath is still not visible on the images of the whole root system. (In 4a, the roots on the right actually appear to be cleaned; they don't seem to have any visible rhizosheath adhering to them.) I don't feel these parts actually contribute to what they're cited for, and I would rather they just be removed. (The left half of 4a, which does clearly show rhizosheath sizes, is fine to stay)

Our response:

We thank the reviewer for carefully revisiting our revision and for the constructive comment regarding Figures 4 and S15. We fully understand the concern about the visibility of rhizosheath soil in the half of Figure 4a.

Following the reviewer's advice, we have removed the whole root images in Figure 4a as well as Figure S15, since they do not add clear value to the manuscript. The left-hand panels, which clearly illustrate rhizosheath size variation among genotypes, have been retained.

Reviewer #3:

Thank you for addressing all my comments so carefully. Great job!

Our response:

We sincerely thank the reviewer for the positive evaluation and encouraging feedback. We are very glad that our revisions have addressed all concerns.

Reviewer #4:

I appreciate the authors' work to address the concerns I raised in my original review. I did not have time to read and consider the other reviewers' concerns and evaluate whether they were addressed. My biggest concern—the consistent overstatement of results—was mostly addressed. In a few places, the authors still imply a causal relationship that was not supported by their data (examples: line 41, 87-88, 321-322, 464 “rhizosheath-mediated heterosis”). Conversely, in other places they actually understate their results, mainly describing an experimental manipulation (e.g., an inoculation) as “associated with” an outcome (examples: lines 255-256). When a factor was directly manipulated, it is appropriate to say that it caused an effect.

Our response:

We thank the reviewer for carefully evaluating our revisions and for the constructive guidance on balancing our language between correlation and causality. We

acknowledge that in several instances we inadvertently overstated causal implications, and in other cases we understated results of direct experimental manipulations.

In the revised manuscript, we have carefully re-evaluated and adjusted the phrasing throughout:

- **Lines 41, 87–88, 321–322, and 464:** We rephrased these statements to avoid implying causal relationships where our data are correlative, e.g., replacing “rhizosheath-mediated heterosis” with more precise wording (“rhizosheath traits associated with heterosis”).
- **Lines 255–256 and related passages:** We now explicitly describe causal relationships where experimental manipulations (e.g., inoculations) were performed and resulted in measurable effects. Thus, we changed “associated” to “led to” in the revision.

Overall, we also checked through the whole manuscript, and revised the overstatements for line 155-157, line 236-237, line 245-246, line 275-276, line 369-370, line 411-415, line 429, line 443-445 in the revised manuscript.

I found the revised version harder to read than the original—it would benefit from substantial re-structuring to avoid repetition and clearly explain the differences between experiments. For example, the clause “To determine whether *Massilia* influences heterosis of root development under nitrogen-poor conditions...” (or slightly re-worded) appears 3 times in quick succession (lines 156, 175, 201) - and each time, it is used to introduce a different experiment or analysis.

Our response:

We thank the reviewer for this valuable observation. We agree that the current version introduces multiple experiments with similar phrasing, which can feel repetitive and may obscure the differences between the experimental designs. In the revised manuscript, we have restructured these sections to:

1. **Streamline introductions to experiments** by reducing repetitive phrasing.
2. **Clarify the logical flow** — moving from sterilization experiments, to inoculation, to trait-level analysis, and then to metabolomics.
3. **Explicitly highlight what is novel in each experiment** and how it builds on the previous one.

Specifically, we replaced the repeated introductory sentences (former lines 156, 175, and 201) with distinct phrasings:

- L156: *“Building on our earlier finding that Massilia promotes lateral root formation in mutants, we next asked whether this isolate could also affect heterosis in root development under nitrogen-poor conditions.”*
- L175: *“To further dissect the specific role of Massilia, we conducted additional pot experiments across genetically diverse hybrid combinations and soil types.”*
- L201: *“Finally, to test whether single Massilia inoculation impacts heterosis in a broader set of root traits and nitrogen acquisition, we quantified multiple architectural traits across genotypes.”*

Moreover, we have checked through the whole results parts, and restructured with more logic and explained the differences and objectives of each experiment for line 142-244, line 179-181, line 205-210 in the revision.

Additionally, there are several extremely long paragraphs. Revision is needed to organize the content in a more logical way.

Our response:

We thank the reviewer for this constructive suggestion. We agree that several paragraphs in the Results and Discussion were overly long and contained multiple points, which may reduce readability. In the revised version, we have:

1. **Shortened long blocks of text** into 2–3 smaller paragraphs, each focused on a single experiment or idea.
2. **Reorganized the Results** to clearly indicate the progression from (i) soil sterilization, (ii) Massilia inoculation, (iii) additional genotypes and root traits, (iv) metabolomics, (v) rhizosphere association, and (vi) N₂O emissions.
3. **Improved the Discussion flow** by separating correlation/association findings from mechanistic interpretations, avoiding large multi-idea paragraphs.

Some of the conclusions seem a bit circular. For example, does Massilia induce apigenin or does apigenin promote Massilia colonization? Both are implied almost interchangeably throughout the paper. I found the mechanistic diagram in Figure 6 to be quite vague and it did not really help answer my questions about the apparent circularity of the relations between Massilia, rhizosphere, and flavonoids.

Our response:

We thank the reviewer for highlighting this important point. We agree that our manuscript may have conveyed circularity in describing the interactions among *Massilia*, flavonoids, and rhizosphere formation. Our current data and published results provides evidence for **both directions** of interaction:

1. *Host derived apigenin promotes Massilia colonization* — as shown previously (Yu et al. 2021, *Nat. Plants*; He et al., 2024, *Nat. Plants*), flavones recruit Oxalobacteraceae including *Massilia*.
2. *Massilia inoculation affects flavonoid profiles* — in this study, inoculation increased apigenin levels in root exudates (Fig. 3a).

However, we do not claim that one strictly causes the other in a unidirectional pathway. Instead, our results support a **feedback model**: root exudates (flavonoids) shape microbial recruitment, while colonization by *Massilia* in turn alters root metabolism and exudation patterns, reinforcing rhizosheath development.

To address the reviewer's concern:

- We have **revised the whole Results and Discussion** to explicitly state that our data suggest a *feedback loop* rather than a single causal direction, and also propose further perspective on this aspect of functional role of plant and microbial metabolism.
- We have **clarified Figure 6**, replacing the linear arrows with bidirectional arrows and adding explanatory text in the legend to highlight that the relationships are reciprocal and not fully disentangled.

I request a clearer and more thorough discussion of whether there is any evidence at all, in this work or in other published work, that the rhizosheath is a mechanism of heterosis, or somehow more important than other plant traits for the manifestation of heterosis. My a priori expectation is that it is just another trait that is better in hybrids than inbreds, like height or yield. But no one is arguing that height is a “mechanism” of heterosis even though bigger plants can do more photosynthesis, etc.

Our response:

We thank the reviewer for raising this critical point. We agree that our current phrasing may give the impression that rhizosheath formation is a *mechanism* of heterosis, while the available evidence is more consistent with rhizosheath being a *trait associated with heterosis*. In other words, larger rhizosheaths in hybrids may reflect underlying differences in root architecture and exudation that also support heterotic growth performance, rather than being the causal driver of heterosis itself.

In the revised Discussion, we now:

1. **Explicitly acknowledge the reviewer's point** that rhizosheath, like plant height or biomass, may represent a correlated trait rather than a mechanistic basis of heterosis.

2. **Clarify the evidence** from our study: rhizosheath size is larger in hybrids, correlates with lateral root density and biomass heterosis, and is responsive to microbial inoculation. However, these results do not prove causality.
3. **Contextualize with literature**: previous studies have established rhizosheath as a stress-adaptive trait linked to nutrient and water acquisition (He et al. 2025 *Trends Plant Sci.*; Xu et al. 2023 *Nat. Commun.*), but no published work has demonstrated that rhizosheath is a mechanism of heterosis.
4. **Reframe our interpretation**: we now describe rhizosheath as a *promising phenotypic indicator* of belowground heterosis, particularly under nitrogen deprivation, and as a trait that may help dissect underlying genetic and microbial interactions — but not as a proven mechanistic driver.

Figure 1e is somewhat improved but still unclear to me. 1) It seems to imply that the different host genotypes had different numbers of ASVs (shown by differing bar lengths), is that correct? 2) The bars for “Kingdom” are the same length as the bars for “Heterosis type”, implying that 100% of all ASVs showed heterosis. 3) Heterosis is a property of a hybrid, not an inbred line, and the genotypes in the dendrogram are inbred lines - so how should we interpret the “heterosis type” panel? 4) I can’t tell what the “compartment” panel is showing. What is meant by “enrichment in the rhizosphere or root compartment” - does this just show whether the ASV’s relative abundance was higher in roots or rhizosphere? Please add a clearer explanation. 5) What is meant by “most prevalent bacterial families exhibiting microbiome heterosis across genotypes” (line 947 referring to the rightmost panel)

Our response:

We thank the reviewer for carefully examining Figure 1e and for raising important points of clarification. We agree that the current presentation was not sufficiently clear, and we have revised both the figure and its legend for better interpretation.

1. **Bar lengths across genotypes (ASV numbers).**
The varying bar lengths indeed represent the number of ASVs detected in each hybrid triplet. We now explicitly state this in the legend.
2. **Bars for “Kingdom” vs. “Heterosis type.”**
We apologize for the confusion. The intention was not to suggest that 100% of ASVs showed heterosis. Rather, the “heterosis type” bars indicate the number of ASVs that exhibited midparent or better-parent heterosis. We have revised the legends to highlight this point in the legend.
3. **Heterosis applies to hybrids, not inbreds.**
The reviewer is correct. The dendrogram shows inbred parental genotypes, but the “heterosis type” panel summarizes ASVs evaluated in their corresponding

hybrids crossed with the “common maternal genotype B73”. We now clarify in the legend.

4. Compartment panel (root vs. rhizosphere).

We revised the legend to clearly state: “..... ASV showed greater relative abundance in root tissue or in rhizosphere soil.”

5. Rightmost panel (prevalent families).

We have clarified the wording in the legend: “The rightmost panel highlights the most frequently enriched bacterial families across all hybrids that displayed microbiome heterosis, providing a summary of lineages most consistently involved.”

Despite these lingering concerns, I think the dataset is a very valuable one and the authors describe some extremely interesting patterns that are certainly worth publishing. After some additional revision, I do think this paper will advance the field and inspire a lot of follow-up work on the involvement of microbes in heterosis.

Our response:

We sincerely thank the reviewer for the positive evaluation and encouraging perspective on our work. We greatly appreciate the recognition that our dataset is valuable and that the patterns we describe will advance the field of plant–microbe interactions and heterosis.

Some minor things:

Lines 229-230: What do these percentages refer to?

Our response:

We thank the reviewer for noting this ambiguity. The percentages refer to the proportion of identified metabolites that were annotated to each KEGG pathway category, which has been modified in the revision.

Line 236: What does “assembly of metabolic features” mean?

Our response:

We We thank the reviewer for pointing out this unclear wording. By “assembly of metabolic features,” we intended to describe the overall composition and relative abundance structure of the detected metabolites under different soil treatments. To avoid ambiguity, we have revised the sentence as follows: “.....altered the overall composition and relative abundance patterns of detected metabolites.....”

Line 242: How were those cutoffs for L2FC determined?

Our response:

We thank the reviewer for this question.

Here indeed has one typo, which should be 0.8, not 0.83.

The cutoffs for fold change (≥ 1.2 or ≤ 0.8) were chosen based on standard practice in metabolomics studies to balance sensitivity and stringency. A fold change of 1.2 (or its reciprocal, 0.8) corresponds to a 20% change in relative metabolite abundance, which is commonly used as a biologically meaningful threshold in non-targeted metabolomics when combined with statistical significance ($p < 0.05$) and model-based VIP ≥ 1 . These criteria together help to minimize false positives while capturing robust differential metabolites. We have clarified this in the Methods.

Lines 376-377: positive R indicates a positive correlation, not negative

Our response:

Thanks for this point and we have revised the text and figure accordingly. Here, indeed it is negative correlation.

Line 414: nitrogen use efficiency has a specific definition and I do not see any prior mention to this metric

Our response:

We thank the reviewer for pointing this out. We agree that nitrogen use efficiency (NUE) has a specific agronomic definition, which was not directly quantified in our study. To avoid confusion, we have revised the wording to describe our findings more precisely. Instead of referring to NUE, we now state that Massilia-associated root traits and rhizosheath formation were linked to “enhanced nitrogen uptake” and biomass accumulation under nitrogen-poor soil. This phrasing avoids implying that we measured NUE in the formal agronomic sense.

Line 453: “expression of genes” appears to be referencing metagenomics data, which describes the abundances of functional gene groups in the combined genomes of the microbiome - it says nothing about the expression of those genes

Our response:

We thank the reviewer for pointing out this misstatement. We agree that metagenomic data reflect the relative abundance of functional gene groups within the microbial community, not their transcriptional activity. To correct this, we have revised the text to avoid implying gene expression and instead accurately describe gene abundance inferred from metagenomic annotation.

Line 699-700: Provide more detail about root hair measurement. How many hairs were measured per root, from what part of the root, etc.

Our response:

We thank the reviewer for this suggestion. We agree that our original description of root hair measurement was insufficiently detailed. We have now expanded the Methods to specify the sampling strategy, number of root hairs measured per root, and how averages were calculated.